# DNA polymerase ε harmonizes topological states and R-loops formation to maintain genome integrity in Arabidopsis

Qin Li[1], Jincong Zhou [1,2,6], Shuai Li[1,2,6], Weifeng Zhang [1,2,6], Yingxue Du[1,2,6], Kuan Li[1,2,3,6], Yingxiang Wang [4,5] & Qianwen Sun [1,2] ✉

Genome topology is tied to R-loop formation and genome stability. However, the regulatory mechanism remains to be elucidated. By establishing a system to sense the connections between R-loops and genome topology states, we show that inhibiting DNA topoisomerase 1 (TOP1i) triggers the global increase of R-loops (called topoR-loops) and DNA damages, which are exacerbated in the DNA damage repair-compromised mutant *atm*. A suppressor screen identifies a mutation in POL2A, the catalytic subunit of DNA polymerase ε, rescuing the TOP1i-induced topoR-loop accumulation and genome instability in *atm*. Importantly we find that a highly conserved junction domain between the exonuclease and polymerase domains in POL2A is required for modulating topoR-loops near DNA replication origins and facilitating faithful DNA replication. Our results suggest that DNA replication acts in concert with genome topological states to fine-tune R-loops and thereby maintain genome integrity, revealing a likely conserved regulatory mechanism of TOP1i resistance in chemotherapy for ATM-deficient cancers.

The topological organization of a genome is highly associated with DNA transaction activities including replication, transcription, repair, and other cellular processes[1,2]. DNA topology changes dramatically during cell division, especially in DNA replication[3,4]. Positive supercoils arise ahead of replication forks and regulate the entire replication process, while aberrant replication contributes substantially to genome instability if supercoils are not unwound properly[5].

R-loops are usually co-transcriptionally formed on negative DNA supercoils behind RNA polymerases[6–12]. R-loops thus acting as barriers hinder DNA replication and even induce DNA breaks at common fragile sites[13]. RNA and DNA polymerases occurring at the same genomic loci could lead to transcription and replication collisions (TRCs) that are normally associated with constant and stabilized R-loops. In addition, R-loops can stall or even stop and backtrack RNA polymerases, in

which endonucleases XPG and XPF in the transcription-coupled nucleotide excision repair (TC-NER) pathway can generate single-stranded DNA gaps and eventually induce DNA double-strand breaks (DSBs)[14,15]. Thus, in vivo R-loop levels need to be faithfully modulated by numerous factors, such as RNase H, helicase, chromatin remodeler and DNA topoisomerase[16–18].

DNA topoisomerases regulate topological events like DNA supercoils derived from replication and transcription[3]. Topoisomerase 1 (TOP1) plays a vital role in resolving either positive supercoils ahead of the DNA replication forks and transcription machinery, or negative supercoils behind them[3,19]. TOP1 relieves DNA torsions by nicking ssDNA, forming transient TOP1 cleavage complexes (TOP1cc), and re-ligating the nicks after a controlled rotation[20–22]. Camptothecin (CPT) and its analogs are TOP1-specific inhibitors (TOP1i), blocking the TOP1

[1]Center for Plant Biology, School of Life Sciences, Tsinghua University, Beijing 100084, China. [2]Tsinghua-Peking Center for Life Sciences, Beijing 100084, China. [3]Chinese Institute for Brain Research, Beijing 102206, China. [4]College of Life Science, South China Agricultural University, Guangdong Laboratory for Lingnan Morden Agriculture, Guangzhou 510642, China. [5]State Key Laboratory of Genetic Engineering, Institute of Plant Biology, School of Life Sciences, Fudan University, Shanghai 200438, China. [6]These authors contributed equally: Jincong Zhou, Shuai Li, Weifeng Zhang, Yingxue Du, Kuan Li. ✉e-mail: sunqianwen@mail.tsinghua.edu.cn

re-ligation activity and accumulating persistent TOP1cc that threatens genome stability[23]. Moreover, topoisomerases are major chemotherapeutic targets and TOP1i is particularly effective in treating breast and cervical cancer[24]. In homologous recombination repair (HRR), the mutation of breast cancer susceptibility genes, such as *BRCA1* and *ATM* (ataxia-telangiectasia mutated protein), leads to hypersensitivity to clinical DNA-damaging drugs, whereas various susceptibilities to chemotherapy agents of different genetic backgrounds challenge current strategies for tumor treatment.

TOP1 activity could influence R-loop levels. In human cells, TOP1 depletion stimulates R-loops at both promoters and terminators of long, highly expressed genes[25]. DSBs also accumulate at transcription termination sites (TTS), activating persistent ATM or ataxia telangiectasia and Rad3-related protein (ATR) and impeding global replication fork progression[25,26]. Dividing cells show hyper-cytotoxicity to CPT, whereas R-loop-induced TRCs and TC-NER are major sources of DSBs that threaten genome stability[15,27]. Because TOP1 is essential for controlling genome topology during replication and transcription, dysfunction of TOP1 promotes R-loop accumulation and genome instability. However, the global effects of DNA topological stress on R-loop dynamics and replication alteration, and the mechanisms by which resulting R-loops and replication stress threaten genome stability, are not well investigated. Also, it is unclear how TOP1i contributes to different phenotypes of cells displaying DNA repair defects.

In this study, we show that TOP1i globally increases R-loops (termed 'topological R-loops', or topoR-loops) on active transcription sites and DNA replication origins, which causes replication stress and DNA damage. ATM controls both homologous recombination (HR) and non-homologous end joining (NHEJ) pathways to repair topoR-loop-induced DNA damage. Furthermore, through a suppressor screen, we find that a mutation in POL2A, the catalytic subunit of DNA polymerase ε engaged in leading-strand synthesis[28], is resistant to TOP1i and rescues the defects of R-loop accumulation and genome instability in *atm*. Additionally, we find a conserved junction domain between the exonuclease and polymerase domains of POL2A that ensures proper DNA replication under topological stress and limits the misincorporation of ribonucleotide to safeguard replication fidelity. Mutations in this conserved fidelity domain of POL2A neglect the mistaken ribonucleotide, and thus accelerate replication in *atm* with TOP1i. Our studies reveal new roles of replication stress controlled by DNA polymerase ε in TOP1-dependent R-loop modulation and genome stability. The conserved junction domain and the resistance to TOP1i of *pol2a* raise potential directions for TOP1i therapeutic resistance in ATM-deficient tumors.

## Results

### TOP1i triggers topological R-loop accumulation and genome instability

CPT specifically inhibits the activity of TOP1 at highly conserved sites, typically stimulating R-loops in genomes[17,25,26,29–31]. To investigate how DNA topological change influences R-loops, we applied CPT treatment in different genetic backgrounds of *Arabidopsis thaliana* (Fig. 1, Supplementary Fig. 1). We found the root growth of wild-type Col-0 was significantly inhibited in the presence of 30 nM CPT compared to the control treated with DMSO, a solvent used to dissolve CPT (Supplementary Fig. 1a, b). Interestingly, the root growth of *atm*, the mutant of a key transducer in the DNA repair pathway, was significantly restrained upon CPT treatment compared to Col-0 (Fig. 1a). However, the root growth of *atr*, another protein kinase required for DNA damage repair[32], was much more resistant than *atm* (Supplementary Fig. 1a, b), suggesting a specific defect of TOP1i in the *atm* mutant. Additionally, by staining the root of Col-0 with propidium iodide (PI), a counterstain marking dead cells, we found that CPT treatment could induce the death of root stem cells and reduce the length of meristem zones (Fig. 1b, Supplementary Fig. 1c). These growth defects were

intensified in *atm* mutant, as the root tips of *atm* showed severe damage and disorganized morphology, while cell death in *atr* was comparable to that in Col-0 (Fig. 1, Supplementary Fig. 1a–c).

To illustrate whether the defects of root growth caused by CPT were related to R-loop changes, we detected total R-loop levels by slot-blot assay[33]. The results showed slightly elevated R-loops in *atm* relative to wild-type Col-0 (Fig. 1c). Upon CPT treatment, R-loop levels increased in Col-0, and increased dramatically in *atm*, to a much higher level (Fig. 1c). To further dissect genome-wide R-loop changes in response to CPT treatment, we performed ssDRIP-seq (a single-stranded DNA ligation-based library construction from DNA:RNA hybrid immunoprecipitation, followed by sequencing)[34,35] in Col-0 and *atm* with and without CPT treatments. The results of principal components analysis showed that these samples were clustered in replicates and the *atm*-CPT sample presented distinctive R-loop patterns compared to others, consistent with the results of the slot-blot assay and the severe root developmental phenotype of CPT-treated *atm* (Supplementary Fig. 2a). Further analysis of R-loops on all annotated elements observed that *atm* showed a higher level of R-loops near TSS than Col-0 (Supplementary Fig. 2b, c). Interestingly, TOP1i promoted accumulations of R-loops located on TSS in Col-0, while elevating the R-loops on gene body in *atm* (Supplementary Fig. 2b, c). Moreover, a large number of R-loops were remarkably increased on genes in *atm* after CPT treatment, but more genes showed decreased R-loops in Col-0 after CPT treatment (Supplementary Fig. 2e). We next intersected the significantly up-regulated or down-regulated genes in *atm* and Col-0 after CPT treatment. There are little overlapped genes that are commonly regulated in *atm* and Col-0 after CPT treatment (Supplementary Fig. 2f), indicating R-loop dynamics triggered by CPT vary in different backgrounds. Since TOP1i results in topological conformation change, we named these CPT-induced R-loops 'topological R-loops' (topoR-loops in short). In addition, our analysis showed that the topoR-loops do not correlate with mRNA levels after CPT treatment (Supplementary Fig. 3), suggesting that they do not result from gene expression differences.

As increased R-loop levels could lead to DNA damage, we next investigated the levels of DNA damage mark γH2AX (phosphorylation states of H2AX[36,37]) with western blot and immunostaining. There is a tendency, but not statistically significant, towards the increase of γH2AX accumulation in Col-0 and *atm* after CPT treatment by western blot, while the γH2AX levels of *atm* in the control condition showed a decreasing trend compared to Col-0 (Fig. 1d). The immunostaining results confirmed that CPT increased γH2AX signals in Col-0 and *atm* root tips (Fig. 1e, Supplementary Fig. 1d), which is consistent with its phosphorylation activity of H2AX. These results reveal that TOP1i could boost R-loop levels in Arabidopsis, the root defects might be ascribed to the R-loop-induced DNA damage and subsequent cell death, and these responses are enhanced in the DNA repair mutant *atm* (Fig. 1f).

### AtRNH1A overexpression releases topological stress

RNase H1 proteins are evolutionarily conserved R-loop removers that specifically degrade the RNA moiety of RNA:DNA hybrids[38,39]. AtRNH1A is one of three RNase H1 proteins in Arabidopsis, localized in the nucleus[33]. The recombinant protein AtRNH1A has RNA:DNA cleavage activities in vitro, and a mutation D160N in the RNase H domain could disturb its catalytic activity (Supplementary Fig. 4). To validate the roles of CPT-induced topoR-loops in root growth inhibition, we introduced an estrogen-inducible AtRNH1A expression vector into Col-0 and *atm*, respectively, to modulate R-loop levels.

Upon estrogen treatment for the FLAG-tagged AtRNH1A transgenic plants, FLAG-fused AtRNH1A proteins were accumulated (Fig. 2c), and inducible overexpression of functional AtRNH1A[WT] could partially rescue the CPT-mediated defects of root development in both Col-0 and *atm* (Fig. 2a–c). However, inducible overexpression of

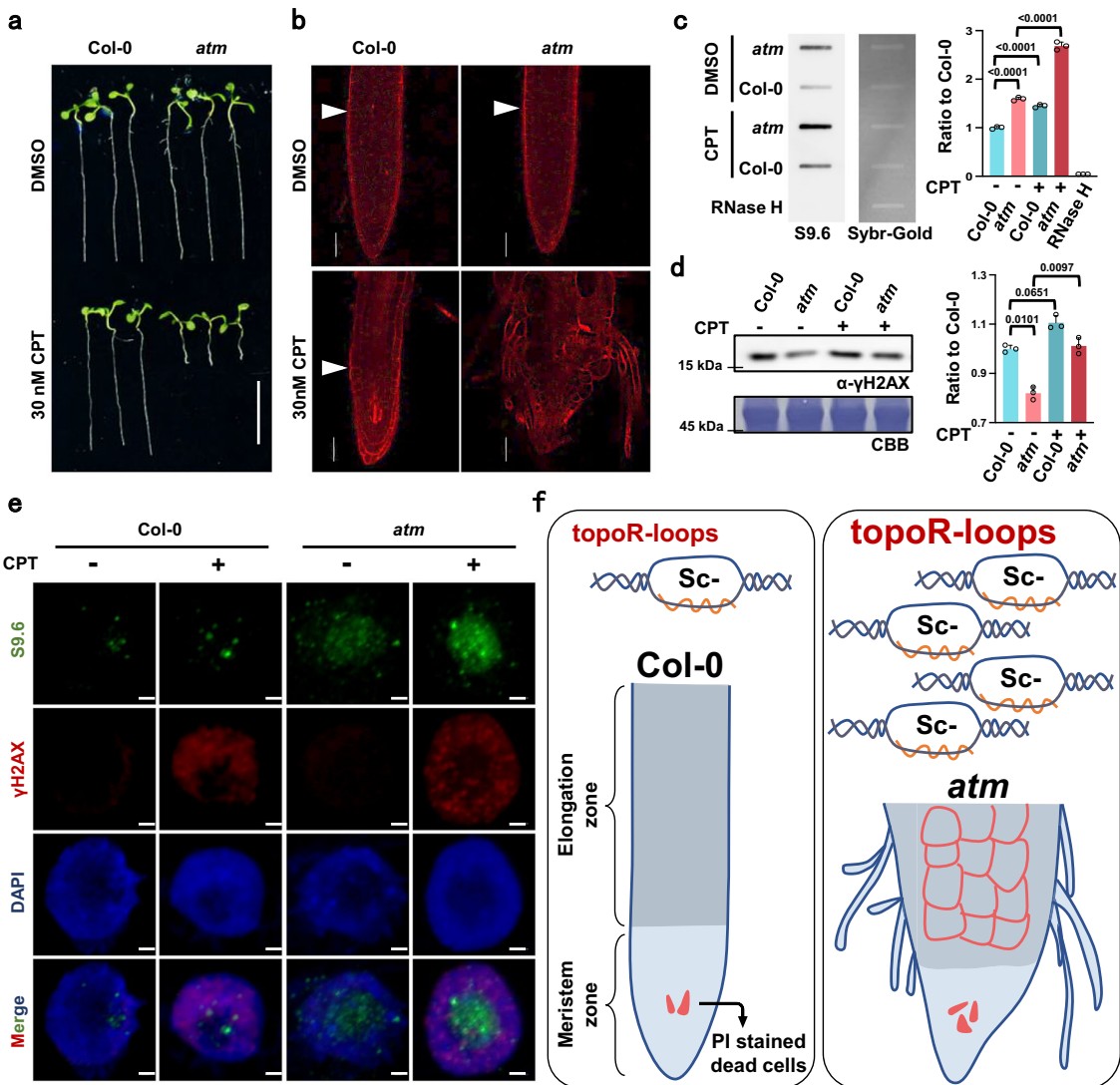

**Fig. 1 | *atm* is hypersensitive to TOP1i and accumulates R-loops and DNA damage. a** Phenotypes of plants root treated with DMSO (control) or 30 nM CPT (TOP1i) for 5 days. Scale bar: 1 cm. **b** Root meristem structures of 5-day-old Col-0 and *atm* seedling root tips stained with PI. The white triangle indicates the boundary of the meristematic zone and elongation zone. Scale bar: 50 μm. Experiments were repeated three times with similar results. **c** 50 ng genomic DNA of Col-0 and *atm* root tips (with and without 2 h of 1 μM CPT treatment) was detected by slot-blot assay, antibody S9.6 was used to detect R-loop signals. The SYBR-gold staining was used as the loading control. For the statistical analysis, band signals of each sample are calculated three times with Image J. Bars in the plot represent mean ± SD (one-way ANOVA). **d** The DNA damage level of Col-0 and *atm*

root tips (with and without 2 h of 1 μM CPT treatment) was detected by γH2AX western blot. Membrane stained with CBB (Coomassie Brilliant Blue) was used as a loading control. Band signals of each sample are calculated three times with Image J. Bars in the plot represent mean ± SD (one-way ANOVA). **e** Immunolabeling of root tip nuclei using R-loop antibody S9.6 and DSB marker γH2AX. The seedlings were treated with and without 2 h of 1 μM CPT. Scale bar: 1 μm. Experiments were repeated three times with similar results. **f** Schematic model for TOP1i threatening genome stability in the root of Col-0 and *atm*. After CPT treatment, topoR-loops increased and thus caused root stem cell death and root growth inhibition (in Col-0). In the absence of ATM, aberrant accumulation of topoR-loops resulted in severe root cell death and destruction. Source data are provided as a Source Data file.

mutated AtRNH1A^(D160N) caused shorter roots and more severe root phenotypes (Fig. 2a–c). PI staining revealed AtRNH1A^(WT) overexpression reduced cell death and meristem zone inhibition, whereas mutated AtRNH1A^(D160N) overexpression intensified stem cell death and even caused aberrant root morphology (Fig. 2d).

We then performed further analysis of R-loop and DNA damage levels with S9.6 and γH2AX immunostaining (Supplementary Fig. 5). We detected increased S9.6 and γH2AX signals after CPT treatment compared to the DMSO condition, and AtRNH1A^(WT) overexpression decreased S9.6 and γH2AX levels (Fig. 2e, f). Notably, inducible AtRNH1A^(D160N) overexpression promoted R-loop accumulation in *atm* (Fig. 2e), and γH2AX levels also increased (Fig. 2f). These data indicated that TOP1i-promoted accumulation of topoR-loops which can be

globally removed by functional AtRNH1A^(WT) is what endangers genome stability. In contrast, when overexpressing the mutated AtRNH1A^(D160N), the efficiency of R-loop clearance was blocked, thus increasing topoR-loop accumulation artificially, which could further exacerbate the topological stress and subsequent root growth defects in *atm*.

### Identification of the *atm* suppressor *asr20*

TOP1i-promoted topoR-loops were significantly accumulated in the *atm* mutant (Fig. 1c), and caused acute and visible defects in root growth (Figs. 1, 2), suggesting a tight connection among root length, R-loop levels and topological states under TOP1 inhibition (Fig. 1f). The visible defects of root growth promoted us to use Arabidopsis root as the sensor to investigate the connections between R-loop levels and

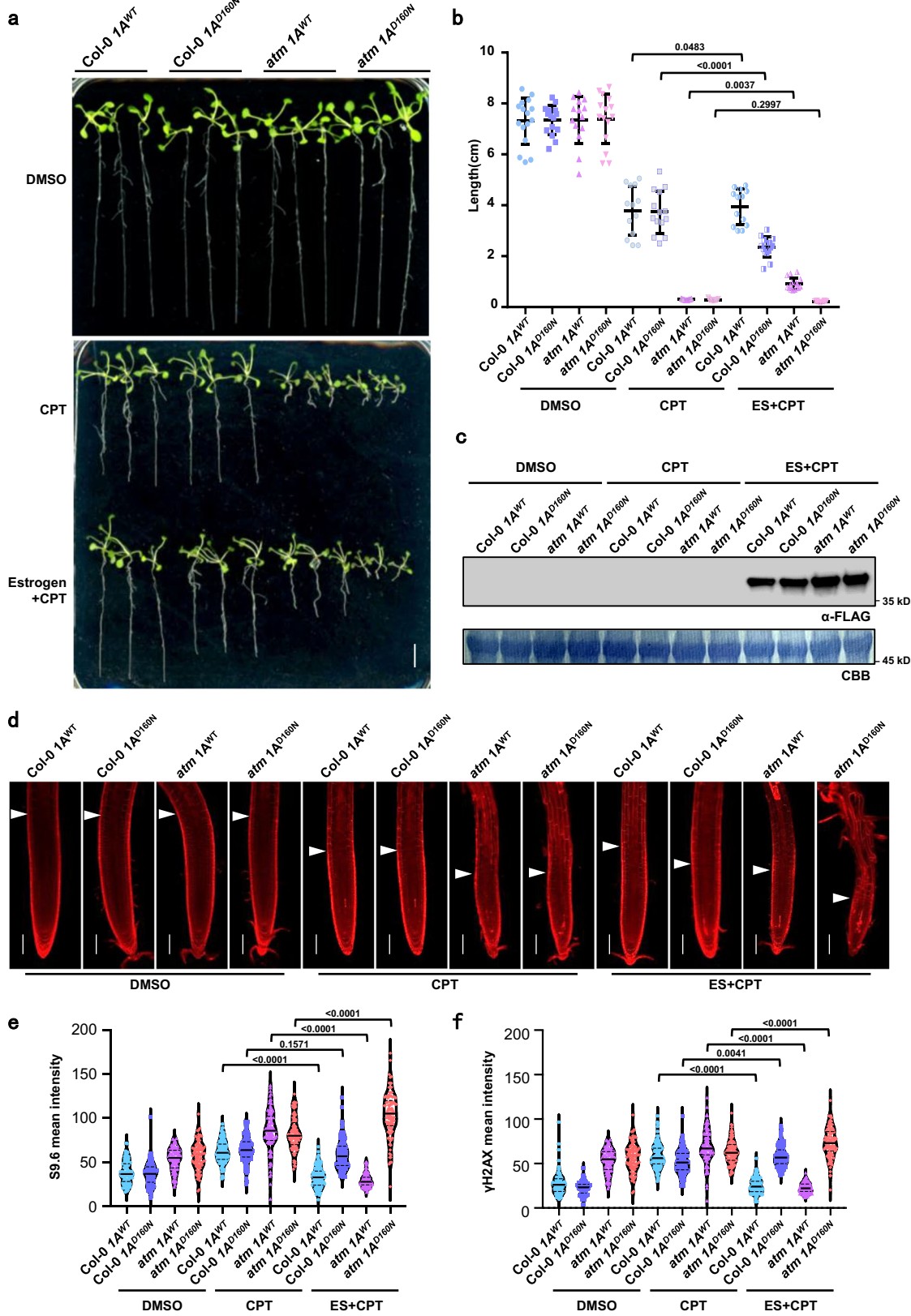

genome topology states (Fig. 1f). We then conducted a forward genetic screen to isolate suppressors of *atm* under TOP1i (Fig. 3a). Among isolated <u>a</u>tm <u>s</u>hort-<u>r</u>oot recovery (*asr*) mutants, we focused on the first isolated suppressor mutant *asr20*, whose root length was significantly recovered in CPT treatment when compared with *atm* (Fig. 3b, c). In addition, the morphogenesis of the root tip in *asr20* is comparable to

that in Col-0, with normal root shape and less CPT-induced cell death compared to *atm* (Fig. 3d). To investigate the R-loop levels in *asr20*, we performed slot-blot assay and found less global R-loop accumulation in the nuclei of *asr20* relative to *atm* (Fig. 3e). Additionally, the γH2AX level was higher in *asr20* than in *atm*, and further increased after CPT treatment (Fig. 3f). We also performed immunostaining of S9.6 and

**Fig. 2 | AtRNH1A^WT and AtRNH1A^D109N alleviate and reinforce the inhibition of CPT, respectively. a** Root phenotypes of 15-day-old Col-0 and *atm* with estrogen-induced AtRNH1A^WT (abbreviated as 1A^WT) and AtRNH1A^D160N (abbreviated as 1A^D160N) overexpression. After germination (~3 days) on 1/2 MS medium, the plants were transferred to 1/2 MS medium containing 50 μM β-estradiol (abbreviated as ES) and induced for 36 h, further grown on 1/2 MS medium containing DMSO, 30 nM CPT or 30 nM CPT with 50 μM β-estradiol for 10 days. Scale bar: 1 cm. **b** The quantitative data of root lengths for Fig. 2a. Lines represent mean ± SD, *n* = 15 (one-way ANOVA). **c** Western blot analysis of 15-day-old seedlings corresponding to Fig. 2a, with FLAG antibody. CBB staining was used as the loading control. **d** PI staining corresponding to Fig. 2a. The white triangle indicates the boundary of the meristematic zone and elongation zone. Scale bar: 100 μm. **e** R-loop levels indicated by S9.6 in mutants related to Supplementary Fig. 5. Mean intensity was evaluated using Image J. Lines represent mean ± SD, *n* = 100 (one-way ANOVA). **f** DNA damage levels indicated by γH2AX in mutants related to Supplementary Fig. 5. Mean intensity was evaluated using Image J. Lines represent mean ± SD, *n* = 100 (one-way ANOVA). Source data are provided as a Source Data file.

γH2AX to further validate the R-loop and DNA damage levels in *asr20* (Fig. 3g, h), and found that the R-loops in *asr20* were increased under CPT treatment accompanied with more γH2AX (Fig. 3e−h). This suggests that both the *asr20* double mutant and the topoR-loops could trigger H2A.X phosphorylation which is most possibly ATR-dependent.

We re-sequenced the *asr20* mutant population and located the mutation in chromosome 1 (Supplementary Fig. 6a)[40], and found there were 5 genes with mutations in the regions (Supplementary Fig. 6a). After allelic crossing with mutants of candidate genes with *atm*, we verified that *abo4-1*(G522N) and *til1-4*(G469R) could rescue CPT-induced defects in *atm* (Supplementary Fig. 6b−f). Both *abo4-1* and *til1-4* are weak alleles of *POL2A*, and their mutation sites are closed to *asr20* (L473F) (Supplementary Fig. 6b). These three mutants have mutations at highly conserved positions among model species (Supplementary Fig. 6g). Mutant *abo4-1*(G522N) is mutated inside the DNA polymerase domain of POL2A (closed to the left border), while *til1-4* (G469R) and *asr20* (L473F) carry mutations in the junction domain between exo-nuclease and DNA polymerase domains, within a very conserved region (Supplementary Fig. 6g). The root lengths of *atm*/*abo4-1* and *atm*/*til1-4* are significantly longer than that of *atm* in the presence of CPT (Supplementary Fig. 6c−f). Interestingly, *asr20* and *atm*/*abo4-1* were blocked with an increasing concentration of CPT (Supplementary Fig. 7a, b), whereas both *pol2a^L473F* − the *pol2a* single mutant separated from *asr20* − and *abo4-1* showed significant resistance to CPT compared with Col-0 (Supplementary Fig. 7a, b). In addition, the root length of *pol2a^L473F* was longer than that of *abo4-1* and had fewer dead cells under high a concentration of CPT (Supplementary Fig. 7a−c), and *asr20* showed a more structured root morphology compared with *atm*/*abo4-1* (Supplementary Fig. 7c). These results suggest the conserved junction domain of POL2A should have an essential role in modulating topological stress during replication, and residue mutations in the junction and DNA polymerase domain of POL2A have conserved effects of varying degrees on the release of topological stress and then the topoR-loops, which is in line with the location of the mutations in the functional domain of the POL2A protein (Supplementary Fig. 6g).

## TopoR-loops globally changed and especially decreased near the replication origins in *asr20*

Results from ssDRIP-seq suggested that global R-loop levels near TSS decreased significantly in *asr20* but increased on gene body compared with *atm*, and CPT treatment slightly increased global R-loop levels (Fig. 4a). Interestingly, *pol2a^L473F* had lower R-loop levels on TSS than Col-0 and *atm*, but higher levels on gene body, while had overall higher levels than *asr20* (Fig. 4a). CPT elevated and slightly decreased the R-loop levels on gene body and TSS respectively in *pol2a^L473F* (Fig. 4a). Further analysis observed that R-loop levels at protein-coding genes decreased significantly in *asr20* compared with Col-0 and *atm*, and CPT treatment had little additive effect on increasing R-loop levels (Supplementary Fig. 8a). *pol2a^L473F* had lower overall R-loop levels than Col-0 and *atm*, but higher levels than *asr20*, and CPT elevated the R-loop levels(Supplementary Fig. 8a). The antisense R-loops that enriched in the transcription start sites resembled trends of un-stranded R-loop variations(Supplementary Fig. 8a). However, the sense R-loops decreased only near TTS sites in *pol2a^L473F*, and the extent of the decrease was magnified in *asr20* (Supplementary Fig. 8a). As indicated in the scatterplot shown in Fig. 4b, R-loop levels were significantly altered in only a few protein-coding genes in *asr20* after CPT treatment, and the number of differential R-loops was lower compared with *pol2a^L473F*. Nonetheless, compared with *atm*, a large number of R-loops on coding genes in *asr20* showed significant increases or decreases under DMSO control (Fig. 4b); this suggests POL2A functions in maintaining genome R-loops, which might be partially dependent on ATM.

To explore the features of differential R-loops on genes in *asr20*, we intersected the genes possessing up-regulated R-loops in *asr20* with the genes having CPT-induced R-loops in *atm* (Supplementary Fig. 2e). A total of 2304 genes containing R-loops were overlapped, which indicates POL2A deficiency triggers topoR-loop dynamics like the effects of TOP1i (Fig. 4c). We further analyzed the associations of various chromatin markers[34,41] with the total, common, and unique R-loops that are up-regulated in *atm* after CPT treatment, as well as in *asr20* compared with *atm* (Fig. 4d). The total and common R-loops were extensively enriched in transcriptional active markers and DNA replication origins, and opposed to H3K27me3 (Fig. 4d). However, the unique R-loops changed in *atm* after CPT treatment were negatively correlated with H3K9me2, in contrast to other types of R-loops, while the unique R-loops up-regulated in *asr20* were strongly associated with H3K9me2 and not correlated with H3K36me3 (Fig. 4d). Overall, the increased R-loops in coding regions co-activated by TOP1i or POL2A deficiency were generally accompanied by active transcription marks, as well as the replication origins, whereas uniquely increased R-loops in *asr20* were highly correlated with repressive markers on heterochromatin.

The above data demonstrated that POL2A, the DNA polymerase, could modulate R-loop homeostasis. The increased R-loops in *asr20* were enriched in DNA replication origins (Fig. 4d), and the correlation rate gradually decreased with a decreasing number of up-regulated R-loops. We wondered whether POL2A could regulate R-loop dynamics by affecting DNA topology through its replication activity, especially during early replication steps. To test this hypothesis, we investigated the R-loop levels centered on DNA replication origins. Compared with Col-0, *pol2a^L473F* showed higher R-loop signals on the origins; these signals were further intensified by CPT treatment, and CPT notably strengthened R-loop increase and enrichment at the origins in *atm* (Fig. 4e). However, R-loops remained stable around the origins, with a low level in *asr20*, regardless of CPT treatment, that was even lower than that in Col-0 (Fig. 4e, g and Supplementary Fig. 8b, c). We next separated the origins into two clusters according to their R-loop levels, and observed that the distance between the origin and their neighboring origins in cluster1 (C1) was shorter than that in cluster2 (C2) (Fig. 4f). These results indicate POL2A might modulate R-loops by controlling DNA replication, probably during the early replication steps.

Accordingly, we treated the plants with a DNA synthesis inhibitor, hydroxyurea (HU), that arrests DNA replication by depleting dNTPs pools[42,43], and found that, *atm*, *pol2a^L473F*, and *asr20* were tolerant to HU (Supplementary Fig. 9a, b). Additionally, a low dosage of HU (0.25 mM) affected the relative relief of TOP1i-induced root

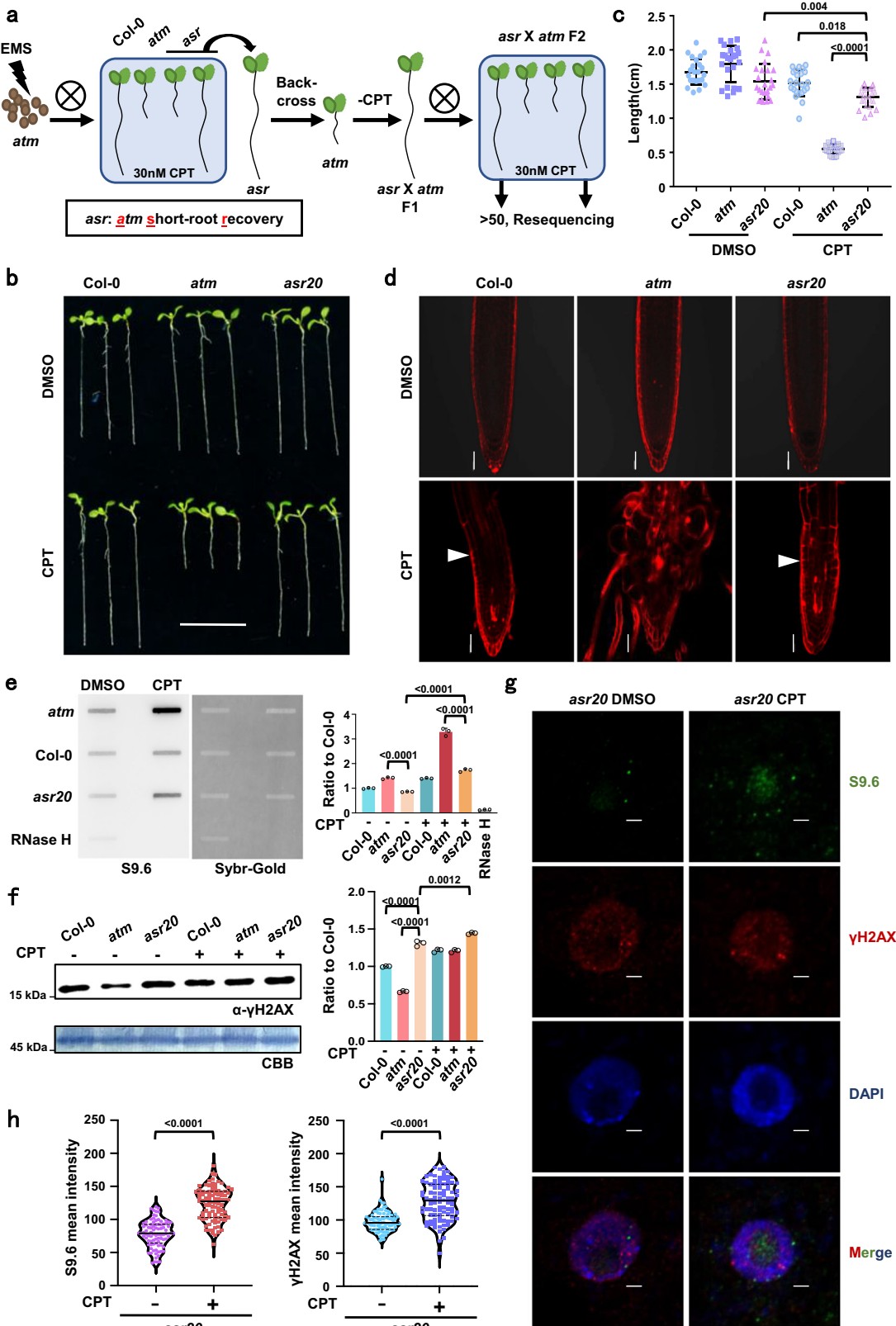

growth defects when the mutants were treated with HU and CPT together (Supplementary Fig. 9a–c). Global R-loop levels detected by slot-blot assay decreased after HU treatment in all tested mutants (Supplementary Fig. 9d). Collectively, these results suggest that DNA replication stress caused by POL2A deficiency or HU is important for topoR-loop regulation.

## *asr20* remodels S-phase progress

Given that POL2A is a major DNA polymerase responsible for leading strand synthesis, and that the allelic mutant *til1-4* (G469R) and *abo4-1* (G522N) show aberrant cell cycle features[44,45], we were interested in analyzing the replication states in plants from different genetic backgrounds by performing EdU (5-ethynyl-2′-deoxyuridine) staining. EdU

**Fig. 3 | *asr20* suppresses *atm* sensitivity to TOP1i. a** Diagram illustrating a forward genetic screen of *atm* short-root suppressors. **b** *ASR20* mutation recovered *atm* hypersensitivity to CPT. Representative root growth phenotypes of 6-day-old seedlings treated with DMSO or 30 nM CPT. Scale bar: 1 cm. **c** The quantitative data of root lengths for Fig. 3b. Lines represent mean ± SD, *n* = 20 (one-way ANOVA). **d** PI staining corresponding to Fig. 3b showed *asr20* restored root meristem structures like Col-0 but had more dead cells. Scale bar: 10 μm. Experiments were repeated three times with similar results. **e** 50 ng genomic DNA of root tips (with and without 2 h of 1 μM CPT treatment) was detected by slot-blot assay using the antibody S9.6. The SYBR-gold staining was used as the loading control. **f** The γH2AX level of root

tips in Col-0, *atm* and *asr20* (with and without 2 h of 1 μM CPT treatment) was detected using Western blot. CBB staining was used as the loading control. The statistical analysis of (**e**, **f**) were shown in Fig. 1c, d. Bars in the plot represent mean ± SD (one-way ANOVA). **g** Immunolabeling of *asr20* root tips (with and without 2 h of 1 μM CPT treatment) using S9.6 and γH2AX antibody. Scale bar: 2 μm. **h** Left, R-loop levels indicated by S9.6 in *asr20* corresponding to Fig. 3g. Right, DNA damage levels indicated by γH2AX in *asr20* corresponding to Fig. 3g. The mean intensity was evaluated using Image J. Lines represent mean ± SD, *n* = 90 (one-way ANOVA). Source data are provided as a Source Data file.

staining could detect DNA proliferation by directly measuring DNA synthesis labeled with thymidine analog EdU[46]. After 2 h of DMSO treatment (T2) and an additional 4 h of recovery (T2 + R4) (Supplementary Fig. 10), EdU-positive (EdU + ) cells of *atm* were similar to that of Col-0, while *asr20* and *pol2a^L473F* had more EdU+ cells than Col-0, but the increasing speed of EdU+ cells in *asr20* and *pol2a^L473F* is slower than in Col-0 and *atm* (Supplementary Fig. 10b, left). This indicated the POL2A mutation stimulated a much faster rate of DNA replication, or much more cells stalled in the S phase for continuous replication. At the T2 time point with CPT treatment, EdU labeling was strongly inhibited by CPT compared with control treatment, except *atm* which was slightly inhibited. However, at 4 h after CPT removal, *atm* and *pol2a^L473F* showed many more EdU+ cells (Supplementary Fig. 10b), suggesting that DNA replication and cell cycle in *atm* is not efficiently inhibited with TOP1i, or that the repairing system is activated to maintain the cell cycle progress in *pol2a^L473F*. Meanwhile, the EdU+ cells in *pol2a^L473F* were more than that in *asr20* after CPT removal (Supplementary Fig. 10b, CPT T2 + R4), indicating *pol2a^L473F* recovers DNA replication process from topology stress faster than *asr20*, and ATM is involved in this progress of *pol2a^L473F* in responding to topology stress.

Results from EdU staining measured the number of total cells entering or undergoing S-phase, but could not evaluate the speed of DNA replication and cell cycle progression. To clarify the DNA replication progression, we used a double-labeling strategy to further evaluate the S-phase progression of root tips (Fig. 5)[47]. The plants were firstly labeled with EdU for 15 min, then chased with thymidine for 0, 1 or 2 h, and finally labeled with a second 15 min pulse of BrdU (Fig. 5a, c, e, top). With the chasing times between the EdU and BrdU pulses increasing, the percentage of the co-labeled (EdU+BrdU + ) and single-labeled (EdU+BrdU-) nuclei will progressively change, depending on the number of cells that enter and finish the S-phase before the second pulse. Thus, measuring the proportion of nuclei with the two or one labels after different chasing times between the two pulses could effectively clarify the S-phase progression and DNA replication progress in different genetic backgrounds (Fig. 5a, c, e). When the two labelings are consecutive (0 h) in the control condition, nearly all the cells are co-labeled in Col-0 and mutants with or without CPT (Fig. 5a, b). At 1 h chase, the percentage of EdU+BrdU+ cells in Col-0 is about 55% and decreases to 45% after 2 h chase, indicating cells enter G2 phase from S-phase as chase time increases (Fig. 5d, f). After 1 h chase, the percentages of EdU+BrdU+ cells in *atm* and *pol2a^L473F* are 32% and 59%, respectively, and more EdU single-labeled cells in *atm*, suggesting more cells exist S-phase in *atm* and more cells stall in S-phase in *pol2a^L473F*, compared to Col-0 (Fig. 5c, d). These results indicate *ATM* mutation may disable the cell cycle checkpoint, and *POL2A* mutation delays S-phase progression. The proportion in *asr20* is similar to *pol2a^L473F*, suggesting that *POL2A* mutation limits S-phase progression in an ATM-independent manner. However, under CPT treatment, there are different proportional patterns in WT and mutants. After 1 h CPT treatment and chase, fewer EdU+BrdU+ nuclei are observed in Col-0 compared with the control condition, and the percentage decreased to 7% after 2 h chase (Fig. 5d, f). This implies CPT could effectively inhibit DNA replication making BrdU hard to be incorporated. Interestingly, the inhibition pattern of single mutant *pol2a^L473F* is similar with Col-0 at

1 h chase timepoint, but the proportion of EdU+BrdU+ nuclei stays stable in *pol2a^L473F* after 2 h chase, indicating *pol2a* could alleviate the DNA replication or S-phase progression inhibition of TOP1i (Fig. 5d, f). Instead, at 1 h chase time with CPT, there is still a large fraction of EdU+BrdU+ nuclei in *atm* and *asr20*, suggesting most S-phase cells of *atm* and *asr20* are still replicating DNA even in CPT stress, while the DNA replication is stalled in Col-0 and *pol2a^L473F* (Fig. 5c, d). At 2 h chase time, the inhibition is still weaker in *atm* and *asr20* as the EdU+BrdU+ nuclei are much more than that in Col-0 but less than 1 h chase condition (Fig. 5e, f). Overall, these results suggested that *pol2a^L473F* and *asr20* delay the DNA replication and S phase progression in normal growth conditions, but alleviate the DNA replication and S phase progression stalling under consistent topological stress.

As both POL2A mutation and TOP1i affect DNA replication, we further carried out immunolabeling of the DSB mark γH2AX after EdU staining (Fig. 6a). Without TOP1i, EdU and γH2AX signals were only co-localized partially in all tested samples, and the γH2AX signals were increased significantly in *asr20*. We then grouped the cells into four types defined according to their EdU and γH2AX signal intensities (Fig. 6b). Under CPT treatment, we noted increasing DNA damage in EdU-positive cells even though very few DNA breaks in non-S phase cells could also be observed (Fig. 6c). Moreover, we detected substantially fewer EdU+γH2AX+ cells in *asr20* compared to either Col-0 or *atm* after CPT treatment (Fig. 6c). These results indicate that TOP1i triggers DSBs restricted in replicating cells. Accumulated R-loops at TTS regions promote TRCs[26,48], up-regulated sense R-loops increased at TTS of coding genes and the locations of the genes were enriched with the DNA origins (Fig. 4, Supplementary Fig. 8). Together with the findings that R-loops on DNA origins decreased in *asr20* and stayed stable after CPT treatment (Fig. 4e, Supplementary Fig. 8b, c), and the fact that genome replication initiates from multiple origins, these results suggest that DSBs could result mainly from the TRCs induced by topoR-loops and the replication stress which could be weakened by POL2A mutation, thus generating less DNA damage.

## The conserved junction domain of POL2A is essential for faithful DNA replication

DNA replication progress was accelerated in *asr20* under CPT treatment compared to Col-0 (Fig. 5), possibly due to the error-prone replication that could mistakenly incorporate abundant ribonucleotides (rNTPs) into genomes during DNA replication without efficient proofreading and mismatch repair[49,50]. In yeast, POL2A mutants rely on the ribonucleotide excision repair (RER) pathway to guarantee genome stability[51,52]. RNase HII is an endonuclease that specifically excises rNTPs in the DNA, and genomic DNA that incorporates rNTPs will be cleaved into short fragments[52]. Through RNase HII digestion, we detected a high level of rNTPs in the *asr20* genome after CPT treatment (Fig. 7a, b), while *atm/abo4-1* showed a similar rNTPs incorporation pattern (Supplementary Fig. 11a, b). Furthermore, we generated an *atrnh2a/pol2a^L473F* and *atrnh2a/abo4-1* double mutant by crossing *atrnh2a*, the defective mutant of Arabidopsis RNase HII[53], with *pol2a^L473F* and *abo4-1*. We found the root growth of *atrnh2a/pol2a^L473F* and *atrnh2a/abo4-1* was inhibited by TOP1i compared to the individual single mutants (Fig. 7c, d, Supplementary Fig. 11c, d). These findings

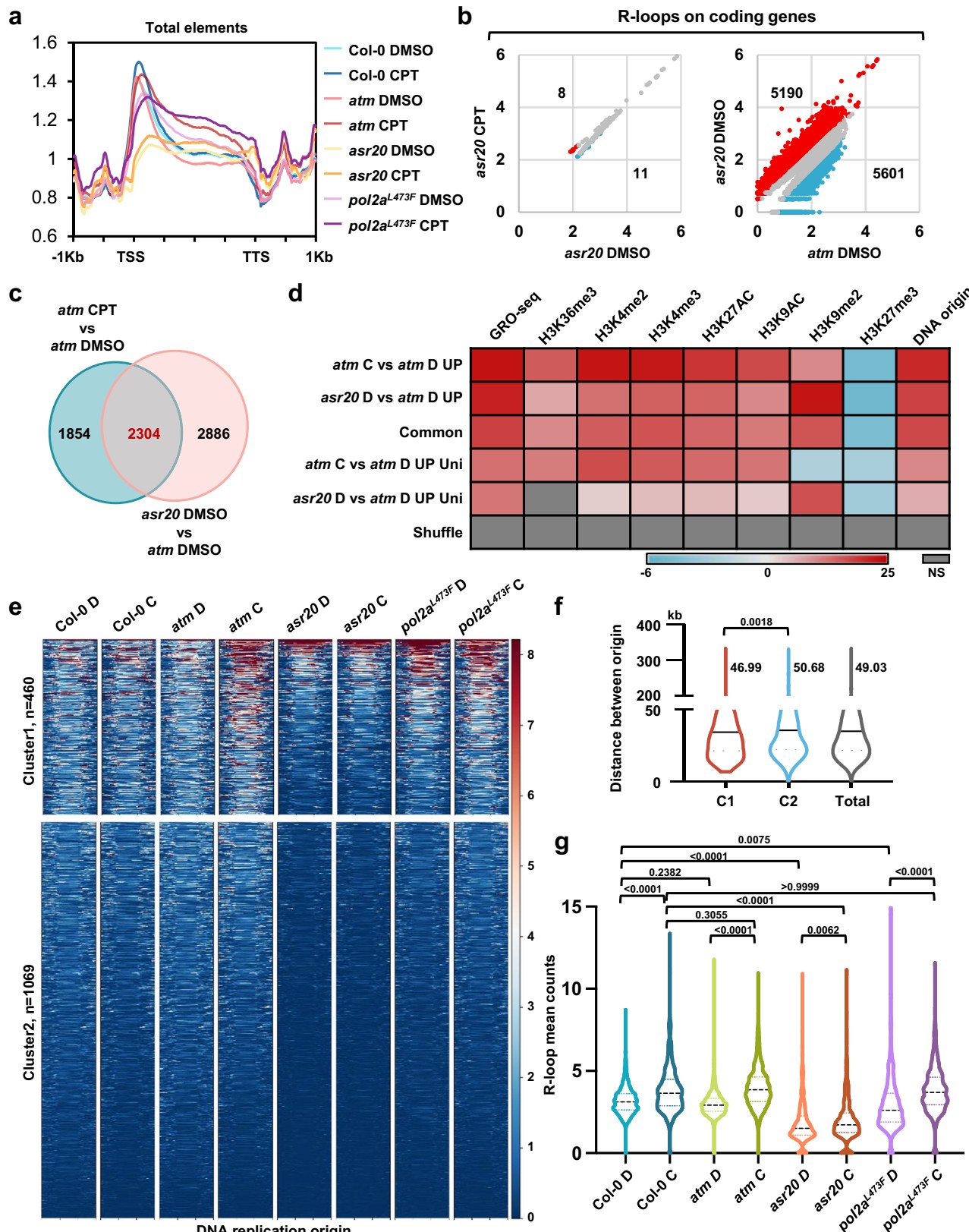

revealed that POL2A mutants fail to proofread and repair on time: The unrestricted rapid DNA replication process may facilitate the mis-incorporation of rNTPs into the genome under consistent topological conditions (Fig. 5 and Fig. 7).

In addition, we applied DNA primer extension assay to investigate the effect of POL2A^L473F mutation on POL ε activity in vitro. We

expressed and purified the catalytic core of WT (POL2A^WT) and mutant (POL2A^L473F) POL2A (1-1123aa) proteins (Supplementary Fig. 12a, b), including polymerase and exonuclease domains which are sufficient for DNA synthesis[54]. The results showed that POL2A^L473F synthesized slightly less DNA than POL2A^WT under the same conditions (Supplementary Fig. 12c), indicating L473F mutation impaired DNA

**Fig. 4 | POL2A affects genomic R-loop dynamics especially on DNA replication origins. a** Metaplot results of unstranded R-loops centered on total annotated genes in Col-0 and mutants with DMSO or CPT treatment. TSS: transcription start site, TTS: transcription termination site. R-loop levels were plotted with the average score of two biological replicates. **b** Scatterplots showing R-loop gene numbers that significantly changed between different samples or treatments. Normalized read counts are shown as $\log_{10}(n+1)$. Red dots: $q$-value < 0.05, $\log_2FC > 1$; blue dots: $q$-value < 0.05, $\log_2FC < -1$; gray dots: other. **c** Venn diagram of overlap between up-regulated R-loop genes in *atm* after CPT treatment in Supplementary Fig. 2e (left), and up-regulated R-loop genes in *asr20* compared to *atm* in Fig. 4b (right). **d** Heatmap showing fold enrichment of total, common, and unique up-regulated R-loop genes in Fig. 4c with different histone marks and DNA replication origins via permutation test. Gene sites were shuffled with bedtools as the control. NS: no significant difference. DMSO or CPT are abbreviated to D or C, respectively, and

unique was indicated as Uni in short. **e** Heatmaps of ssDRIP-seq signals in WT and mutants centered on DNA replication origins. The signals are divided into 2 clusters by k-means clustering, and peak numbers are shown on the left. **f** Violin plots showing the distance between each DNA origin and its nearest DNA origin in cluster1, cluster2, and total origins; 95%confidence interval. The start site of each DNA origin is used to calculate the distance between it and its closest DNA origins. The average distance between the closest DNA origins in cluster 1, cluster 2 and total was 46.99, 50.68 and 49.03 kb, respectively. The median (black solid lines) and interquartile range (dotted lines) are shown. (one-way ANOVA). **g** Violin plots of the mean counts of R-loop peaks plotted on the middle regions between DNA replication origins of cluster 1 and their closest replication origins. 95% confidence interval. Medians (black dotted lines) are shown (one-way ANOVA). Source data are provided as a Source Data file.

polymerase activity, in line with the double-labeling results (Fig. 5). Besides, we performed rNTPs insertion experiments in vitro using POL2A[WT] and POL2A[L473F], and the rGTP incorporation levels are comparable between WT and mutant POL2A (Supplementary Fig. 12d). The results of polymerase activity in vitro demonstrate that POL2A mutation could delay DNA synthesis but does not enhance rNTPs mis-incorporations. Combining the in vivo results (Fig. 7a, b, Supplementary Fig. 11a, b) which showed high levels of rNTPs were detected only in *asr20* after CPT treatment, we conclude that TOP1, ATM and POL2A could cooperatively participate in DNA replication progression and replication fidelity in different conditions.

We further treated the mutants with DNA alkylating agent methyl methanesulfonate (MMS), which could cause DNA mutagenesis and block replication[55]. The mutant *abo4-1* was sensitive to MMS as previously reported[56]; however, *pol2a[L473F]* did not show MMS sensitivity (Supplementary Fig. 13). Root lengths of both *asr20* and *atm/abo4-1* were restrained by MMS, and were obviously shorter than those of *pol2a[L473F]* and *abo4-1* (Supplementary Fig. 13a, b). The number of dead cells stained with PI was consistent with the inhibited root length (Supplementary Fig. 13b, c). Mutations of *pol2a* activated the response to DNA damage and replication fork blockage induced by MMS, and this effect varied with different mutation sites (Supplementary Fig. 6g), especially the *abo4-1* mutation located in the Type-B polymerase domain that is responsible for DNA synthesis, while the *pol2a[L473F]* and *til1-4* mutations are in the conserved junction domain (Supplementary Fig. 6g). These results suggest a previously unknown function of the conserved junction domain in securing the fidelity of DNA replication (mainly in vivo) and also engaging in response to direct DNA damage caused by non-replicative stress, which relies on ATM activity.

### TopoR-loop-induced genome instability is exacerbated by transcription and repaired by both HR and NHEJ pathways

TOP1i favors R-loop accumulation behind the active RNA polymerase, which promotes TRCs and TC-NER and then causes DSB[7,26]. We jointly used the transcription inhibitor flavopiridol (FLV)[18,57] and CPT to treat the relative mutants. We found that FLV treatment obviously alleviated root growth defects caused by TOP1i (Fig. 8a, b). Slot-blot results showed that FLV could effectively inhibit R-loops accumulated by CPT treatment (Fig. 8c). These results imply that the topoR-loop accumulation and DNA damage rely on transcription activity.

DSBs are cytotoxic lesions that cause chromosome recombination or cell death, and HR and NHEJ are the major repair pathways for DSBs. HR is a high-fidelity but template-dependent repair pathway that is restricted in S phase[58], while NHEJ functions on a wide range of DNA-end configurations in a cell cycle-independent manner. ATM participates in both the HR and NHEJ pathways[59]. To determine which pathway(s) is primarily involved in R-loop-induced DNA damage repair, we crossed *atm* with *rad52*, a defective mutant of Rad52 that is required in HR, and *ku70*, a mutant of DSB end-binding protein Ku70 in NHEJ. We found that the *atm/ku70* double mutant showed a short root

phenotype, which suggested most DSB repair in normal cell division relies greatly on NHEJ (Fig. 8e, f). However, *rad52* and *ku70* presented insensitivity to CPT, and the roots of both *atm/rad52* and *atm/ku70* were slightly inhibited by CPT, which is different from the *atm* single mutant (Fig. 8e, f, Supplementary Fig. 13d). Conclusively, the NHEJ pathway ensures normal root growth and development regardless the cell cycle phase; and TOP1i-increased topoR-loops are probably persistent and cause DNA damage throughout the cell cycle, with HR and NHEJ involved to repair the breaks.

## Discussion

TOP1i could produce single-ended DSBs (seDSBs) and R-loops[3,22,23,60]. TOP1 inhibitors could be used to kill cancer cells, and are most effective in HR-defective cells[61,62]. Here we showed that TOP1i induces cell death and inhibits root growth in Arabidopsis. Based on this, we established a screening system and identified both DSB repair kinase ATM and DNA polymerase ε catalytic subunit POL2A play a vital role in regulating topology-associated R-loops and genome stability (Fig. 9). Notably, the severely aberrant root growth of *atm* mutant treated with CPT resulted from increased topoR-loop accumulation and DNA breaks (Figs. 1, 2, Supplementary Fig. 2). Interestingly, TOP1i-induced topoR-loops enriched at TTS of protein coding genes and were highly correlated with DNA replication origins in *atm* (Fig. 4c–e). Previous study showed that R-loops in TTS are prone to transcription and replication collisions in TOP1 defect cells[26]. Collectively, the results suggest TOP1i accumulated topoR-loops could primarily trigger cascade events leading to TRCs-induced DSBs and activating ATM repair pathways (Fig. 9).

POL2A is relatively conserved in eukaryotes (Supplementary Fig. 6g), and it is known that its DNA polymerase domain and exonuclease domain are crucial for DNA replication and nucleotide excision repair (NER)[28,50]. In Arabidopsis, POL2A plays an important role in cell cycle regulation, chromosome recombination, transcriptional gene silencing (TGS), and epigenetic regulation[45,56,63–67]. Here we identified *pol2a* is the *atm* suppressor under topological stress (TOP1i), rescuing the short and abnormal root growth of *atm* under CPT treatment (Fig. 3, Supplementary Fig. 6). POL2A mutation may trigger ATR-dependent signaling and WEE1-dependent cell cycle inhibition for which *abo4-1/atr* and *abo4-1/wee1* are embryonic lethal[45]. We showed that *pol2a[L473F]* is resistant to CPT (Supplementary Fig. 7), while *asr20* showed higher γH2AX levels than Col-0 (Fig. 3f–h), suggesting *pol2a[L473F]* could activate ATR-dependent signaling and WEE1-dependent cell cycle regulation. Additionally, ribonucleotide incorporation during DNA replication is removed by RNase H2-dependent ribonucleotide excision repair (RER)[52,53]. In RER-defective yeast, Top1 specifically incises ribonucleotides in the nascent leading strand[68], which is synthesized by DNA polymerase ε. Only *asr20* treated with TOP1i accumulated a very high level of ribonucleotides (Fig. 7a, b, Supplementary Fig. 11a, b), *pol2a[L473F]*-caused ribonucleotides incorporation is TOP1-dependent, and further lead to DNA breaks, and

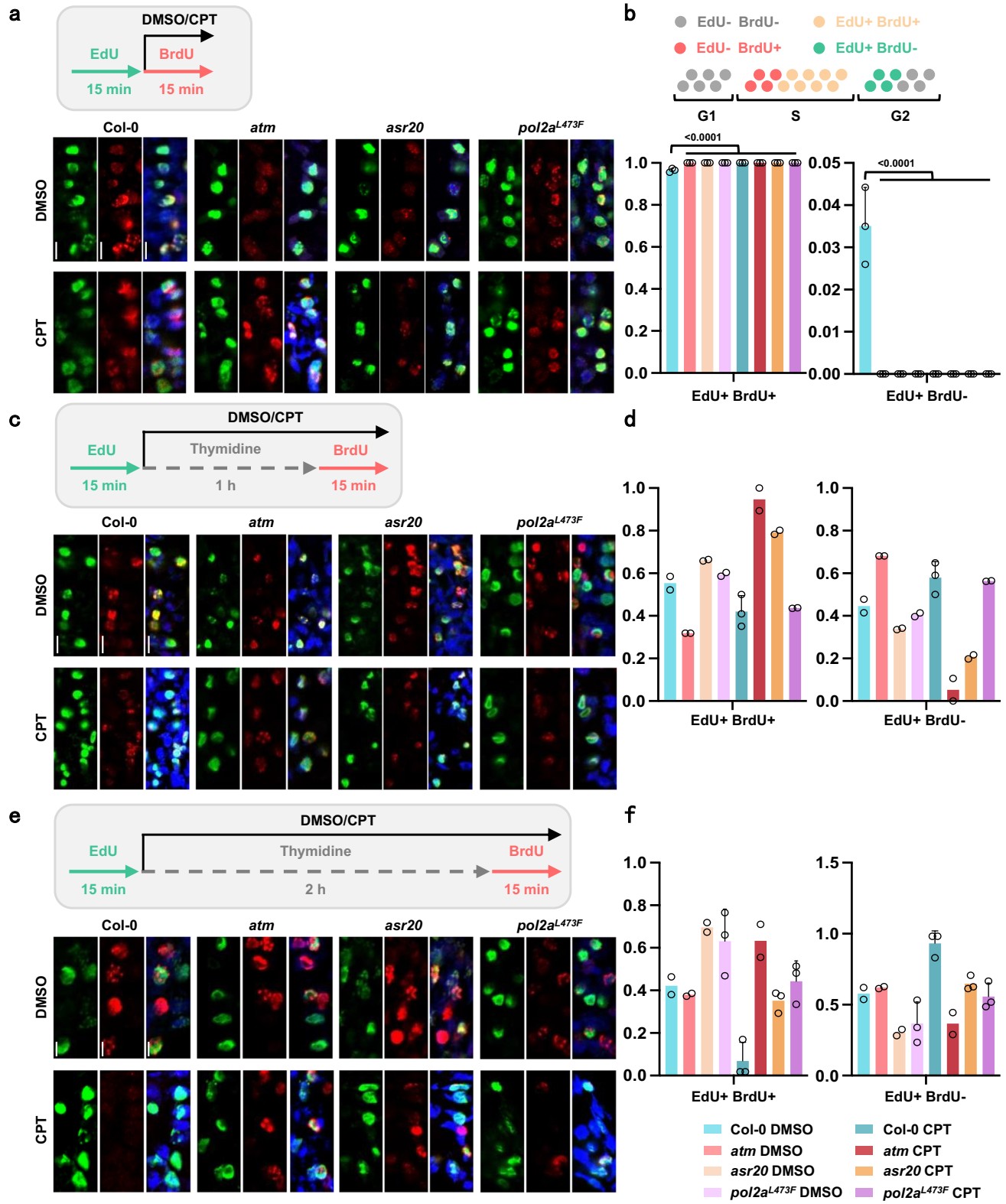

this relies on ATM for repair (Fig. 9). In addition, combining with genetic results that *atrnh2a* abolished the resistance of *pol2a* to TOP1i (Fig. 7c, d, Supplementary Fig. 7 and 11c, d), and R-loops near DNA origins decreased in *asr20* (Fig. 4f), we speculate that AtRNH2A could remove R-loops near DNA replication regions, especially on nascent DNA containing high ribonucleotides (Fig. 9).

DNA replication stalling could further avoid TRC, which may be the reason for the significant reduction of DSB signals in replicating

cells after CPT treatment (Fig. 6). R-loops are increased at replication origins in *pol2a^{L473F}* mutants in control condition (Fig. 4f), possibly due to DNA replication stalling that hampers the resolution of nearby R-loops, and this effect is more visible in *atm* background (Fig. 5 and Fig. 9). In topological stress with TOP1i, POL2A deficiency extricates DNA replication inhibition and decreases R-loop accumulation near replication origins (Fig. 4d, f, and Fig. 5), which is in agreement with the observation that increased TOP1 expression causing low R-loop levels

**Fig. 5 | POL2A mutation normally delays S-phase progression but promotes it in topology stress. a** S-phase progression of root tips measured by double-labeling strategy in 4-day-old Col-0, *atm, asr20, pol2a^{L473F}*. 4-day-old seedlings were firstly pulsed with 10 μM EdU for 15 min, a second 50 μM BrdU pulse for 15 min with DMSO or 1 μM CPT, as the schematic workflow shown in the upper left. A diagram of cell types labeled by EdU or BrdU in the cell cycle was shown in the upper right. **b** Percentage of double-labeled (EdU and BrdU), single-labeled (EdU) nuclei related to Fig. 5a, >300 nuclei from 3 roots were scored for each sample (one-way ANOVA). **c** S-phase progression of root tips measured by double-labeling strategy in 4-day-old seedlings. After firstly pulsing with 10 μM EdU for 15 min, a 1 h 20 μM thymidine pulse with DMSO or 1 μM CPT, and then a second 50 μM BrdU pulse for 15 min with DMSO or 1 μM CPT, as the schematic workflow shown above. **d** Percentage of double-labeled (EdU and BrdU), single-labeled (EdU) nuclei related to Fig. 5c, >300 nuclei from 2 or 3 roots were scored for each sample. **e** S-phase progression of root tips measured by double-labeling strategy in 4-day-old seedlings. After firstly pulsing with 10 μM EdU for 15 min, a 2 h 20 μM thymidine pulse with DMSO or 1 μM CPT, and then a second 50 μM BrdU pulse for 15 min with DMSO or 1 μM CPT treatment, as the schematic workflow shown above. **f** Percentage of double-labeled (EdU and BrdU), single-labeled (EdU) nuclei related to Fig. 5e, >300 nuclei from 2 or 3 roots were scored for each sample. Source data are provided as a Source Data file.

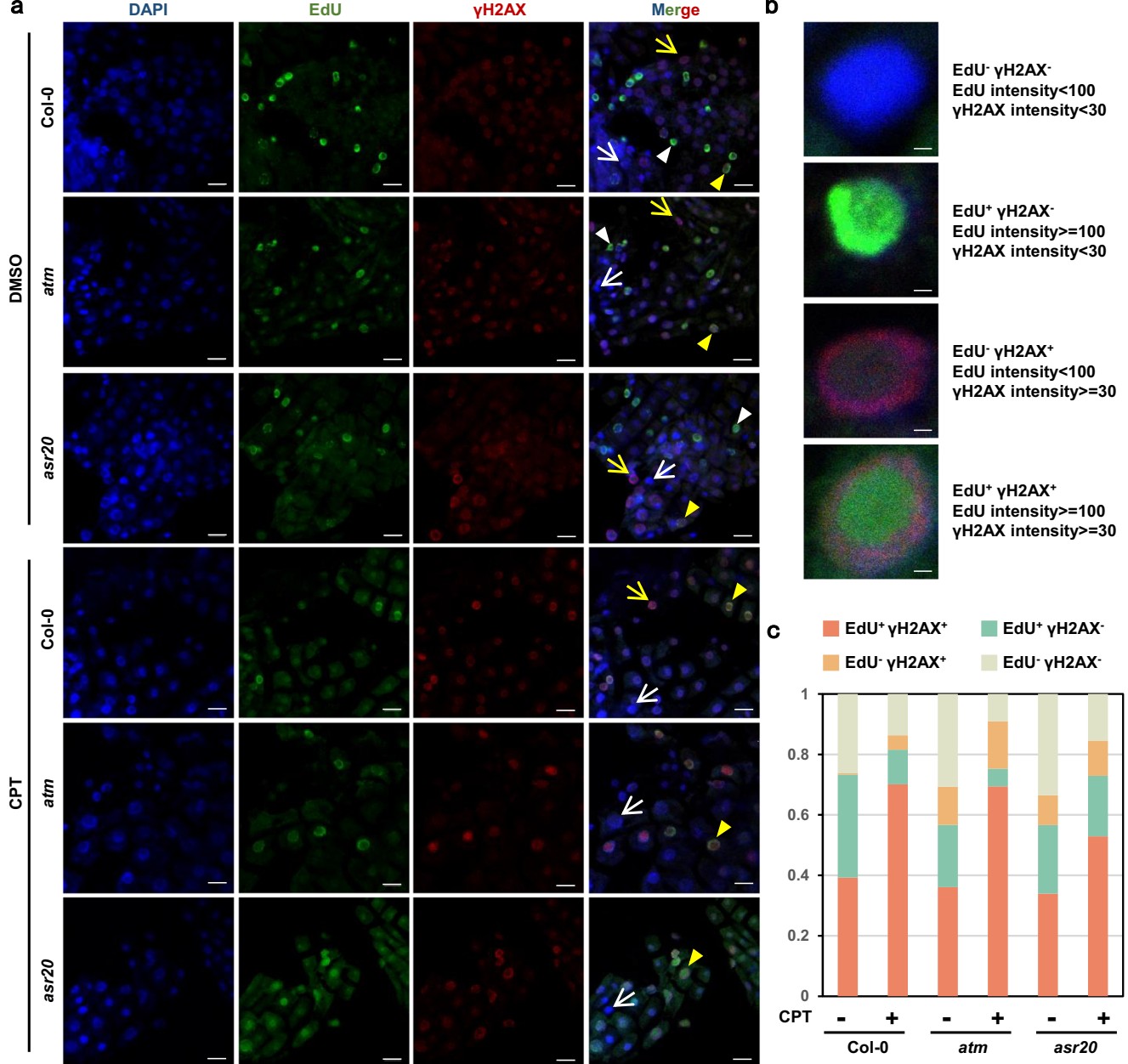

**Fig. 6 | Effects of ATM and POL2A in DNA replication and DNA damage. a** Immunolabeling of root tips nuclei from 5-day-old Col-0, *atm*, and *asr20* treated with DMSO or 1 μM CPT for 2 h using EdU and γH2AX antibody. Yellow triangle: EdU +γH2AX+ nuclei; white triangle: EdU+γH2AX- nuclei; yellow arrow: EdU-γH2AX+ nuclei; white arrow: EdU-γH2AX- nuclei. Scale bar: 10 μm. Experiments were repeated three times with similar results. **b** Strategies of cell type classification according to EdU and γH2AX intensity. **c** Percentage of DNA damage occurring during replication corresponding to Fig. 6a. More than 150 nuclei were scored for each sample.

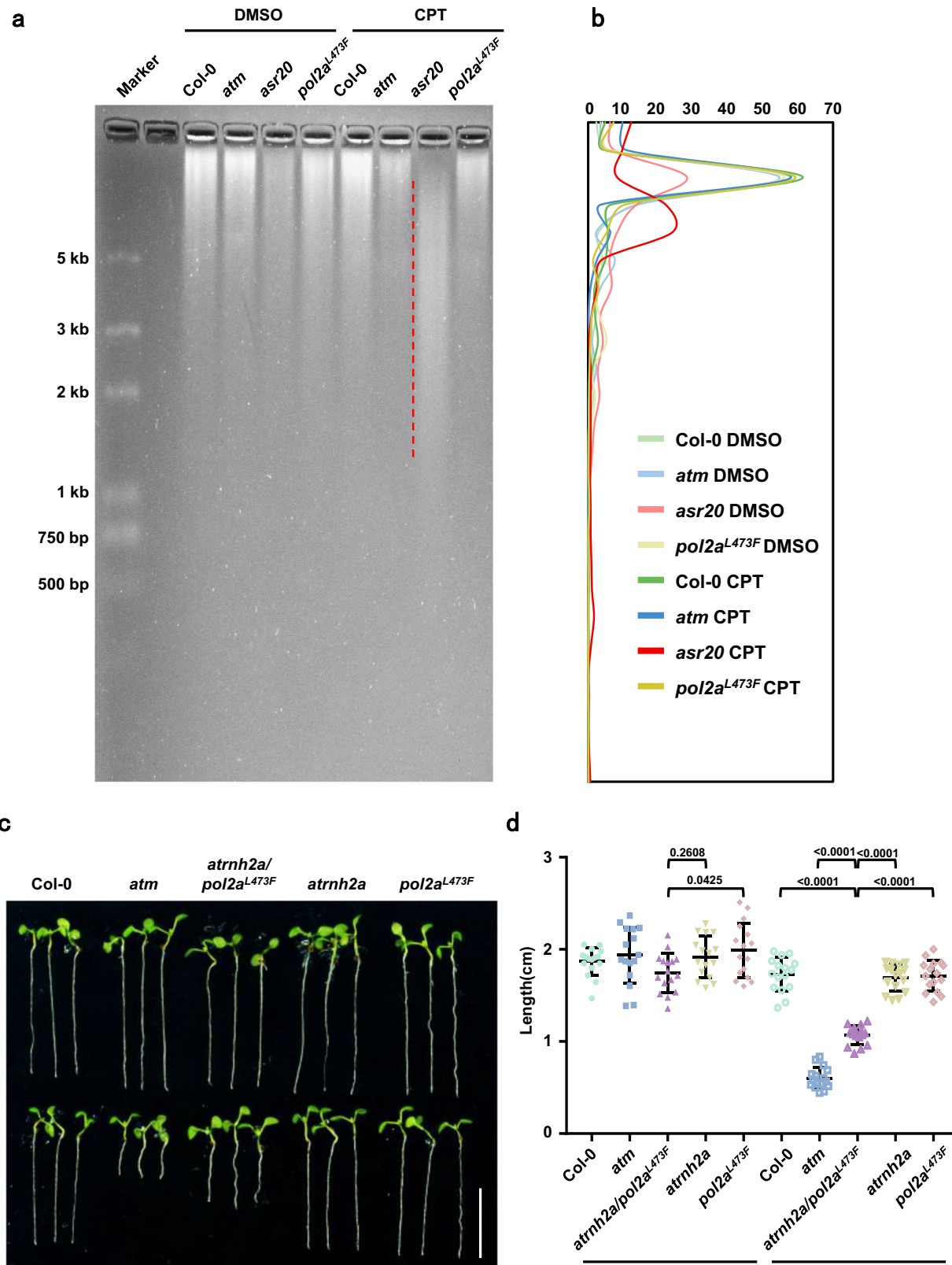

**Fig. 7 | POL2A and ATM guard the fidelity of DNA replication. a** Ribonucleotide incorporation detection with RNase HII cleavage products of genomic DNA extracted from Col-0, *atm*, *asr20*, and *pol2a$^{L473F}$* treated with DMSO or 30 nM CPT for 7 days. The rNTPs incorporation ratio was elevated in the *asr20* genome after CPT treatment as 2–5 kb DNA fragments (dashed red line) were predominantly detected. Experiments were repeated three times with similar results. **b** Densitometry plot of lanes corresponding to Fig. 7a. **c** Root phenotypes of 7-day-old Col-0, *atm*, *atrnh2a/pol2a$^{L473F}$*, *atrnh2a*, and *pol2a$^{L473F}$* treated with DMSO or 30 nM CPT. Scale bar: 1 cm. **d** The quantitative data of root lengths for Fig. 7. Lines represent mean ± SD, *n* = 16 (one-way ANOVA). Source data are provided as a Source Data file.

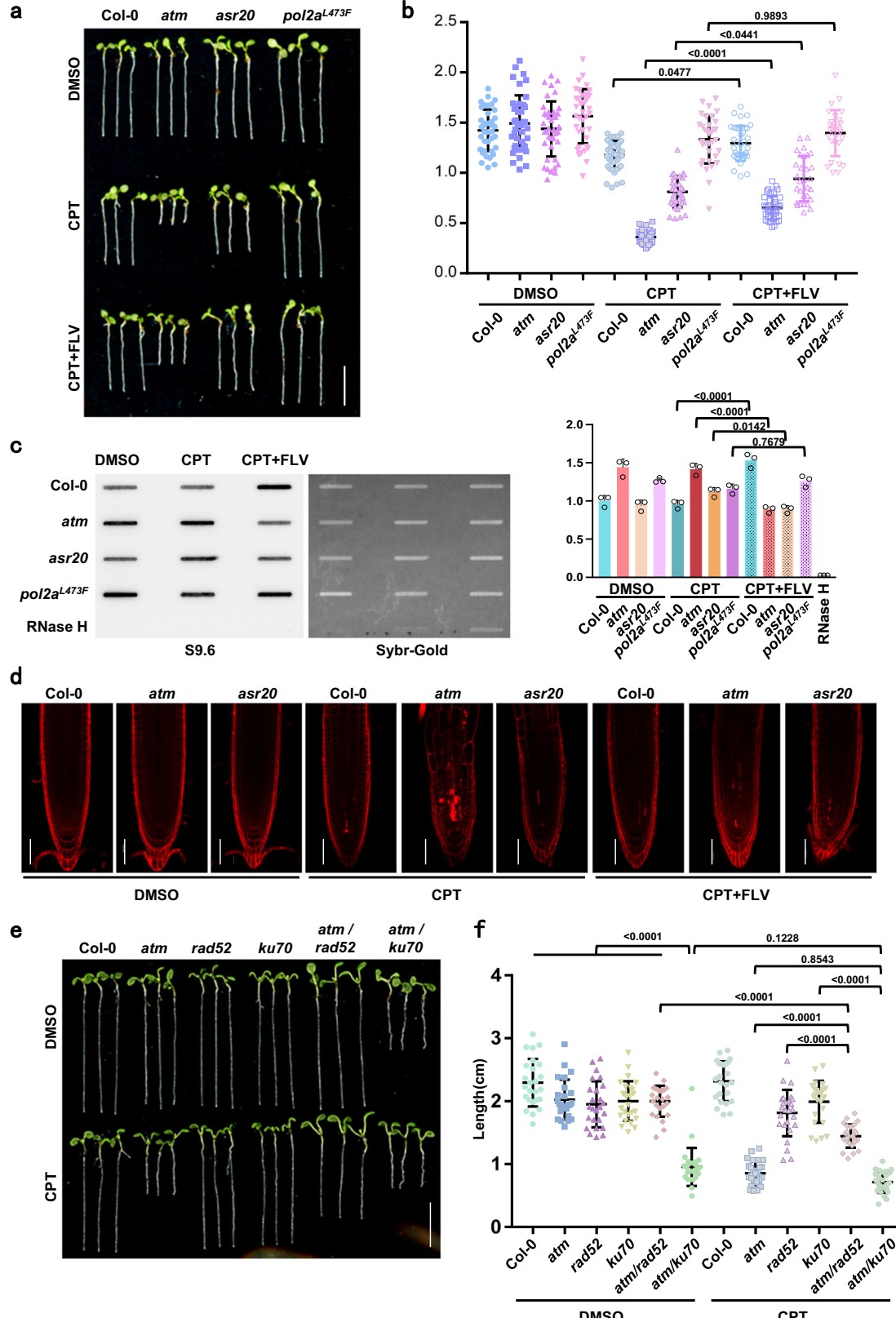

**Fig. 8 | Transcription and DSB repair factors function in R-loop dynamics.**
**a** Root length phenotypes of 5-day-old seedlings treated with DMSO, 30 nM CPT or 30 nM CPT plus 1 μM transcription inhibitor flavopiridol (FLV). Scale bar: 1 cm.
**b** The quantitative data of root lengths for Fig. 8a. Lines represent mean ± SD, $n = 20$ (one-way ANOVA). **c** Slot-blot assay showing the R-loop levels from 50 ng genomic DNA of plants treated with CPT or CPT plus FLV. Band signals of each sample are calculated three times with Image J. Bars in the plot represent mean ± SD (one-way ANOVA). **d** PI staining showing the root meristem phenotype of plants treated with CPT or CPT plus FLV. Scale bar: 50 μm. **e** Root growth phenotypes of 8-day-old seedlings treated with DMSO or 30 nM CPT. Scale bar: 1 cm. **f** The quantitative data of root lengths for Fig. 8e. Lines represent mean ± SD, $n = 20$ (one-way ANOVA). Source data are provided as a Source Data file.

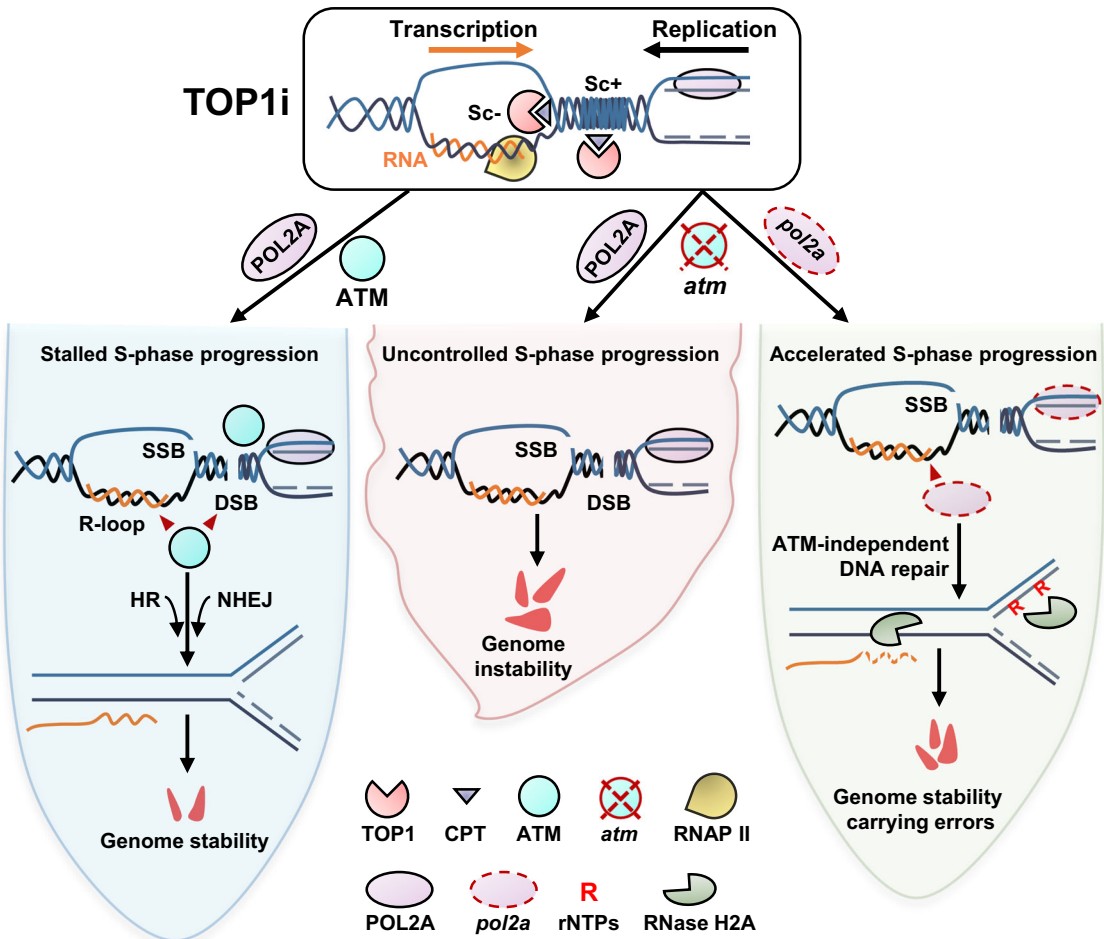

**Fig. 9 | Working model of ATM and DNA polymerase ε functions in topoR-loop-triggered DNA damage.** TOP1i induces DNA topology stress in the transcription and replication active regions, and TOP1 cleavage complex promotes topoR-loop accumulation at the negative supercoils behind the transcription bubbles, which triggers TRCs, resulting in DSB. In Col-0, S-phase progression is stalled, R-loops accumulation is controlled and the DNA damage repair pathway is activated under topological stress, HR and NHEJ repair pathways participate in repairing DSB in time to maintain genome stability with only little cell death. In *atm*, uncontrolled S-phase progression, largely accumulated R-loops and unrepaired DNA damages severely endanger genome stability and cause plenty of cell death and eventually destroy root development. Further POL2A mutation in *atm* rescues the genome instability by regulating the DNA replication and S-phase progress which may trigger ATM-independent repair pathways, and limit R-loop accumulations near DNA replication origins with rNTPs misincorporation, and thus activate AtRNH2-dependent RER. As a result, the root growth returns to normal (compared to *atm*).

and promoting accelerated replication[69]. We propose that *pol2a* might promote S-phase progression, therefore, to decrease R-loops under topological stress (Fig. 9). The *pol2a* mutations that we used are located in the conserved and uncharacterized junction domain (Supplementary Fig. 6g). There may be other factors that modulate DNA replication speed and fidelity of *pol2a* and *asr20* under topology stress through interacting with POL2A, especially in this junction domain, as it is known that POL2A-interacted proteins are involved in modulating its biological functions. For instance, it is reported that the C-terminal portion of POL2A could interact with POLE2, which is essential for cell proliferation[70,71]. Moreover, the POL2A C-terminal zinc finger domain specifically binds to histone H3.1-H4 dimer or tetramer, which is important for meiotic heterochromatin condensation[66]. Also, POL2A could maintain H3K27me3 modification and gene silencing through its C-terminal interaction with PRC2 complex[63]. Recent results unveiled that the N-terminal of POL2A interacts with SU(VAR)3-9 homologs SUVH2 and is recruited to meiotic DSB sites, thereby repressing the expression of DSB-associated genes[72]. Besides, CHG methylation and H3K9me2 are significantly increased in *pol2a* or under HU treatment in the centromere region, and POL2A defect is associated with fragmentation of heterochromatin[65]. In addition, CPT treatment or TOP1 inhibition could release TE silencing through impacting DNA

methylation and H3K9me2[73]. We also found increased R-loops in *asr20* enriched with H3K9me2 (Fig. 4c, d). Altogether, POL2A may act as the core player bridging R-loops, DNA replication and epigenetic modifications through interacting and recruiting replication, transcription regulators and R-loop resolving factors.

For DSB repair, NHEJ is favored as it is not limited by cell cycle[74]. In contrast, seDSBs generated for example by TOP1cc, lack a second DNA end for end-joining[3], and thus are repaired through HR in the S/G2 phase of the cell cycle when homologous templates presenting on the sister chromatid[58,75]. The root growth of *atm/ku70* double mutants is shorter than that of *atm* and *ku70* in normal growth conditions, and *atm/ku70* and *atm/rad52* are only slightly sensitive to CPT (Fig. 8e, f). Considering γH2AX levels are elevated in replicating cells (Fig. 6), we are convinced that NHEJ is preferred for DNA damage repair in normal plant development, and TOP1i triggered both double- and single-ended breaks that are repaired collaboratively by NHEJ and HR (Fig. 9).

Patients with ataxia telangiectasia (A-T) caused by either ATM deficiency or mutation are prone to various cancers[76]. As a tumor suppressor gene, *ATM* mutations are hypersensitive to DNA damage from chemotherapeutic drugs like TOP1i and poly (ADP-ribose) polymerase inhibitors[62,76,77]. Tumor cell resistance to molecularly targeted drugs is a challenging problem in cancer research and therapy. Hence,

understanding the mechanisms of chemotherapeutic resistance would provide new therapeutic strategies for the combination of DNA-damaging agents and better patient stratification[78]. Our results indicate that the conserved junction domain between exonuclease and polymerase domains of DNA Pol ε is essential for genome integrity, and the weak mutation of which could lead to *atm* resistance to TOP1i. Given the crucial and conserved roles of DNA Pol ε in genome duplication and tumor suppression[79], our study revealed that this highly conserved domain in POL2A among different species highlights the potential mechanism countering TOP1i or other DNA-damaging agents in *atm* or even HR-defective tumors, which could promote understanding of drug tolerance and cancer therapy.

## Methods

### Plant materials and growth conditions

The T-DNA insertion mutants *atm* (SALK_089805), *atr* (SALK_032841), *rad52* (SAIL_25_H08), *ku70* (SALK_123114), *atrnh2a* (GABI-139H04) were purchased from NASC (The Nottingham Arabidopsis Stock Centre). The *abo4-1* was a gift from Professor Zhizhong Gong in CAU[56]. The other mutants were generated in this study. Primers used for genotyping are listed in Supplementary Data 1. The sterilized Arabidopsis seeds were sown on 1/2 Murashige and Skoog (MS) medium with 1% sucrose and 0.8% phytagel for root observation and drug treatment, and seeds were stratified at 4 °C in the dark for 2 days then grown vertically under long-day conditions (16 h light/8 h dark) at 22 °C in a growth chamber.

### Treatment

For CPT sensitivity assays, seeds were grown on 1/2 MS supplemented with DMSO or 30 nM CPT for 5–7 days. The root length was measured with ImageJ software. For CPT sensitivity assays of estrogen-inducible AtRNH1A overexpression materials, plants were germinated on MS medium for 3 days, then transferred to MS medium containing 50 μM β-estradiol to induce AtRNH1A overexpression for 36 h. After induction, the plants were grown on MS supplemented with DMSO, 30 nM CPT or 30 nM CPT and 50 μM β-estradiol for 10 days. For FLV and HU sensitivity assays, seeds were grown on 1/2 MS containing 30 nM CPT and 1 μM FLV or 0.25 mM HU for 5–7 days.

5-day-old seedlings grown vertically were used for short time treatment. Plants were transferred to liquid 1/2 MS medium containing DMSO, 1 μM CPT, 10 μM FLV or 10 mM HU for 2 h at 22 °C in a growth chamber, then quickly washed with 1X PBS twice. The root tips were cut for further experiments including slot blot, western blot, immunolabeling and ssDRIP-seq and qPCR. For immunolabeling of estrogen-inducible AtRNH1A overexpression materials, 3-day-old seedlings were transferred to 1/2 MS supplemented with 100 μM β-estradiol to induce AtRNH1A overexpression for 36 h, and transferred to 1/2 MS supplemented with 100 μM β-estradiol and DMSO or 1 μM CPT for 2 h.

### Cloning procedures

The cDNA of *POL2A* (AT1G08260, 1-3369 bp) and *AtRNH1A* (AT3G01410, full length) were respectively cloned into the pGEX-4T1 vector fused with glutathione S-transferase (GST) at 5′-end through FastCloning[80]. The point mutation of *POL2A*[L473F] and *AtRNH1A*[D160N] was generated by inverse nested PCR. The two plasmids were transformed and expressed in BL21 (DE3) cells.

The coding sequence of *AtRNH1A* was cloned to the pER8 vector[81] with 3X FLAG through FastCloning and the *pER8::AtRNH1A*[D160N]*–3XFLAG* construct was generated by inverse nested PCR. The two plasmids were transformed into the materials by the floral dip method[82].

### Staining and microscopy

For PI staining, roots were mounted in 100 μg/ml propidium iodide solution for 1 min and visualized using a confocal laser scanning microscope (LSM 780, ZEISS) after dip washing in 1X PBS.

For EdU staining, 5-day-old seedlings were transferred into 1/2 MS liquid medium supplemented with 1 μM DMSO or CPT for 1.5 h, then added 10 μM EdU (5-ethynyl-2-deoxyuridine) and incubated for 30 min. Half plants were fixed in 3.7% formaldehyde for 2 min, then washed 3 times with 1X PBS and further processed using the kit Click-iT EdU Alexa Fluor 488 for 45 min. The other half was transferred to 1/2 MS liquid medium containing 10 μM EdU for 4 h after dip washing in 1X PBS 3 times. Plants were fixed and labeled as described before. All the incubations were performed in liquid 1/2 MS at 22 °C in a growth chamber. The seedlings could be stored at 4 °C in the dark or visualized immediately using a confocal laser scanning microscope (LSM 780, ZEISS).

The nuclei immunolabeling of root tips was performed according to reference[83]. Briefly, the seedlings were fixed in 3.7% formaldehyde for 2 min in 1X PBS and then washed 3 times with PBS. They were digested at 37 °C for 5 min with cell wall lysis buffer (3% (w/v) cellulase and 0.5% (w/v) macerozyme). The root tips were cut and gently squashed onto poly-L-lysine coated slides; the slides were put into liquid nitrogen for 30 s then the cover slips were removed. After blocking with 5% BSA in PBS, the slides were incubated with primary antibody γ-H2AX (Abclonal: AP1267, diluted 1:200) or S9.6 (produced with HB-8730 (ATCC) cell line, diluted 1:200) at 4 °C overnight, and the secondary antibody with Alexa Fluor (Goat anti-Rabbit, Alexa Fluor™ 555: A27039, Goat anti-Mouse, Alexa Fluor™ 488: A28175) diluted 1:500 for 1 h at room temperature. Finally, slides were mounted in DAPI solution. The slides could be either stored at 4 °C in the dark or visualized immediately using a confocal laser scanning microscope (LSM 880, ZEISS).

### S-phase progression assessment

The S-phase progression assessment of root tips was performed referred to[47,84] with some modifications. 4-day-old seedlings were used for the analysis of S-phase progression. The plants were firstly labeled with 10 μM EdU for 15 min, then washed off the EdU and chased with 25 μM thymidine for increasing times (0, 1, 2 h), and finally, after washing off thymidine, 50 μM BrdU (5-bromo-2-deoxyuridine) was used for second labeling for 15 min. For the S-phase progression assessment under CPT stress, 1 μM CPT was added and incubated with thymidine and BrdU simultaneously. All the incubations were performed in liquid 1/2 MS at 22 °C in a growth chamber. Roots were then processed as described in the EdU staining protocol. After Click-iT EdU reactions with the kit Click-iT EdU Alexa Fluor 488 for 1 h, the roots were placed on the Superfrost plus slides and fixed to the slides through air-drying for >12 h and protected from light. Root cell walls were digested with Driselase (20 mg/ml, Sigma: D8037) for 45 min at 37 °C, and then slides were washed with 1X PBS. Samples were then permeabilized with 10% DMSO and 3% Igepal CA-630 in 1X PBS for 1 h at room temperature. After washing, the DNA was denatured with 2.5 M HCl for 75 min at room temperature. Samples were blocked with 5% BSA in PBS for 1 h at room temperature, then incubated with anti-BrdU antibody diluted 1:100 (Invitrogen: B35128) overnight at 4 °C. The samples were incubated with secondary antibody diluted 1:500 for 1 h at room temperature (Goat anti-Mouse, Alexa Fluor™ 555: A28180). Finally, slides were mounted in DAPI solution. The slides could be either stored at 4 °C in the dark or visualized immediately using a confocal laser scanning microscope (LSM 880, ZEISS).

### Protein extraction and western blot

Root tips were ground into powder in liquid nitrogen for the total protein extraction. For γH2AX detection, 1X SDS buffer (10 mM Tris, pH 7.2, 2% SDS, 10% glycerol, 0.1% bromophenol blue, and 1% β-mercaptoethanol) was added to the powder and boiled at 95 °C for 5 min. After centrifugation of 12000 x g, the supernatant was used for western blot. After blocking, the membranes were incubated with primary antibody γ-H2AX (AP1267, Abclonal, 1:3000 dilution) at 4 °C overnight

and the secondary antibody diluted 1:5000 for 1 h at room temperature. The Chemiluminescence Lumilight reagent was used for the final images. For *AtRNH1A* overexpression analysis, 200 µL protein extraction buffer (50 mM Tris-HCl pH 7.5, 150 mM NaCl, 10% glycerin, 0.5% Triton X-100, 0.5% NP-40, 10 mM DTT, 1 mM PMSF, 1X cocktail) was added for 100 mg powder, lysed on ice for 10 min with gentle vortex every 3 min. 5X SDS loading buffer (Yeasen: 20315ES05) was added to the samples and boiled at 95 °C for 5 min. The other steps are performed as described above except for the use of a primary antibody (Anti-FLAG: F1804, Sigma-Aldrich, 1:5000 dilution).

## ssDRIP sequencing

Seedlings grown on 1/2 MS for 5 days were transferred to MS liquid medium containing 1 µM DMSO or CPT for 2-h treatment, and genomic DNA was extracted from roots tips using Honda buffer without filtering as described in[34,35]. Then other steps followed the ssDRIP-seq protocol according to references[34,35]. In brief, 1 µg fragmented DNA (digested by DdeI (R0175S, NEB), MseI (R0525S, NEB), NlaIII (R0125S, NEB) and MboI (R0147S, NEB) endonucleases) was used for S9.6 immunoprecipitation. Meanwhile, the same amount of RNase H-treated fragmented DNA was used for S9.6 immunoprecipitation as the negative control. For each immunoprecipitation assay, 8 µg S9.6 antibody and 40 µL Protein G beads were used. The precipitated DNA was used for sequencing library constructions using an ssDNA library preparation kit (ND620, Vazyme).

## DRIP-qPCR

Quantitative real-time PCR of DRIP was performed using a LightCycler 480 (Roche, Basel, Switzerland). The reaction mixture (11 µl) contained 0.5 µL DNA, 5.5 µL of 480 SYBR Green I Master (Roche), and 2.5 pmol forward and reverse primer listed in Supplementary Data 1. The input was used for normalization.

## Slot blot

The nucleic DNA of root tips was extracted as ssDRIP assay, the genomic DNA was firstly digested with RNase III (M0245S, NEB) at 37 °C for 2 h, then purified with HiPure Gel Pure DNA Mini Kit (D2111-03, Magen). After quantification with Qubit, the DNA as indicated in the figure was slotted onto a nylon membrane (Amersham Hybond-N +) and detected by S9.6 antibody following[85].

## Electrophoretic mobility-shift assay (EMSA)

The sequences of RNA:DNA hybrids used in this study were the same as described in[35] (DNA: 5′- CACATGTCGGTATACCTACCGGGTC AACGTAGTGTTA −3′; RNA: 5′-FAM- UAACACUACGUUGACCCGGU AGGUAUACCGACAUGUG −3′). The 1 µM RNA:DNA hybrids were incubated with the increasing amount of proteins in digestion buffer (15 mM Tris-HCl pH 7.5, 50 mM KCl, 1 mM DTT, 0.1 mM EDTA, 10 mM MgCl$_2$, 0.1 mg/mL BSA and 5% glycerol) for 5 min at room temperature, and then loaded onto 5% PAGE gels, the mixtures were separated in 0.5x TBE buffer at 10 V/cm for 20 min. The gel was visualized by fluorescence scanning with the Typhoon FLA 9500.

## Alkaline agarose gel electrophoresis

Nucleic DNA from root tips was extracted using Honda buffer as described in[34,35] with minor modifications. During the nucleic lysis step, RNase A was added to avoid RNA contamination. Then the DNA was precipitated with 2.5 volumes of 100% ethanol and 0.1 volumes of 3 M sodium acetate pH 5.2. 2 µg DNA was further treated with RNase HII at 37 °C for 2 h and then was ethanol precipitated, finally dissolved in 24 µL loading buffer (90% formamide, 20 mM EDTA, 0.5% xylene cyanol FF). The samples were loaded into 1% alkaline agarose gel (Melt 1 g agarose in 95 mL H$_2$O, cool to 60 °C. Add 5 mL 1 M NaOH and 0.2 mL 0.5 M EDTA, pH 8.0, mix and pour) and electrophorese at 30 V for 30 min, then run at 10 V for 18-20 h at room temperature.

Neutralize the gel in Neutralization buffer (1 M Tris−HCl, 1.5 M NaCl) for 45 min at room temperature with agitation for two times. Immerse the gel in H$_2$O and stain using 0.5 µg/mL SYBR Gold (Invitrogen) for 45 min at room temperature with agitation. Visualize DNA using a UV trans-illuminator.

## Template-directed DNA synthesis

100 µM DNA template (Temp1: 5′- GGGGGGGGGGGGGGGGGGGGGG GGGGGGATT −3′ and Temp2: 5′- GGGGGGGGGGGGGGGGGGGGGG GGGGGGGTCAG −3′) and its primer (Prim1: 5′- AATCCCCCCC −3′ and Prim 2: 5′- CTGACCCCCC −3′) were equally mixed in annealing buffer (10 mM Tris-HCl pH7.5, 1 mM EDTA, 50 mM NaCl, 10 mM KCl) and then heated to 95 °C for 5 min and slowly cooled down to room temperature. DNA synthesis reactions were assayed with 10 nM DNA template 1 or 2, indicated amount of POL2A proteins in Supplementary Fig. 12, 10 µM dATP/dGTP/dTTP, and 2 µCi of [α−32P]-dCTP (NEG513H) in a buffer containing 40 mM Tris-HCl pH 7.5, 10 mM MgCl$_2$, 50 mM potassium glutamate and 10 mM DTT. After incubation at 30 °C for 60 min, 2 µL loading buffer (Beyotime: D0071) was added to the reaction products. The samples were then fractionated on a 20% denaturing polyacrylamide gel containing 3 M urea, and imaged using Typhoon FLA 9500.

## rGTP incorporation experiments

100 µM DNA template 3 (5′- CCCCCCCCCCCCCCCCCCCCCCCCC CCCCCCCTCAG −3′) and its primer (5′- CTGAGGGGG −3′)were annealed as described above. rGTP incorporation experiments were assayed with 10 nM DNA template 3, 1 pmol POL2A protein, 10 µM dATP/dCTP/dTTP, 5 µM dGTP, and 2 µCi of [α−32P]-rGTP (NEG006H) in a buffer containing 40 mM Tris-HCl pH 7.5, 10 mM MgCl$_2$, 50 mM potassium glutamate and 10 mM DTT. The reactions were stopped by adding 2 µL loading buffer (Beyotime: D0071) after incubation at 30 °C for 10 min, 20 min, 40 min, and 60 min, then the products were detected as described above.

## Sequencing data analysis

The ssDRIP-seq data was processed as described previously[32,33]. Briefly, reads were aligned to the TAIR10 genome using Bowtie2 (V2.3.5.1) and duplicates were removed with Picard tool (V2.24.2). The aligned reads files (BAM) were converted to normalized coverage files (bigWig) with 5-bp bins using bamCoverage from deep-Tools (3.5.0). Snapshots of the data were constructed using the Integrative Genomics Viewer (IGV). Metaplots on related elements and heatmaps on DNA origins were generated with deepTools (3.5.0), representing the mean of read coverage, the 1 kb upstream and downstream regions were included. We set the "binSize" as 5, the "regionBodyLength" (protein coding genes or all annotated genes) as 2,000 (400X "binSize"), and the upstream and downstream regions (1 kb each) were each set as 1,000 (200X "binSize"). Differential R-loop gene analysis was performed using DESeq2, while q-values (adjusted *P*-values) and log2FC values were used to determine significant differences. regionR was used for enrichment analysis between R-loops and histone modifications or DNA replication regions. The histone mark region sequence was downloaded from NCBI GSE2839842, The ORC1 and CDC6 binding sites were defined as DNA replication origins referred to[41], and the data was downloaded from GSE21928.

SHOREmap was used for candidate-gene identification with default parameter[40].

## Statistical analysis

Statistical analysis was performed using Tukey's multiple comparisons test with One-Way ANOVA. NS, no significance; *$p < 0.05$; **$p < 0.01$; ****$p < 0.001$; ****$p < 0.0001$. The uncertainty in the mean is reported as the standard deviation (SD) of the mean.

## Reporting summary

Further information on research design is available in the Nature Portfolio Reporting Summary linked to this article.

## Data availability

The raw sequencing data and processed files in this study have been deposited in the NCBI's Gene Expression Omnibus (GEO) database and are accessible through GEO Series accession number GSE216170. Other datasets including the histone marker region and DNA replication region discussed in this study are available at NCBI's GEO through accession numbers GSE28398 and GSE21928. Source data are provided with this paper.

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

## Acknowledgements

We greatly thank all the members in The Sun Lab for their helpful discussions and constructive suggestions, and Professor Zhizhong Gong from China Agricultural University for providing the *abo4-1* seeds. This work was funded by grants from the National Natural Science Foundation of China (grant nos. 32261133529, 91940306 and 31822028 to Q.

Sun, 32100428 to J. Zhou, and 32070651 to W. Zhang). The Sun Lab is supported by Tsinghua-Peking Center for Life Sciences, W. Zhang is supported by the China Postdoctoral Science Foundation Project (2019M660610), and J. Zhou and W. Zhang are supported by postdoc fellowship from Tsinghua-Peking Center for Life Sciences.

## Author contributions

Q.S. conceived the study, and designed the experiments with Q.L. J.Z. assisted in POL2A purification and ssDRIP-seq and DRIP-qPCR experiments. S.L. performed the RNase H activity of AtRNH1A. W.Z. preformed the polymerase activity of POL2A. Y.D. assisted EdU labeling experiment. K.L. assisted in data analysis. Y.W. provided *til1-4* mutant and polished the manuscript. Q.L. performed the rest of the experiments. Q.L. and Q.S. wrote the manuscript. Q.Li., J.Z. and Q.S. revised the manuscript and all authors read and approved the final manuscript.

## Competing interests

The authors declare no competing interests.
