## [Peer Review File · Nature Communications]

DNA polymerase ϵ harmonizes topological states and R-loops formation to maintain genome integrity in ArabidopsisEditorial Note: Parts of this Peer Review File have been redacted as indicated to maintain the confidentiality of unpublished data and to remove third-party material where no permission to publish could be obtained.

REVIEWER COMMENTS

Reviewer #1 (Remarks to the Author):

Replication-transcription collision is one of the most severe challenges for genome stability. Li et. al. found that inhibiting DNA topoisomerase 1 (TOP1i) triggers the global increase of R-loops and relevant DNA damages. In addition, the effects are exacerbated in the DNA damage repair mutant *atm*. It implies except ATR, ATM is also involved in resolving replication-transcription collision to maintain genome stability. Furthermore, the authors identified a mutation in a conserved domain of the POL ϵ catalytic subunit, the domain is required for faithful DNA synthesis, and the mutant rescues TOP1i-induced R-loop accumulation and cell death in *atm*. The authors finally concluded that DNA replication is in concert with topological states to finetune R-loops and thereby maintain genome integrity. The data of the study is clear and nearly enough to support the conclusions. Moreover, the study found a possible requirement of ATM in the replication-transcription collision. However, it's a pity that the paper doesn't show why ATM is required, or why ATR is not enough to overcome the replication-transcription collision. So I will only be positive about the publication unless they can address my major concerns below:

Major concerns:

1. The most ambiguous point in the study is the effect of *pol2aL473F* in global replication. The authors showed that the ribonucleotide mis-incorporation into the genome is only dramatically increased when *asr20* (*atm* and *pol2aL473F* double mutant) is treated with CPT. But it's not enough to understand what's the exact impact of *pol2aL473F* on DNA replication. That must be helpful to understand how *pol2aL473* rescues TOP1i-induced R-loop accumulation and cell death in *atm*. Here are two suggestions for this point. First, can authors show average replication fork speed in wt or mutants? I noticed that the authors did EdU incorporation experiments shown in Fig. 7, but it's not enough to show the exact effect of *pol2aL473* mutation on replication. As the relevant, secondly, if the authors can show the impact of *pol2aL473* on POL ϵ 's activity in vitro, that will be also helpful. The idea is to purify wt and mutant POL ϵ to do the primer extension experiment. I think the impact of *pol2aL473* mutant on POL ϵ and replication has to be clear to support the model of the paper.
2. Can authors show the protein level of endogenous RNH1A or RNH2A in wt and mutants? One unlikely possibility is RNase level is changed in mutants which causes relevant R-loop level change. It's better to exclude this possibility.
3. Can authors show the ATR activation level in wt and mutants (like CHK1 or RPA phosphorylation)? It's also worth to know how ATR behaved in *atm* and *asr20* mutants. And as a part of the first major concern, the author must be clear whether *pol2aL473* and other POL2a mutants can activate ATR or ATM checkpoint under normal condition or CPT stress.
4. How is the re-productibility of some data? I must point out that some results lack quantitation and repeats. Especially, the γ H2AX results in Fig.1D and Fig.3F. The differences between samples are mild, the results must be repeated 3 times and quantitative. In addition, if authors can repeat all of the slot blot experiments to detect the R-loop level 3 times and then give quantitation for each sample, that will be perfect.

Minor concerns:

1. In the paragraph of "**Identification of the *atm* suppressor *asr20***", the authors mentioned, "This suggests that both the *asr20* double mutant and the topoR-loops could trigger ATM-independent H2A.X phosphorylation pathways." I don't really get what the authors mean here. The *asr20* mutant is ATM-deficient, so H2AX must be phosphorylated by ATR rather than ATM.
2. In Fig.9D, why authors used *abo4-1* instead of *pol2aL473*? Back to the first major concern, if the authors think mutants *til1-4*, *pol2aL473*, and *abo4-1* show similar phenotypes, please show more evidence (ex: in vitro activity) apart from root length.
3. One more unlikely possibility, can authors show the protein level of POL2a in wt and mutants? Just to exclude the possibility that the mutations reduce the protein level of POL ϵ .

Reviewer #2 (Remarks to the Author):

The paper by Li et al focused on understanding of mechanisms of Top1 and DNA topology in regulating R-loops. They carried this analysis in Arabidopsis, employing Top1 inhibitor, camptothecin (CPT). Initially the authors have shown that treatment with CPT induced R-loops accumulation and inhibits root growth, and these phenotypes are exacerbated in ATM mutant. Furthermore, the authors carried out a genetical screen which identified DNA pol subunit mutation which suppresses R-loop accumulation and genome instability observed in ATM-mutant cells treated with CPT. These results suggest that DNA replication acts in concert with topological stress to fine-tune R-loops and maintain genome integrity. The authors also propose that these results may point towards the conserved regulatory mechanism of Top1 inhibitors resistance in chemotherapy for ATM-deficient tumors.

Overall speaking I found the paper to be quite difficult to read and follow. The authors have carried out a lot of experiments but unfortunately, they have not been put together in a nice and cohesive story. Previous work in plants have demonstrated the role of Topoisomerase in R-loop biology and root development in rice (Ref 29 in the paper). Work in bacterial and human systems have also demonstrated the role of transcription-replication conflicts in R-loop biology (Cimprich and Merrikh labs). So personally I am struggling to see the novelty in this story (apart from the fact that it was done in a different organism, i.e. Arabidopsis). This in addition with multiple comments (see below) related to the quality of the presented data, lack on controls in many experiments and overall poor presentation and explanation of the results, makes this paper a weak contender for publication in Nature Comm journal in its current form.

Major comments:

1. The paper uses S9.6 antibody to assess R-loop status in Arabidopsis. This antibody have received a lot of criticism in the literature due to the ability of S9.6 to recognize ds RNA. Unfortunately, there is no RNase H control for the slot blots in the paper (Fig.1C, 3E, 10C). This is particularly important, since the authors seem to extract total R-loop content from the root. Quantification of all slot blot gel should be provided next to the gel.
2. I am concerned regarding the experimental set-up of CPT experiments. According to the Methods section the seeds were grown in CTP for 5-7 days. CTP is known to substantially affect transcription by Pol II within 30 min in human cells. I wonder why the authors have chosen this time point, and how sure are they that the effects of CTP on R-loop are direct, and not just overall toxic effects of CTP on transcription due to Topoisomerase inhibition? Did they carry out some CTP titration experiments to select this time-point? Is transcription affected by this treatment?
3. In figure S2 the authors have carried out R-loop DRIP-seq, however this experiment also does not have RNase H control. The authors show the track for 2 replicas. However, the authors really need to show the input lane for these tracks, to demonstrate the level of enrichment. Here we see that CPT treatment reduced R-loops (which will be in line with my point above suggesting a decrease of transcription). In general, I do not see any tracks suggesting R-loop increase following CTP treatment (in WT conditions), also the bioinformatics data does not show it, so I am not exactly sure why the authors claim this in the paper? This CTP-associated increased in seen in Fig 1C but not reproduced in Fig. 3E (in fact we see a decrease of R-loops) as observed in DRIP-seq experiments. Similarly, R-loop decrease is observed in *asr20* mutant following CTP treatment. Is this also due to transcriptional decrease?
4. This paper contains multiple DRIP-seq genome-wide analysis and contain bioinformatic analysis (Fig. 5 and S2). However, none of the conclusions have been validated by DRIP-qPCR on specific

genomic locations. This has to be incorporated to ascertain the strengths of the conclusions. Especially the finding that accumulation of R-loops in CTP conditions on replication origins needs DRIP-qPCR validation compared to non-origin genomic regions. It is also not clear to me how the origins were defined bioinformatically since this is not mentioned in the methods either.

5. The figure 6C (analysis of replication origins and R-loop shifts) was not clear to me and needs to be explained in a better way. I could not see how the authors concluded that there were shifts of R-loops in respect to the origin. This analysis also was not included in the Methods section and the statistical analysis is missing as well.

6. I do not fully agree with the authors regarding their conclusions that HU relieved CPT effects. HU did not have any effect on root growth. Most probably (based on the slot blot result), HU is blocking transcription, and hence less R-loops are formed in the first place to cause any effect, and therefore combined treatment is less toxic. Did the authors check the level of transcription in HU cells?

7. The paper proposes some ideas on how the regulation of topoR-loops may happen, however I would wish the authors have tested their ideas a bit more extensively. So far the paper relies on genetic screens and DRIP-seq analysis of cell populations. The authors need to provide some clear validations of their data on single gene loci to support the validity of their models. Some parts of the paper are made unnecessary complicated. In my opinion, the paper needs to be re-written in a more focused way.

Minor comments:

1. The paper contains 11 main and 11 Supplementary figures, which is excessive. They can be re-organized into less figures, especially for the main figures.

2. The paper would benefit from substantial English editing. Some sentences do not make a lot of sense. Some factual scientific meaning is obscured/mis-represented in multiple places (for example, 2nd introduction paragraph: Third sentence `RNA and DNA etc...' should have a stop at the end. The next sentence `In addition' Contains awkward grammar `in where exonucleases XPG and XPF...' – this should be corrected).

Fig 4 legend – Pol 2 is conserved

3. The detail of the activity assay in Fig S3 is missing. There should be an indication of the amounts of substrate (and its nature) and recombinant protein added. The mutant enzyme contains a big band at ~40 kDa which may affect its activity in vitro irrespective of catalytic mutation.

4. Overall paper contains very brief methodology, making it impossible for anyone to follow or reproduce any of the experiments. The details need to be expanded. There is no information on DRIP-seq, bioinformatics analysis or suppressor screen to identify *ars20* (how long was the drug treatment?). Supplementary information can be also used for that.

Reviewer #3 (Remarks to the Author):

In their work entitled « DNA polymerase ϵ strengthens topological R-loops-promoted genome instability in Arabidopsis », Li and colleagues investigated the hypersensitivity of the *atm* mutant to the Topoisomerase I inhibitor camptothecin (CPT), finding that CPT induces R-loops accumulation leading to the formation of DSBs. They identify a mutation in the replicative polymerase DNA pol ϵ as a suppressor of *atm*'s hypersensitivity to CPT, hinting at transcription/replication conflicts as the primary trigger for CPT-induced DSB formation.

This manuscript contains a considerable amount of work and data, and the findings are very exciting. I am convinced this work has the potential to bring very important knowledge that will be of interest for a broad community. However, in its present form it remains preliminary, both because some conclusions are not sufficiently supported by the data or sometimes even in complete contradiction with what is shown on figures, and because it is in places not sufficiently clearly written. Below are listed a number of suggestions and comments for the authors that I think should be addressed

- Results
- I have concerns about data related to gamma H2AX accumulation. Authors write "The γ H2AX signals in Col-0 and atm were increased by CPT treatment, while atm impeded γ H2AX levels in the control » but the western blot quantification does not show any increase. If anything, the signal is slightly lower in atm treated with DMSO than in all other samples. Regarding the immuno-staining, authors chose to quantify global fluorescence instead of counting foci, or nuclei with foci. Looking at the Figures, I understand why they did that, because foci are not clearly visible, but that raises questions about the specificity of their antibody. They can check the following references doi: 10.15252/embj.201694571, doi: 10.1093/plcell/koab158 or doi: 10.1105/tpc.15.00898 for pictures of specific gamma H2AX labelling and appropriate quantification methods. This comment applies to Figures 1, 3, 8 and S4
 - I think there is something I am missing in the way authors generated the metaplots shown throughout the manuscript to illustrate the overall distribution of R-loops on genes. For example, it is clear from the PCA shown on Figure S2 that the atm CPT sample is different from all other samples. Authors write in the text that « Results showed slightly elevated R-loops in atm relative to wild-type (...) and [R-loops] increased dramatically in atm upon CPT treatment ». This is clear from the screenshot shown on Figure S2D. Yet, on the metaplots, only the atm DMSO sample seems to differ from the others and to display lower levels of R-loops. This looks like a normalization issue and should be solved. This comment also applies to Figure 5. The question here is really crucial, because if the metaplots were to be trusted, on Figure 5 I would conclude that the pol2a mutation isolated in the screen leads to a massive decrease in R-loop accumulation compared to the wild-type even under control conditions, and that this is true not only near the TTS, where authors think it reflects TRC, but also on the opposite strand near the TSS. By the way, the biological meaning of these TSS R-loops is never described or discussed in the text, and I think this is really missing for readers who would not be familiar with R-loops and their roles in the cell.
 - A lot of information is missing from methods and figure legends, and that makes the manuscript really difficult to read. For example, by comparing Figures 1 and 2, it is obvious that either the dose of CPT, or time of treatment, or both are not the same: on Fig1, the atm mutant shows a completely disorganized root tip, whereas on Fig2 it is not so affected, but I could not find the information anywhere.
 - I did not manage to understand what I was supposed to see on Figure 6C. I think this is possibly because the legend is not sufficiently detailed.
 - I fully agree with authors that the effect of the pol2A mutation on R-loop formation could reflect a change in replication speed, and that attempting to measure it would provide very valuable information. I have strong concerns about the way the experiment was performed and what authors conclude from it. First, I do not quite follow how EdU labelling can continue to increase if the molecule is washed away after 30 min, but maybe there is still enough of it inside the cells to allow continued incorporation. However, I am not convinced by the quality of the data. The flow-cytometry graphs show extremely poorly-defined groups of nuclei, which is likely to prevent robust quantification. Besides, these data cannot easily be translated into replication speed. It is well documented that pol epsilon mutants show a higher proportion of S-phase cells when EdU incorporation is performed for a relatively short time, but this increase results from a dramatic lengthening of S-phase, the consequence of which is that S-phase represents a larger proportion of the total cell cycle length, and that at any given time point, a larger proportion of the cells are in S-phase (doi:10.1104/pp.17.00031). Performed in this way, the experiment cannot be conclusive in the way authors would like to use it. Even if their conclusions were more robust, the increase in the proportion of EdU labelled nuclei would reflect a combination of things (replication speed + speed of cell cycle progression). To conclude on replication speed, they should perform DNA combing or spreading to prove replication fork speed is modified in their mutant.

- In relation with the above-mentioned point, I think the novel pol epsilon allele isolated here is insufficiently characterized. In terms of root growth, it seems less affected than *abo4-1* on control conditions, but a more detailed phenotypic description would be extremely important to decipher to what extent the L473F mutation has unique consequences on the replication process compared to other point mutations affecting pol epsilon.
- On Figure 9, authors show that *asr20* accumulates ribonucleotides in its DNA. However, they introduce the *rnh2a* mutation in the *abo4-1* background rather than in *ars20* or *pol2aL473F*. Why is that? A mutation in the *rnh2* gene has been isolated as a suppressor of the hypersensitivity of the *wee1* mutant to hydroxyurea (doi: 10.1080/15592324.2014.1001226). How does this fit with the authors results?
- When describing Figure 10, authors write “*rad52* and *ku70* presented CPT tolerance and partially restored the short root phenotype of *atm* after CPT treatment ». This is simply not what the figure shows. The single mutants behave like the wild-type, so they cannot be called tolerant, and although I agree the *rad52 atm* mutant grows better than *atm*, this is clearly not the case for *atm ku70*, if the difference is really significant as suggested by the stars above the plots (although the difference is extremely slight), the double mutant does worse than *atm*, which would make sense if NHEJ is required to repair DSBs caused by R-loop accumulation.
- I am not sure the experiment with MMS treatment is really useful. This drug has extremely complex effects as it does block replication fork progression, but also causes the formation of DNA breaks. It is not clear to me what hypothesis authors wanted to test with this experiment, but I would suggest a more specific treatment, depending on the point they are trying to make.
- I am surprised there is no access given to reviewers to the deep-sequencing data for the reviewing process

Discussion

Given the considerable amount of data generated, I think the discussion is really incomplete, and does not do justice to the work in its present form. In the specific points I listed for the results, I mention a number of references that authors could/should have cited to put their work in the global context of the plant DNA damage response, as their results can really bring novel and valuable information for this field. Also, it is apparent that they have written this section a bit rapidly. For example, they wrote “Importantly, the biological requirements for ATM and ATR are particularly different, as ATR, but not ATM, is essential for embryogenesis and viability. While *atr* shows no visible growth differences from wildtype Col-0, *atm* is partially sterile » This makes no sense if you don’t specify that ATR is essential IN MAMMALS, but shows no growth difference with the wild-type IN ARABIDOPSIS (which by the way is not quite true in maize or barley).

Overall, I think the working model and conclusions are still not clear. Authors write that *rhnh2* suppresses the resistance of pol epsilon mutant to CPT, but these mutants are not resistant to CPT, what the mutation does is that it makes them sensitive. How this happens since the levels of ribonucleotides in the DNA are not measured in the *abo4* mutant is not clear. Authors should also clarify how they think this incorporation happens. I am not sure whether they mean they are incorporated by Pol epsilon itself, or whether they consider TLS polymerases are recruited to resolve stalled forks and are responsible for this phenomenon. More generally, I am not sure of the model shown on Figure 11 and title of the paper. As I mentioned earlier, I am not at all convinced that pol epsilon mutants show uncontrolled replication, to me, previous knowledge rather points to the opposite. It is indispensable to clarify this point by performing appropriate experiments, or to rethink the model taking into accounts alternative hypotheses. Likewise, I am not convinced that DNA polymerase epsilon “strengthens R-loops”. Based on Figure 6A, I would say that R-loops are dramatically increased at replication origins in *pol2a* mutants, possibly due to fork stalling that hampers the resolution of these R-loops. This effect is less pronounced, but still visible in the *atm* background, so the authors’ conclusions are really not in agreement with the data.

I am sorry I cannot give a more positive evaluation of this work at this stage, and there are many more examples of places in the manuscript where the conclusions are either not clear, or not quite in agreement with what is shown in the Figures. I am convinced however that if authors address all these comments, and carefully polish the writing giving more details in the text, in the legend of figures and discussing their work more in depth, their work has the potential to make an extremely

good paper.

First of all, we sincerely thank all the reviewers for their perceptive and constructive comments, which give us the chance to improve the quality of our research. In our revision we addressed all the issues mentioned, and answered all the questions. We modified the original figures and text accordingly, and the revised version contains 11 figures and 13 supplementary figures.

Reviewer #1 (Remarks to the Author):

Replication-transcription collision is one of the most severe challenges for genome stability. Li et. al. found that inhibiting DNA topoisomerase 1 (TOP1i) triggers the global increase of R-loops and relevant DNA damages. In addition, the effects are exacerbated in the DNA damage repair mutant *atm*. It implies except ATR, ATM is also involved in resolving replication-transcription collision to maintain genome stability. Furthermore, the authors identified a mutation in a conserved domain of the POL ϵ catalytic subunit, the domain is required for faithful DNA synthesis, and the mutant rescues TOP1i-induced R-loop accumulation and cell death in *atm*. The authors finally concluded that DNA replication is in concert with topological states to finetune R-loops and thereby maintain genome integrity. The data of the study is clear and nearly enough to support the conclusions. Moreover, the study found a possible requirement of ATM in the replication-transcription collision. However, it's a pity that the paper doesn't show why ATM is required, or why ATR is not enough to overcome the replication-transcription collision.

Thanks for the perceptive comments. ATR is activated in response to DNA lesions induce single-strand DNA formation, while ATM is activated after DSB formation. As discussed in our revised version, TOP1i produced single-ended DSBs and topoR-loops accumulations could primarily trigger a cascade events leading to TRCs-induced DSBs and activating ATM repair pathway. Moreover, in Arabidopsis, *atr* shows no visible growth differences from wildtype, but *atm* is partially sterile, suggesting ATM plays a broader role than ATR in Arabidopsis. These may be the reason why *atm* but not *atr* is hypersensitive to CPT.

So I will only be positive about the publication unless they can address my major concerns below:
Major concerns:

1. The most ambiguous point in the study is the effect of *pol2aL473F* in global replication. The authors showed that the ribonucleotide mis-incorporation into the genome is only dramatically increased when *asr20* (*atm* and *pol2aL473F* double mutant) is treated with CPT. But it's not enough to understand what's the exact impact of *pol2aL473F* on DNA replication. That must be helpful to understand how *pol2aL473* rescues TOP1i-induced R-loop accumulation and cell death in *atm*. Here are two suggestions for this point. First, can authors show average replication fork speed in wt or mutants? I noticed that the authors did EdU incorporation experiments shown in Fig. 7, but it's not enough to show the exact effect of *pol2aL473* mutation on replication. As the relevant, secondly, if the authors can show the impact of *pol2aL473* on POL ϵ 's activity in vitro, that will be also helpful. The idea is to purify wt and mutant POL ϵ to do the primer extension experiment. I think the impact of *pol2aL473* mutant on POL ϵ and replication has to be clear to support the model of the paper.

Thanks for these constructive suggestions. To address your comments, we had spent long time to perform the DNA fiber spreading assay (see Response Figure 1), however, we failed to setup this experiment in our system. DNA spreading assay is rarely done in plants. We first extracted the nuclei

of root tips for the spreading test referred to PMID: 34260408. In the beginning, we used the superfrost microscope slides referred to “DNA Fiber Spreading Assay to Test HDACi Effects on DNA and Its Replication” (PMID: 27761816), but we could not observe DNA fibers at all. Then, we tried to use the silanized slides for DNA spreading according to the publication “DNA fiber combing protocol using in-house reagents and coverslips to analyze replication fork dynamics in mammalian cells” (PMID: 35573479), and observed thick filaments of varying lengths spreading from the lysis nuclei, but these DNA fibers are not clear for quantitative analysis (Response Fig. 1A). Therefore, we assumed that the extracted nuclei are not efficient for DNA spreading on slides. Then we further purified the genomic DNA from the nuclei of root tips for spreading, the DNA fiber was straighter and more uniform (Response Fig. 1B). We then decided to apply the genomic DNA, that are sequential pulse labeling with EdU and BrdU, for spreading. By microscopy detection, we observed that the DNA fiber were stained by YOYO-1 but the newly synthesized DNA labeled by EdU or BrdU was not observed (Response Fig. 1C). After multiple attempts, we still could not have results with labeled replication forks. It is a pity that we could not set up the DNA fiber assay in our system.

Response Figure 1. Images of DNA spreading fiber. (A) Spreading nuclei stained with YOYO-1. (B) Spreading genomic DNA stained with YOYO-1. (C) Spreading genomic DNA labelled with EdU and BrdU and stained with YOYO-1. Diagram of DNA spreading are shown above. The 4-day-old seedlings were sequential labelled with EdU and BrdU, nuclei and genomic DNA were extracted from root tips, DNA was combed onto silanized coverslip. Green tracks represent the first pulse labeling of EdU detected by Alexa Fluor 647 dye after click reaction. Red tracks represent the second pulse labeling of BrdU detected by specific anti-BrdU antibody and Alexa555-coupled secondary antibody. Green tracks represent total DNA fibers stained with YOYO-1. Scale bar: 20 μ m.

Alternatively, we adopted the double-labeling strategy referred to “Distinct roles of Arabidopsis ORC1 proteins in DNA replication and heterochromatic H3K27me1 deposition” (PMID: 36882445) to interpret a more detailed replication progression of WT and mutant polymerase in vivo. Briefly, the seedlings are firstly incubated with EdU (15 min), after different time chases, and then labeled with BrdU to determine the S phase progression and reflect the DNA replication progression. As the results shown in Fig. 7 (Line 281-310), *atm* promotes S-phase progression and *pol2a* delays it in normal conditions, and *asr20* had similar results to *pol2a*, indicating that *pol2a* could stall DNA replication. However, at CPT stress, S-phase progression was severely inhibited both Col-0, but only slightly restricted in the absence of ATM, especially in *asr20* mutants. Consequently, these results suggest that *pol2a*^{L473F} and *asr20* delay the DNA replication and S phase progression in normal growth conditions, but interrupt the DNA replication and S phase progression stalling under consistent topological stress.

Moreover, we had purified wild-type and mutant POL2A proteins for primer extension experiments as suggested. Given the fact that the molecular weight of POL2A is very large (~250kDa), we failed to purify these full-length proteins either in prokaryotic or mammalian expression systems. Therefore, we purified the truncated POL2A (1-1123 aa, ~135 kDa, including functional domains) and mutant POL2A^{L473F} with GST-tag (~160 kDa in total) (Fig. S12A-B). The primer extension results indicated that POL2A^{L473F} slightly delay the DNA replication comparing to that of wild-type POL2A^{WT} (Fig. S12C). Additionally, POL2A^{WT} and POL2A^{L473F} exhibited comparable efficient of rNTP incorporation in vitro with the substrate alpha-32P-rGTP (Fig. S12D). Combining the results (Fig. 9A-B and S11A-B) which high levels of rNTP were detected only in *asr20* under topological stress, we concluded that TOP1, ATM and POL2A might collaboratively function in DNA replication progression and replication fidelity.

Our DNA polymerase activity results in vitro showed that POL2A mutation has slightly defects in DNA replication speed, but did not increase the rNTP incorporation level, which could not explain the activity of DNA polymerase ϵ in vivo (Fig. 7, Fig. 9 A-B and Fig S11A-B). POL ϵ works as a complex and POL2A interacts with many other factors in vivo, so the mutation site may not change the DNA replication speed directly but also the interaction patterns which further change the noncanonical and unclarified role beyond replication, that is why we can only detect more rNTP incorporation errors in *asr20* after CPT treatment. Our genetic results had given clear evidences that mutation of POL ϵ speeds up DNA replication and increases the rNTP incorporation in *atm* mutant background under topological stress.

In addition, this suppressors screen given us the chances to systematically dissect the topological stress, R-loops, and DNA damage repair. Our ongoing screen had isolated many mutants (Response Fig. 2), and some of them might be in regulation the activity of POL2A^{L473F} thus influences genome integrity and the root length.

Response Figure 2. Phenotypes of other *atm* suppressors. Root phenotypes of other *asr* mutants treated with 30 nM DMSO or CPT for 5 days. Scale bar: 1 cm.

2. Can authors show the protein level of endogenous RNH1A or RNH2A in wt and mutants? One unlikely possibility is RNase level is changed in mutants which causes relevant R-loop level change. It's better to exclude this possibility.

Thanks for this valuable suggestion. Due to the lack of available anti-AtRNH1A or AtRNH2A native antibodies, we generated transgenes plants with native promoter-driven AtRNH1A-GFP constructs in Col-0, *atm*, *asr20* and *pol2a*^{L473F}, and the western blotting results show that there are comparable AtRNH1A protein levels among these genetic materials (Response Fig. 3A). We also generated native promoter-driven AtRNH2A-GFP transgenes, but we did not detect the GFP signal by western blot. Alternatively, we performed RT-qPCR to detect the endogenous expressions of AtRNH2A. The RT-qPCR results show that AtRNH2A expression level has no significant differences in Col-0 and mutants (Response Fig. 3B). Combining with our results of inducible AtRNH1A and AtRNH1A^{mut} (Fig. 2), we believe these results could support the conclusion that the changed R-loop levels in the mutants are not due to the different amounts of endogenous AtRNH1A or AtRNH2A among Col-0 and mutants.

Response Figure 3. Endogenous AtRNH1A and AtRNH2A expression levels are comparable in WT and mutants. (A) Protein levels of native promoter driven AtRNH1A-GFP in Col-0, *atm*, *asr20* and *pol2a*^{L473F} under normal conditions. CBB stained membrane of total protein was used as loading control. (B) AtRNH2A expression in in Col-0, *atm*, *asr20* and *pol2a*^{L473F} under normal conditions using real-time qRT-PCR. Data are normalized to UBC9 gene and shown as the mean ± SD, NS: not significant (one-way ANOVA).

3. Can authors show the ATR activation level in wt and mutants (like CHK1 or RPA phosphorylation)? It's also worth to know how ATR behaved in *atm* and *asr20* mutants. And as a part of the first major concern, the author must be clear whether *pol2a*^{L473F} and other POL2a mutants can activate ATR or ATM checkpoint under normal condition or CPT stress.

Thanks for the suggestions. In Arabidopsis, the ATR is activated by RPA complex, but there is no CHK1 homologous protein downstream of ATR pathway (PMID: 26653616, PMID: 31164899, PMID: 28622525). Downstream to both ATM and ATR, SOG1 and WEE1 are factors responding to DNA damage in Arabidopsis (PMID: 26653616, PMID: 31164899). Due to the lack of available antibodies of proteins mentioned above, we could not obtain the results to directly show whether the ATR pathway is activated in wt Col-0 vs mutants. Alternatively, we had performed the RT-qPCR of *RPA1C*, *SOG1* and *WEE1* to check the expression levels of ATR or ATM checkpoint among mutants under normal conditions or CPT stress (Response Fig. 4). The results showed that the expression level of *RPA1C*, *SOG1* and *WEE1* had no significant differences between wt and mutants. CPT activates *RPA1C* and *WEE1* expression in Col-0 and *pol2a*^{L473F} but not in *atm* and *asr20*. Even though, these results could not indicate there is no activation of ATM or ATR, because we indeed detected the changed γ H2AX levels in wt vs *asr20* under normal growth condition (Fig. 3F). Combining the similar phenotype results in Fig. 4, Fig. 9, Fig. S6-7, Fig. S11, and Fig. S13 and Response Fig. 6, we think the other two *pol2a* mutants should have the same effects on activating ATR or ATM checkpoint as *pol2a*^{L473F}.

Response Figure 4. Expression of genes related to ATR pathway are comparable in WT and mutants. The relative expression levels of *RPA1C*, *SOG1* and *WEE1* in Col-0, *atm*, *asr20* and *pol2a*^{L473F} under control (DMSO) and CPT conditions using qRT-PCR. Data are normalized to UBC9 gene and shown as the mean \pm SD. Different letters indicate statistically significant differences according to a one-way ANOVA analysis with the Kruskal–Wallis test and Dunn’s multiple comparisons ($p < 0.05$). NS: not significant.

Moreover, according to PMID: 24292646, the β -strand formed by conserved residues 533–555 (LLASETYVGGHVESLESGVFRSD) in the palm domain of POL2A in yeast is important for the polymerase activity of Pol ϵ , the mutation and deletion of the palm domain impaired DNA processivity of Pol ϵ . *abo4-1* (G522N) mutant sites (the underlined and magnified G) of Arabidopsis just located in this region, which hint mutations of the conserved junction domain in Arabidopsis impede DNA polymerase activity of POL2A.

4. How is the re-reproducibility of some data? I must point out that some results lack quantitation and repeats. Especially, the γ H2AX results in Fig.1D and Fig.3F. The differences between samples are mild, the results must be repeated 3 times and quantitative. In addition, if authors can repeat all of the slot blot experiments to detect the R-loop level 3 times and then give quantitation for each sample, that will be perfect.

Thanks for the suggestions. In this revised version, we had added the replicates (n=3) for γ H2AX and slot blot experiments, and also provided the statistic results for each experiment (Response Fig. 5, Response Fig. 9 and Response Table 1). Although all the replicates showed similar trends as those in the original version, we are not sure it is the good way to quantify and average the blotting results. We had calculated them separately in Response Table 1.

Response Figure 5. Experimental repetition of γ H2AX western blot. (A) The repeated western blot results related to Fig 1D. Coomassie brilliant blue (CBB)-stained membrane of total protein was used as loading control. Three biological replicates showed similar results. Statistic data are normalized to Col-0 in DMSO condition and shown as

the mean \pm SD of three replicates, * P <0.05, ** P <0.01, *** P <0.001 (one-way ANOVA). **(B)** The repeated western blot results related to Fig 3F. CBB-stained membrane of total protein was used as loading control. Three biological replicates showed similar results. Statistic data are data are normalized to Col-0 in DMSO condition and shown as the mean \pm SD of three replicates, * P <0.05, ** P <0.01 (one-way ANOVA).

Minor concerns:

1. In the paragraph of "Identification of the *atm* suppressor *asr20*", the authors mentioned, "This suggests that both the *asr20* double mutant and the topoR-loops could trigger ATM-independent H2A.X phosphorylation pathways." I don't really get what the authors mean here. The *asr20* mutant is ATM-deficient, so H2AX must be phosphorylated by ATR rather than ATM.

Thanks for pointing out this. We have modified the text to "This suggests that both the *asr20* double mutant and the topoR-loops may trigger H2A.X phosphorylation which are most possibly ATR-dependent" (Line 180-181).

2. In Fig.9D, why authors used *abo4-1* instead of *pol2aL473*? Back to the first major concern, if the authors think mutants *til1-4*, *pol2aL473*, and *abo4-1* show similar phenotypes, please show more evidence (ex: in vitro activity) apart from root length.

In the original version, we had not obtained the *pol2a^{L473}/atrnh2a* double mutant, and thus we used *abo4-1* instead. In this revision, we have generated the *pol2a^{L473}/atrnh2a* double mutants, and compared the root length among related mutants (Fig. 9C-D), and the results are similar to that in the previous version (Fig. S11C-D).

We observed vegetative and reproductive development phenotypes of WT and different mutants, and found that all the three mutants showed early flowering phenotype (Response Fig. 6A). The mutants also showed obvious sterility that the siliques were shorter and more shriveled than WT, and *atm* further enhanced these defects (Response Fig. 6B). the similar growth phenotypes of *pol2a^{L473}*, *til1-4* and *abo4-1* and the results we mentioned above (response to Major comments 3) suggest the mutations of the conserved domain have the same effects.

Response Figure 6. *pol2a* impair plant development and fertility. (A) Flowering phenotypes of 15-d-old WT and mutants grown in soil. Scale bar: 1 cm. (B) Comparison of siliques in WT and mutants. Scale bar: 1 cm.

3. One more unlikely possibility, can authors show the protein level of POL2a in wt and mutants? Just to exclude the possibility that the mutations reduce the protein level of POLε.

Thanks for the comments. We could not check the protein level of POL2A in wt and mutants due to

the lack of available native anti-POL2A antibodies, and we either could not generate POL2A transgenes expressed in planta, probably due to the GFP-fused protein size being too large (more than 270 kDa in total). Alternatively, we had performed RT-qPCR to detect *POL2A* expression in wt and mutants. As shown in the results (Response Fig. 7), *POL2A* showed no significant differences in wt and mutants, indicating that there is comparable expression level of POL ϵ in wt and mutants.

Response Figure 7. Expression of *POL2A* are comparable in WT and mutants. The relative expression levels of *POL2A* in Col-0, *atm*, *asr20* and *pol2a*^{L473F} using qRT-PCR. Data are normalized to UBC9 gene and shown as the mean \pm SD. NS: not significant (one-way ANOVA).

Reviewer #2 (Remarks to the Author):

The paper by Li et al focused on understanding of mechanisms of Top1 and DNA topology in regulating R-loops. They carried this analysis in Arabidopsis, employing Top1 inhibitor, camptothecin (CPT). Initially the authors have shown that treatment with CPT induced R-loops accumulation and inhibits root growth, and these phenotypes are exacerbated in ATM mutant. Furthermore, the authors carried out a genetical screen which identified DNA pol subunit mutation which suppresses R-loop accumulation and genome instability observed in ATM-mutant cells treated with CPT. These results suggest that DNA replication acts in concert with topological stress to fine-tune R-loops and maintain genome integrity. The authors also propose that these results may point towards the conserved regulatory mechanism of Top1 inhibitors resistance in chemotherapy for ATM-deficient tumors.

Overall speaking I found the paper to be quite difficult to read and follow. The authors have carried out a lot of experiments but unfortunately, they have not been put together in a nice and cohesive story. Previous work in plants have demonstrated the role of Topoisomerase in R-loop biology and root development in rice (Ref 29 in the paper). Work in bacterial and human systems have also demonstrated the role of transcription-replication conflicts in R-loop biology (Cimprich and Merrikh labs). So personally I am struggling to see the novelty in this story (apart from the fact that it was done in a different organism, i.e. Arabidopsis). This in addition with multiple comments (see below) related to the quality of the presented data, lack on controls in many experiments and overall poor presentation and explanation of the results, makes this paper a weak contender for publication in Nature Comm journal in its current form.

Thanks for your insightful comments and suggestions. We have revised and edited the article based on your kind suggestions.

The reported work of Ref29 mainly studies the formation of R-loop in rice auxin related gene loci, and their effects of gene expression. When OsTOP1 is inhibited or treated with CPT, R-loops on auxin gene accumulate excessively and affect the expression of the gene, and eventually inhibit the growth and development of roots.

Our study focused on the function of TOP1 and DNA topology in R-loop regulation and genome stability in the scope of the whole genome level, and combined with genetic screening and high-throughput sequencing analysis, we found that the replication process itself was affected by R-loop. Moreover, the mutation of POL2A, the catalytic subunit of DNA polymerase ϵ , globally changes the distribution and level of R-loops on the genome by regulating the replication process, exactly demonstrating the direct connections between DNA replication and R-loops. In addition, *pol2a* suppresses the sensitivity of *atm* to CPT, an antitumor drug, which opens up new opportunities for patient stratification and additional therapeutic vulnerabilities for clinical exploitation.

As you could probably find in our study, we had applied the powerful genetics to address the question that how the organisms coordinate the topology states, R-loop formation and genome stability. Our step-by-step dissection first set the system to modulate genome topology (with visible phenotype in our system); and next revealed the ATM but not ATR is responsible for maintaining the topology states triggered genome stability; and then further ectopic expression of RNase H1 proved this is due to R-loops accumulation; finally, through genetic screen we identified DNA polymerase epsilon as the key regulator for triggering DNA damage when *atm* mutated.

Altogether, we think the logic and the way we dissect the biological question in our study is straightforward, and our discoveries are beyond the Arabidopsis system, and would be the general mechanism in DNA topology and R-loop regulation, and could potentially benefit the *atm* patients.

Major comments:

1. The paper uses S9.6 antibody to assess R-loop status in Arabidopsis. This antibody have received a lot of criticism in the literature due to the ability of S9.6 to recognize ds RNA. Unfortunately, there is no RNase H control for the slot blots in the paper (Fig.1C, 3E, 10C). This is particularly important, since the authors seem to extract total R-loop content from the root. Quantification of all slot blot gel should be provided next to the gel.

Thanks for the comments and suggestions. In the publication PMID: 35550870 (Response Fig. 8A), the results show that S9.6 binds much less dsRNA than RNA:DNA at the same condition (Fig 1B&D in PMID: 35550870). The similar conclusion was drawn in PMID: 35347133 that S9.6 antibodies have much higher affinities to RNA:DNA hybrid than to dsRNA (Response Fig. 8B). In addition, all the genomic DNA used in the slot blots of this study are pre-treated with RNase III (see methods, Line 583-587), which could degrade the dsRNA. Based on these, there is almost no effect of dsRNA on the recognition of R-loop by S9.6 antibody in our case.

[redacted]

Response Figure 8. Experimental repetition of slot blot. (A) S9.6 preferentially bind RNA:DNA hybrid than dsRNA. Competition experiments of the 100pM 40-bp RD (colored as blue) bound to 1 pM S9.6 with increasing concentrations of the 40-bp RR (colored as green) from 100 to 800 pM, the RR cannot compete with the RD at this concentration. Data comes from PMID: 35550870. (B) Left: Binding affinity measurements of S9.6 Fab with nucleic acids in a by fluorescence polarization titration. Apparent binding constants (Kds) are indicated. Δ FP: changes in fluorescence polarization, in mP units. ND: not determined. Right: Competition experiments of fluorescently labeled DNA-RNA hybrids bound to S9.6 with increasing amounts of unlabeled nucleic acids, colored as in left. Apparent IC50s are indicated. Data from PMID: 35347133.

In the revised manuscript, we had provided the replicates (n=3) and the quantification for all the slot blot (Response Fig. 9 and Response Table 1). We also added the RNase H control for the slot blots in revision (Response Fig. 9).

Response Figure 9 Experimental repetition of slot blot. (A) The repeated slot blot results related to Fig 1C. Sybr-gold stained membrane of total DNA was used as loading control. RNase H was negative control. Three biological replicates showed similar results. Statistic data are shown in Response Table S1. (B) The repeated slot blot results related to Fig 3E. Three biological replicates showed similar results. Statistic data are shown in Response Table S1. (C) The repeated slot blot results related to Fig 10C. Statistic data are shown in Response Table S1. (D) The repeated slot blot results related to Fig S9D. Statistic data are shown in Response Table S1.

2. I am concerned regarding the experimental set-up of CPT experiments. According to the Methods section the seeds were grown in CTP for 5-7 days. CTP is known to substantially affect transcription by Pol II within 30 min in human cells. I wonder why the authors have chosen this

time point, and how sure are they that the effects of CTP on R-loop are direct, and not just overall toxic effects of CTP on transcription due to Topoisomerase inhibition? Did they carry out some CTP titration experiments to select this time-point? Is transcription affected by this treatment?

Thanks for the comments. It was previously reported that the 10 days-treatment of 25 nM CPT can inhibit root growth (PMID: 12215507). We performed the CPT treatment at a concentration of 10 nM, 20 nM and 30 nM (Response Fig. 10), respectively, and observed that 5 days-treatment of 30 nM CPT is sufficient to the root inhibition (Response Fig. 10), so we choose the concentration of 30 nM for our experiments for at least 5 days.

Response Figure 10. CPT inhibits the root growth of *Arabidopsis thaliana*. (A) Root growth of 6-d-old Col-0 was gradually inhibited by 10, 20, and 30 nM CPT. Scale bar: 1 cm. (B) The root length of Col-0 is inversely linear to CPT concentration, R^2 : coefficient of determination.

TOP1 inhibition leads to consistent negative supercoiling during transcription, which facilitates R-loop accumulation behind RNA polymerase moving so we called these R-loops as topological R-loops in this study. To exclude the possibility that global R-loops alteration was caused by the overall effects of CTP on transcription due to Topoisomerase inhibition, we treated the wt and mutant with transcription inhibitor flavopiridol (Response Fig. 11C-D), the wt and mutants showed the similar sensitivity to FLV (Response Fig. 11A-B), indicating that the differences in the sensitivity to CPT among the mutants were not directly caused by transcription inhibition. Combing the results that AtRNH1A overexpression (Fig. 2), FLV and HU could recover the short-root phenotypes (Fig. 10 and Fig. S9), we conclude TOP1i accumulated topoR-loops that trigger transcription-replication collisions and further induce DSBs are the major cause of cell death and root inhibition.

Response Figure 11. WT and mutants showed comparable sensitivity to transcription inhibitor FLV. (A) Root growth phenotypes after treatment with different concentration of transcription inhibitor flavopiridol (FLV). Scale bar: 1 cm. **(B)** The quantitative data of root lengths for Response Fig. 11A. Lines represent mean \pm SD, $n=15$, NS: no significant (one-way ANOVA). **(C-E)** Immunoblot showing flavopiridol inhibits RNAPIIS2P signals. **(C)** Data comes from PMID: 31127286. **(D)** Data comes from PMID: 34463741. **(E)** 10 μ M flavopiridol gradually decrease RNAPIIS2P levels of Col-0 and mutant x over time. CBB stained membrane of total protein was used as loading control (Data from Dr. Jincong Zhou, in revision).

In addition, we also performed RNA-seq and ssDRIP-seq of Col-0 root tips with and without CPT treatment (data was not shown in this manuscript), the transcription is affected by CPT treatment but the R-loop dynamics was not associated with transcription dynamics. We selected the top 10 upregulated and downregulated genes, then checked the R-loops levels of these genes (Response Fig. 12). These data showed that R-loops levels in these genes are not significantly changed even though the expression are dramatically affected after CPT treatment (Response Fig. 12), suggesting that the effects of CTP on transcription are not the main causes of topological R-loop changes.

Response Figure 12. R-loops are not associated with transcription. Heatmaps of RNA-seq and ssDRIP-seq signals in of top 10 up-regulated (marked with red) and down-regulated genes (marked with blue) in Col-0 after CPT treatment. Two biological replicates are shown.

3. In figure S2 the authors have carried out R-loop DRIP-seq, however this experiment also does not have RNase H control. The authors show the track for 2 replicas. However, the authors really need to show the input lane for these tracks, to demonstrate the level of enrichment.

Thanks for the suggestions. Since we conducted RNaseH treatment and input sequencing of WT as experiment control, we did not include these tracks in the original version. In revision, we provided the tracks for RNase H control of Col-0 and input of Col-0 and *atm* (Fig S2 and S8).

Here we see that CPT treatment reduced R-loops (which will be in line with my point above suggesting a decrease of transcription). In general, I do not see any tracks suggesting R-loop increase following CTP treatment (in WT conditions), also the bioinformatics data does not show it, so I am not exactly sure why the authors claim this in the paper? This CTP-associated increased in seen in Fig 1C but not reproduced in Fig. 3E (in fact we see a decrease of R-loops) as observed in DRIP-seq experiments.

Thanks for the comments. The repeated slot blot results (Response Fig. 9A-B and Response Table 1) showed R-loop increased after CTP treatment both in WT and *atm*. And the metaplot results of ss-DRIP showed both the sense and antisense R-loops on protein-coding genes are increased in WT and *atm* after CPT treatment (Fig S2C). The newly added middle track of Fig. S2D showed the level of antisense R-loop in Col-0 was elevated after CPT treatment.

Similarly, R-loop decrease is observed in *asr20* mutant following CTP treatment. Is this also due to

transcriptional decrease?

As the slot blot results and ssDRIP-seq shown in Fig 3E, Response Fig. 9A-B, Fig5 A , Response Fig. 11 and Response Fig.12, the total R-loops were dramatically decreased in *asr20* following CTP treatment compared to *atm*. As we mentioned above (see the response to Major comments 2), the transcription could be changed, but the effects of transcription due to Top1 α are not the main causes of topological R-loop changes.

4. This paper contains multiple DRIP-seq genome-wide analysis and contain bioinformatic analysis (Fig. 5 and S2). However, none of the conclusions have been validated by DRIP-qPCR on specific genomic locations. This has to be incorporated to ascertain the strengths of the conclusions. Especially the finding that accumulation of R-loops in CTP conditions on replication origins needs DRIP-qPCR validation compared to non-origin genomic regions. It is also not clear to me how the origins were defined bioinformatically since this is not mentioned in the methods either.

Thanks for the suggestions. We performed DRIP-qPCR on selected three replication origins (related to Fig. S8A). As shown in Fig. S8B, the DRIP-qPCR results are consistent with the DRIP-seq data, indicating that CPT promoted R-loop accumulations on the origins in *atm*; which is relieved in *asr20*. In addition, we added the detailed definition of the DNA replication origins (Methods, line640-641).

5. The figure 6C (analysis of replication origins and R-loop shifts) was not clear to me and needs to be explained in a better way. I could not see how the authors concluded that there were shifts of R-loops in respect to the origin. This analysis also was not included in the Methods section and the statistical analysis is missing as well.

Thank you for this useful comment. We have supplemented more details in the paper and Figure legend (Line 247-253 and line 960-970).

6. I do not fully agree with the authors regarding their conclusions that HU relieved CPT effects. HU did not have any effect on root growth. Most probably (based on the slot blot result), HU is blocking transcription, and hence less R-loops are formed in the first place to cause any effect, and therefore combined treatment is less toxic. Did the authors check the level of transcription in HU cells?

Hydroxyurea (HU) is the inhibitor of the enzyme ribonucleotide reductase (RNR), which catalyzes the final reduction step in the production of deoxynucleotide triphosphate (dNTP), resulting in aberrant DNA replication. Therefore, HU has been widely used to stall DNA replication by depleting dNTP pools (PMID: 34356112, PMID: 27869662). When designing the HU treatment experiment, we screened the different concentrations of HU, and 2.5 mM HU treatment indeed delay the root development of all mutants (Response Fig. 13). 0.25 mM HU treatment conducted in the manuscript has a slight effect on replication that was not observed by root growth defects (HU treatment vs DMSO in Fig. S9). However, we indeed detected the differences in root growth under CPT conditions (HU+CPT treatment vs CPT in Fig. S9), suggesting that this 0.25 mM HU-induced effect on replication could partially restore root growth under CPT conditions, so we concluded that HU relieved CPT effects.

As previously mentioned in Major comments 2, the transcription inhibitor FLV treatment has similar

effects on root growth of wt (Response Fig. 11) and mutants and transcription has very little to no effect on R-loop formation on analyzed genes (Response Fig. 12). Combining the results that HU treatment decreases R-loop signals (Fig. S9D), we are convinced the DNA replication stalling and R-loops limitations by HU relieved CPT stress rather than blocking transcription.

Response Figure 13. WT and mutants showed comparable sensitivity to replication inhibitor HU. (A) Root growth phenotypes after treatment with 2.5 mM replication inhibitor hydroxyurea (HU). Scale bar: 1 cm. **(B)** The quantitative data of root lengths for Response Fig. 13A. Lines represent mean \pm SD, n=22, NS: no significant (one-way ANOVA).

7. The paper proposes some ideas on how the regulation of topoR-loops may happen, however I would wish the authors have tested their ideas a bit more extensively. So far the paper relies on genetic screens and DRIP-seq analysis of cell populations. The authors need to provide some clear validations of their data on single gene loci to support the validity of their models. Some parts of the paper are made unnecessary complicated. In my opinion, the paper needs to be re-written in a more focused way.

Thanks for the suggestions. As previously mentioned in major comments 4, we added the DRIP-qPCR results on the replication origins (related to Fig. S8A) to validate the DRIP-seq analysis (Fig. S8B). As suggested, in the revision, we have organized the text in a more focused way.

Minor comments:

1. The paper contains 11 main and 11 Supplementary figures, which is excessive. They can be re-organized into less figures, especially for the main figures.

Thanks for your kind suggestions. We deleted some data and added new results in this revision. The results we showed are important to the conclusions and claims of the paper. As mentioned in the beginning of replying your comments, our study is step-by-step dissection the topological stress, R-loops and genome stability, and each figure aims to explain one questions. Thus, we think it is necessary to organize the figures in current way.

2. The paper would benefit from substantial English editing. Some sentences do not make a lot of sense. Some factual scientific meaning is obscured/mis-represented in multiple places (for example, 2nd introduction paragraph: Third sentence ‘ RNA and DNA etc...’ should have a stop at the end. The next sentence ‘ In addition’ Contains awkward grammar ‘in where exonucleases XPG and XPF....’ – this should be corrected).

Fig 4 legend – Pol 2 is conserved

Thanks for the suggestions. We corrected these errors and carefully revised and polished the paper.

3. The detail of the activity assay in Fig S3 is missing. There should be an indication of the amounts of substrate (and its nature) and recombinant protein added. The mutant enzyme contains a big band at ~40 kDa which may affect its activity in vitro irrespective of catalytic mutation.

Thanks for the comments. We had checked the original data of SDS-PAGE, and found that there is also a band at ~40 kDa in GST-AtRNH1A sample by adjusting the contrast ratio. The band is not shown in GST-AtRNH1A sample might be caused by the different loading amount. We quantified the target band and found that the loading amount of GST-AtRNH1A^{D160N} was 1.5 times of GST-AtRNH1A (Response Fig. 14A).

Additionally, we repeated this assay by other person in the lab, and provided the gel staining picture. As shown in Response Fig. 14B, the non-specific bands at ~40 kDa are present both in wt and mutant protein after affinity purification, and, if this ~40 kDa nonspecific protein have any activity effect in vitro, the effects should be comparable in both wt and mutant proteins (Response Fig. 14B-C). We repeated this RNase H activity detection and also provided the amounts of substrates and proteins in the figure legends. We obtained similar results as the previous version that AtRNH1A has in vitro RNase H activity and D160N mutation could disturb this activity.

[redacted]

Response Figure 14. RNase H activity detection of wide type and mutant AtRNH1A in vitro. (A) The adjusted Fig S3A. Numbers below indicates the amount of proteins. (B) The repeated results of GST-AtRNH1A and GST-AtRNH1A^{D160N} proteins purification related to Fig S3A. Red triangle indicates the proteins. (C) EMSA results of purified AtRNH1A and commercial RNase H incubating with synthetic RNA:DNA hybrids after 5 and 30 minutes. 5 pmol GST-AtRNH1A and GST-AtRNH1AD160N and 5 U commercial RNase H was used. (unpublished data from Dr. Yushun Zhang)

4. Overall paper contains very brief methodology, making it impossible for anyone to follow or reproduce any of the experiments. The details need to be expanded. There is no information on DRIP-seq, bioinformatics analysis or suppressor screen to identify ars20 (how long was the drug treatment?). Supplementary information can be also used for that.

Thanks for the suggestions. We had expanded the main text and legends, and added more details of experiments and analysis in the Methods.

Reviewer #3 (Remarks to the Author):

In their work entitled « DNA polymerase ϵ strengthens topological R-loops-promoted genome instability in Arabidopsis », Li and colleagues investigated the hypersensitivity of the *atm* mutant to the Topoisomerase I inhibitor camptothecin (CPT), finding that CPT induces R-loops accumulation leading to the formation of DSBs. They identify a mutation in the replicative polymerase DNA pol ϵ as a suppressor of *atm*'s hypersensitivity to CPT, hinting at transcription/replication conflicts as the primary trigger for CPT-induced DSB formation.

This manuscript contains a considerable amount of work and data, and the findings are very exciting. I am convinced this work has the potential to bring very important knowledge that will be of interest for a broad community. However, in its present form it remains preliminary, both because some conclusions are not sufficiently supported by the data or sometimes even in complete contradiction with what is shown on figures, and because it is in places not sufficiently clearly written. Below are listed a number of suggestions and comments for the authors that I think should be addressed

Results

1. I have concerns about data related to gamma H2AX accumulation. Authors write “The γ H2AX signals in Col-0 and *atm* were increased by CPT treatment, while *atm* impeded γ H2AX levels in the control » but the western blot quantification does not show any increase. If anything, the signal is slightly lower in *atm* treated with DMSO than in all other samples. Regarding the immuno-staining, authors chose to quantify global fluorescence instead of counting foci, or nuclei with foci. Looking at the Figures, I understand why they did that, because foci are not clearly visible, but that raises questions about the specificity of their antibody. They can check the following references doi: [10.15252/embj.201694571](https://doi.org/10.15252/embj.201694571), doi: [10.1093/plcell/koab158](https://doi.org/10.1093/plcell/koab158) or doi: [10.1105/tpc.15.00898](https://doi.org/10.1105/tpc.15.00898) for pictures of specific gamma H2AX labelling and appropriate quantification methods. This comment applies to Figures 1, 3, 8 and S4

Thanks for the comments. When we were generating this γ H2AX antibody, the irradiated seedlings, which are reported to significantly increase the γ H2AX signals in previous work (PMID: 15772150, PMID: 12509526), were used as positive controls for antibody validation. As shown in Response Fig. 15A, the irradiated samples have significant enrichment of γ H2AX band at ~17 kDa compared to non-irradiated samples, which was not observed in *atm*, indicating the γ H2AX antibody we used has high specificity.

We repeated the immunolabelling experiments using microscopes (Zeiss LSM880 with Airyscan) with higher resolution and the foci are clearly visible (Response Fig. 15B), suggesting γ H2AX antibody is specific and the results in Fig. 1E and Fig. 3G are reliable.

In addition, Figures 8 and S4 are used to calculate the numbers of γ H2AX-labeled nuclei, the resolution are lower than that of Figures 1 and 3.

Response Figure 15. The γ H2AX antibody specificity detection. (A) The western blot of γ H2AX at different time points after 100 Gy ionizing radiation (IR) treatment in Col-0 and *atm*. H3 was used as loading control. -: no treatment. (unpublished data from Dr. Yushun Zhang) **(B)** Immunolabeling of root tip nuclei using R-loop antibody S9.6 and DSB marker γ H2AX in Col-0, *atm* and *asr20* with or without CPT treatment. The nuclei are viewed and imaged using Zeiss LSM880 with Airyscan. Bar: 1 μ m.

2. I think there is something I am missing in the way authors generated the metaplots shown throughout the manuscript to illustrate the overall distribution of R-loops on genes. For example, it is clear from the PCA shown on Figure S2 that the atm CPT sample is different from all other samples. Authors write in the text that « Results showed slightly elevated R-loops in atm relative to wild-type (...) and [R-loops] increased dramatically in atm upon CPT treatment ». This is clear from the screenshot shown on Figure S2D. Yet, on the metaplots, only the atm DMSO sample seems to differ from the others and to display lower levels of R-loops. This looks like a normalization issue and should be solved. This comment also applies to Figure 5. The question here is really crucial, because if the metaplots were to be trusted, on Figure 5 I would conclude that the pol2a mutation isolated in the screen leads to a massive decrease in R-loop accumulation compared to the wild-type even under control conditions, and that this is true not only near the TTS, where authors think it reflects TRC, but also on the opposite strand near the TSS. By the way, the biological meaning of these TSS R-loops is never described or discussed in the text, and I think this is really missing for readers who would not be familiar with R-loops and their roles in the cell.

The scale-regions mode of computeMatirx from deepTools was used to calculate the score per genome region. The genome regions used in Fig5 and S2 are protein-coding genes that exclude non-coding regions such as TE, rRNA, tRNA and intergenic regions. Distance upstream of the TSS and distance downstream of the regions is 1000 bp. The PCA results reflect the changes at the overall level on the entire genome, this is the reason why the PCA results are consistent with slot blot (Fig. 1C). Our FigS2B scatterplot results showed that R-loop levels significantly increased on a large number of protein-coding genes, this promoted us to analyze the R-loop dynamics on protein-coding genes, the metaplot results shown in Fig5 and S2 represent the changes of R-loops on total protein-coding genes but not just the genes with up-regulated R-loops. The average scores could be diminished by genes with decreasing R-loop and have insignificant differences.

According to Ref 24, “Top1 inhibition causes fork pausing at the TTS of highly expressed genes containing R-loops leads to head-on conflicts between replication and transcription”, so we think the R-loops changes near TTS reflects TRC. The biogenesis and functions of antisense R-loops near TSS are not clear at the moment, we are trying to figure out the functions of the R-loops on TSS in other projects in the lab. In order to focus on the topological stress and R-loops in genome stability, and to reduce unnecessary confusion, we have not put effort in discussing the R-loops on TSS.

3. A lot of information is missing from methods and figure legends, and that makes the manuscript really difficult to read. For example, by comparing Figures 1 and 2, it is obvious that either the dose of CPT, or time of treatment, or both are not the same: on Fig1, the atm mutant shows a completely disorganized root tip, whereas on Fig2 it is not so affected, but I could not find the information anywhere.

The dose of CPT used in Fig 1 and 2 are the same, but the time of treatment are different. We have supplemented the information in the methods (Plant materials and growth conditions). and figure legend (Fig2, Fig3, Fig9, Fig10, Fig S1, Fig S6, Fig S7, Fig S9, Fig S11). As we mentioned in the methods, 30 nM CPT was used for all the CPT sensitivity assays, except the specially marked in the Fig S7, but the time of treatment varies and we have supplied the information in the Figure legend. For Fig 2, we firstly transferred the plants to 1/2 MS medium containing estrogen after germination (~3day). After 36 hours of estrogen induced AtRNH1A overexpression, the plants were transferred

to 1/2 MS medium containing 30 nM CPT and estrogen for 10 days.

4. I did not manage to understand what I was supposed to see on Figure 6C. I think this is possibly because the legend is not sufficiently detailed.

We added the detailed interpretation in the text and figure legends of Fig 6C. Fig 6C could not only show the changes of R-loop levels, but also the changes of R-loop peaks position. We found that the level of R-loops in *atm* not only increased significantly, and the position of R-loops was closer to DNA origins after CPT treatment, while the position of R-loop in WT and *pol2a*^{L473F} did not change despite the upregulation of R-loops, such upregulation and position movement of R-loops did not occur in *asr20*.

5. I fully agree with authors that the effect of the *pol2A* mutation on R-loop formation could reflect a change in replication speed, and that attempting to measure it would provide very valuable information. I have strong concerns about the way the experiment was performed and what authors conclude from it. First, I do not quite follow how EdU labelling can continue to increase if the molecule is washed away after 30 min, but maybe there is still enough of it inside the cells to allow continued incorporation. However, I am not convinced by the quality of the data. The flow-cytometry graphs show extremely poorly-defined groups of nuclei, which is likely to prevent robust quantification. Besides, these data cannot easily be translated into replication speed. It is well documented that *pol epsilon* mutants show a higher proportion of S-phase cells when EdU incorporation is performed for a relatively short time, but this increase results from a dramatic lengthening of S-phase, the consequence of which is that S-phase represents a larger proportion of the total cell cycle length, and that at any given time point, a larger proportion of the cells are in S-phase (doi:10.1104/pp.17.00031). Performed in this way, the experiment cannot be conclusive in the 78way authors would like to use it. Even if their conclusions were more robust, the increase in the proportion of EdU labelled nuclei would reflect a combination of things (replication speed + speed of cell cycle progression). To conclude on replication speed, they should perform DNA combing or spreading to prove replication fork speed is modified in their mutant.

Thanks for the suggestions. The EdU was added during the 4 h recovery period, we just wash out the DMSO or CPT. We modified and added details in figure legend and methods. As we mentioned above (please see the response to Major comments 1 of Reviewer #1), we tried hard to clarify the DNA replication fork speed by DNA spreading. It is a pity that we could not set up the DNA fiber assay in our system. Alternatively, we adopted the double-labeling strategy of EdU and BrdU to interpret a more detailed replication progression in vivo (Fig. 7). We suggest *pol2a* stall the DNA replication and S phase progression in normal growth conditions, but interrupt the DNA replication and S phase progression stalling under consistent topological stress. Moreover, we performed DNA polymerase activity experiments in vitro, the results showed that POL2A mutation slightly delayed DNA replication speed (Fig. S12). As you concern about the flow-cytometry graphs, the border of the S-phase and endo-replication is illegible, and the criteria for defining the S-phase nuclei are the same, it can reflect the S-phase procession. And for more rigorous consideration, we decided to delete the EdU flow-cytometry result, since the double-labeling results and in vitro experiments are consistent with it.

6. In relation with the above-mentioned point, I think the novel pol epsilon allele isolated here is insufficiently characterized. In terms of root growth, it seems less affected than *abo4-1* on control conditions, but a more detailed phenotypic description would be extremely important to decipher to what extent the L473F mutation has unique consequences on the replication process compared to other point mutations affecting pol epsilon.

Thanks for the suggestions. As we mentioned above (response to Minor comments 2 of Reviewer #1), we provided the phenotypic pictures of mutants. As shown in Response Fig. 6, *pol2a^{L473F}*, *till-4* and *abo4-1* had similar developmental phenotypes of early flowering, with *abo4-1* flowering earlier than other two. All three mutants cause partial sterility and developmental defects of siliques that are further enhanced by *atm*. The results in Fig. S7 and Fig. S13 showed *pol2a^{L473F}* and *abo4-1* had similar but different degrees of sensitivity to CPT and MMS, suggesting L473F and G469R mutations had similar but different effects than G522N on mitosis during vegetative growth.

7. On Figure 9, authors show that *asr20* accumulates ribonucleotides in its DNA. However, they introduce the *rnh2a* mutation in the *abo4-1* background rather than in *asr20* or *pol2a^{L473F}*. Why is that? A mutation in the *rnh2* gene has been isolated as a suppressor of the hypersensitivity of the *wee1* mutant to hydroxyurea (doi: 10.1080/15592324.2014.1001226). How does this fit with the authors results?

In the origin version, we did not obtain *atrnh2a/pol2a^{L473F}* mutant yet. But in the revision, we generated the *atrnh2a/pol2a^{L473F}* double mutants and compare the root length among related mutants (Fig. 9), and the *atrnh2a/pol2a^{L473F}* showed a similar phenotype to *atrnh2a/abo4-1* (Fig. S11).

RNase H2 recognizes and cuts rNTPs and is the only enzyme known to hydrolyze misincorporated ribonucleotides during genomic replication. RNase H2 complex rescued hypersensitivity of the *WEE1*-deficient plants to HU through overcoming replication stress which correlates with increased incorporation of rNTPs in DNA. This substitution of dNTPs with rNTPs restored the replication kinetics of the *wee1* but resulted in replication errors. As shown in Fig. 9C-D and Fig. S11C-D, the root length of *atrnh2a/pol2a^{L473F}* or *atrnh2a/abo4-1* is shorter than *atrnh2a* and *pol2a^{L473F}* or *abo4-1* single mutant, *asr20* or *atm/abo4-1* accumulates much more rNTPs in genomics under CPT treatment, and *asr20* or *atm/abo4-1* are more sensitive to CPT than related single *pol2a* mutant. Consequently, these results suggest *pol2a* regulate DNA replication and increase incorporation of rNTPs to overcome *atm* inhibition of TOP1i which like the function of *RNase H2* mutation mentioned above. Moreover, in RER-defective yeast, Top1 specifically incises ribonucleotides in the nascent leading strand (PMID: 25751426), indicating TOP1 inhibition promotes rNTP accumulations in *pol2a* which rely on the repair of RNase H2 and ATM also functions in this DNA replication stress. We had modified the discussion according to your comments.

8. When describing Figure 10, authors write “*rad52* and *ku70* presented CPT tolerance and partially restored the short root phenotype of *atm* after CPT treatment ». This is simply not what the figure shows. The single mutants behave like the wild-type, so they cannot be called tolerant, and although I agree the *rad 52 atm* mutant grows better than *atm*, this is clearly not the case for *atm ku70*, if the difference is really significant as suggested by the stars above the plots (although the difference is extremely slight), the double mutant does worse than *atm*, which would make sense if NHEJ is required to repair DSBs caused by R-loop accumulation.

Thanks for your comments. We have modified the presentation “tolerance” in the revised text (Line 389-390). We had double-checked the statistical data, and found the difference between *atm* and *atm/ku70* with CPT treatment is not significant. We are sorry for the mistakes that labeled in the first version, and we had corrected the statistical chart (Fig. 10F). Thank you for pointing out this issue. According to Fig 10 E-F, the root growth of *atm/ku70* is not further inhibited by CPT. Based on these, we claim that NHEJ play major role in DNA damage repair under normal conditions, both NHEJ and HR are required for topo-R-loops induced DNA damage repair.

9. I am not sure the experiment with MMS treatment is really useful. This drug has extremely complex effects as it does block replication fork progression, but also causes the formation of DNA breaks. It is not clear to me what hypothesis authors wanted to test with this experiment, but I would suggest a more specific treatment, depending on the point they are trying to make.

Since MMS leads to both DNA single strand breaks and DNA replication stress, we use MMS to investigate whether *pol2a* functions not only in DNA replications but also DNA breaks repair. *asr20* and *atm* showed similar sensitivity to MMS, while *pol2a^{L473F}* and WT showed comparable resistance to MMS, suggesting POL2A play roles in both DNA replication processes and also DNA damages.

10. I am surprised there is no access given to reviewers to the deep-sequencing data for the reviewing process

In the origin submission, we provided the access to the deep-sequencing data in the cover letter. Our data has already submitted to GEO database, and no. is GSE21617 and “cdcfusuuxpoptyl” the password for reviewers. We had also uploaded the bigwig files onto UCSC, use the link https://genome.ucsc.edu/s/yujw/2022.11.09.liqin_to browse the peaks.

Discussion

Given the considerable amount of data generated, I think the discussion is really incomplete, and does not do justice to the work in its present form. In the specific points I listed for the results, I mention a number of references that authors could/should have cited to put their work in the global context of the plant DNA damage response, as their results can really bring novel and valuable information for this field. Also, it is apparent that they have written this section a bit rapidly. For example, they wrote “Importantly, the biological requirements for ATM and ATR are particularly different, as ATR, but not ATM, is essential for embryogenesis and viability. While *atr* shows no visible growth differences from wildtype Col-0, *atm* is partially sterile » This makes no sense if you don’t specify that ATR is essential IN MAMMALS, but shows no growth difference with the wild-type IN ARABIDOPSIS (which by the way is not quite true in maize or barley).

Thanks for the helpful suggestions. We rewrote the discussion in a focused way and added more specific information about ATM and POL2A relating to R-loops, DNA damage repair and DNA replications. We deleted the section about ATR, since the functions of ATR in topoR-loops and DNA damages were not studied in our paper.

Overall, I think the working model and conclusions are still not clear. Authors write that *rnh2* suppresses the resistance of *pol epsilon* mutant to CPT, but these mutants are not resistant to CPT,

what the mutation does is that it makes them sensitive. How this happens since the levels of ribonucleotides in the DNA are not measured in the *abo4* mutant is not clear. Authors should also clarify how they think this incorporation happens. I am not sure whether they mean they are incorporated by Pol epsilon itself, or whether they consider TLS polymerases are recruited to resolve stalled forks and are responsible for this phenomenon.

Since *atrnh2a*, *pol2a*^{L473F} and *abo4-1* single mutant are not sensitive to CPT (Fig. 9C-D and Fig. S11C-D), and even *pol2a*^{L473F} and *abo4-1* are strongly resistant to high levels of CPT (Fig. S7), but the root length of *atrnh2a/pol2a*^{L473F} and *atrnh2a/abo4-1* double mutants are shorter than single mutants after CPT treatment (Fig. 9C-D and Fig. S11C-D), we conclude *atrnh2a* abolished the resistance of *pol2a* to TOP1i. The rNTP insertion levels are increased both in *asr20* and *atm/abo4-1* with CPT treatment (Fig. 9A-B and Fig. S11A-B), combining the results that the speed of DNA replication in *asr20* are accelerated in CPT condition (Fig. 7), we suggest *pol2a* misincorporated more rNTPs, but also it's possible the translesion synthesis DNA polymerases are recruited to resolve stalled forks and introduce more errors which needs more evidence to prove.

More generally, I am not sure of the model shown on Figure 11 and title of the paper. As I mentioned earlier, I am not at all convinced that pol epsilon mutants show uncontrolled replication, to me, previous knowledge rather points to the opposite. It is indispensable to clarify this point by performing appropriate experiments, or to rethink the model taking into accounts alternative hypotheses. Likewise, I am not convinced that DNA polymerase epsilon "strengthens R-loops". Based on Figure 6A, I would say that R-loops are dramatically increased at replication origins in *pol2a* mutants, possibly due to fork stalling that hampers the resolution of these R-loops. This effect is less pronounced, but still visible in the *atm* background, so the authors' conclusions are really not in agreement with the data.

Thanks for the suggestions. We modified the title and figure legend of Fig11, and supplemented more information in the discussion. As shown in Figure 7 and Figure S12, we concluded *pol2a* delay DNA replication in vitro and in vivo under normal condition, and promote DNA replication in vivo under topology stress. In Figure 6A, the R-loops are increased at replication origins in *pol2a*^{L473F} mutants, but globally decreased on protein-coding genes (Fig. 5A). Besides, the R-loops are significantly decreased in *asr20* than in *atm*, so we changed the statement that DNA polymerase epsilon regulates R-loops but not strengthens. As you suggested, it's possible that fork stalling could hamper the resolution of R-loops, we added the hypothesis in the discussion. Additionally, as we discussed in the revision, POL2A play important role in cell cycle regulation, chromosome recombination, transcriptional gene silencing (TGS), and epigenetic regulation, the R-loop dynamics of *asr20* and *pol2a* could be regulated comprehensively, which requires further research and also provides research direction for the regulation between DNA replication and R-loops.

I am sorry I cannot give a more positive evaluation of this work at this stage, and there are many more examples of places in the manuscript where the conclusions are either not clear, or not quite in agreement with what is shown in the Figures. I am convinced however that if authors address all these comments, and carefully polish the writing giving more details in the text, in the legend of figures and discussing their work more in depth, their work has the potential to make an extremely good paper.

Thanks for these insightful suggestions, which helps us to improve the overall quality of the

manuscript. We have modified and polished the manuscript as much as possible, and we hope this could meet with your comments.

REVIEWER COMMENTS

Reviewer #1 (Remarks to the Author):

The authors have addressed most of my concerns and significantly improved the manuscript. I basically encourage the publication of this manuscript. However, due to the authors cannot apply the assays like the DNA fibre experiment to address my previous first major concern, the authors have to be very careful to make any conclusions or assumptions in the discussion part. Here is my remaining major concern for what the authors mentioned in the discussion and model. Line 435-437, "We propose that pol2a might promote replication fork progression, therefore, to decrease R-loops under topological stress (Fig.11)." I am confused that how a POL ϵ mutant (the mutations not in exonuclease domain) can promote fork speed, can authors give me a possibility. Secondly, back to my previous first major concern, the authors did some experiments to prove that pol2a mutants compromise POL ϵ function and DNA synthesis speed. So I don't understand why the authors can make this assumption. Obviously, this concern is relevant to Fig.11 (model figure). I recommend the authors should be more careful about this point to prevent overstating any conclusions without evidence. Another minor point is in line 443. So far as I know, the POLE3 and POLE4 subunits of POL ϵ are not essential to support cell proliferation, although POLE1 and POLE2 are essential. The final manuscript must cover the concerns above.

Reviewer #2 (Remarks to the Author):

IN RELATION TO MAJOR COMMENT 1:

Unfortunately, the authors failed to do a proper job here. The figures have not changed in the paper, since the RNase H controls were not added to the main and supplementary figures (they have been shown only in the Response figures for the reviewers). The response Table S1 does not provide any statistical analysis (i.e. are the enrichments claimed in the paper statistically significant)? All the replicates need to be analyzed statistically and p values are represented on the graphs and in the paper figures next to slot blots, as originally requested (this is a requirement of Nature publishing journals). This type of information is a 'must' for any paper. The claims made in the paper should be only made if they are supported by a statistically significant data, which is currently not the case for any of the slot blot analysis in the paper.

IN RELATION TO MAJOR COMMENT 2:

The data showing discrepancy between RNA-seq and DRIP-seq is important to include in the paper (response Fig.12), which strengthens the point that the effects of CTP on R-loops are not just due to changes in transcription.

IN RELATION TO MAJOR COMMENT 3:

Slot blot in Fig 3E does not show an increase R-loop following CPT treatment (in WT situation), as compared to slot blot in Fig 1E. The authors need to provide the quantification of the slot blot data with statistic and p values to show the significance of these data (as already requested in my Comment 1 above). This should be presented as a part of this paper.

Furthermore, the DRIP-seq tracks presented do not show increased level of R-loops following CPT treatment – I do not see it, and the authors need to highlight to show where it is and support it by DRIP-qPCR validation for the same gene locus highlighting the significance of these data.

Based on these two evidences, the conclusions that authors see the increase of R-loops following CTP treatment in WT situation is unjustified in this experimental model system.

IN RELATION TO MAJOR COMMENT 4:

The authors need to present which data points where compared for the statistics on the figure S8A. I would like to see the non-replication origin regions as controls, as originally requested, to demonstrate

the specificity of such phenotypes over replication origins.

IN RELATION TO MAJOR COMMENT 6:

Since the authors are convinced that transcription plays very little role in the R-loop phenotypes, do they have any data to support their conclusions that HU acts through DNA replication stalling – i.e. have they measured the rate of DNA replication in these cells to support such conclusion? It is clear from their data that HU has similar effects on the root growth in WT and mutant conditions (just like transcriptional inhibitors)?

Minor comments:

In relation to my previous minor comment 1:

I still think that the number of main figures (11) and 13 supplementary is excessive and it could have been reorganized in a more sensible style.

In relation to my previous minor comment 3: The nature of the RNA/DNA hybrid substrate (length) is still not stated.

Reviewer #3 (Remarks to the Author):

Authors have included a number of new experiments to try and address my comments. Notably, regarding my third point, they have considerably improved the description of the experiments in the methods section and in Figure legends. However, a number of points remain unresolved. The description of the data is still inaccurate or simply contradicting the provided data in a number of places, the interpretation of some experiments is in my opinion questionable, and the english still requires significant improvement.

1- Authors have not convincingly addressed my first point. They do show clear data about the specificity of their antibody. However, they missed the most important part of my remark, that was to point that their WB quantification does not show a significant increase in gamma H2AX accumulation contrarily to what they write. A change from 0.19 to 0.20 in the wild-type can hardly be considered as statistically relevant. There is no statistical test in Response table S1, but if authors were to run one, I doubt they would conclude changes are statistically relevant. To me, it is a major ethical issue to write something in the text that is not supported by the data shown in the Figure. This behaviour shades doubts on all results. Besides, the WB and IF results on gamma H2AX are not consistent with each other. I have the exact same comment about the quantification of gamma H2AX shown on Figure 3.

2- I understand the response made by authors about Figure S2C, ie that metaplots do not show a difference in R-loop levels between atm and Col0 because they averaged all protein-coding genes regardless of whether there was an increase in R-loops or not. However, I think this way of generating the figure is not very informative (besides it still does not explain the global decrease in R-loops that systematically appears in atm DMSO samples). It would be more informative to generate 2 sets of metaplots : one on genes that show more R-loops in atm CPT than in Col0, and one on genes that show less. This would allow appreciating the amplitude of the changes in the 2 cases. If authors are correct in writing that atm has fewer R-loops than Col0 under control conditions, it would be nice to see a scatter plot like the one shown in Figure S2B to confirm this, and also again, to see if the sites

at which R-loop formation is modified are the same, or different, from the ones that vary in response to CPT.

Also it would be interesting to know if the R-loops that increase in atm CPT vs Col0 CPT are the same as the ones that increase in Col0 CPT vs Col0 DMSO.

In the same line as the remarks on Figure S2, on Figure 5, it would be nice to confirm the global decrease in R-loops in asr20 mutants by generating a scatter plot in which they are compared to Col0, to double check that results are consistent.

3- The additions to the legend of Figure 6 do help, however, I still cannot see what authors describe. For example, why are 2 peaks of R-loops drawn on all pannels except for asr20 ? Globally, all plots seem very similar to me except I see much fewer lines in asr20 and pol2aL473F than in Col0 or atm (which does not match the clustering analysis shown on panel A). If anything, I see a reduction in signal intensity in atm + CPT vs atm DMSO, I certainly cannot make out the shifts of positions depicted on the cartoons (and authors forgot to specify if green is DMSO and orange CPT or the other way around).

4- The new Figure 7 is probably a better way of assessing the effect of the pol2a mutation on S-phase progression, but its interpretation is complex because it is the result of control of the G1/S transition and the S-phase progression. I wonder whether it wouldn't be easier to focus only on EdU + nuclei and to count the proportion of BrdU+ nuclei among those in each experimental set up. By doing this, authors are sure to focus on cells that are in S at the beginning of the experiment, and to test whether they have reached G2 or not by the time they perform the BrdU pulse. Dual labelling means they are still in S at the time of the BrdU pulse. The longer cells spend into S-phase, the higher this proportion should be after a long chase interval. If data are looked at this way, the conclusions of the experiment may change. Indeed, atm and asr20 have much more dually labelled cells than Col0 in the presence of CPT, which would indicate they lag into S longer. It is true however, that asr20 mutants seem to progress through S-phase faster than atm, but all this is only a visual impression of course because I don't have the raw numbers. Analyzing the data in this manner would probably make the text accompanying the Figure easier to read. One point that I cannot understand is the behaviour of BrdU-only nuclei. Why are they completely absent in Col-0 + CPT after a 2h thymidine chase ? This would suggest entry into S-phase is completely blocked by this treatment, which complexifies the picture. Besides, this lacks replicates and statistical analysis.

5- The experiment on Figure S12 lacks quantification, I cannot see any difference on the gels, contrarily to what authors describe in the text.

6- Regarding the experiment with rad52 and ku70, although authors claim they have corrected the text, they still write « both atm/rad52 and atm/ku70 exhibited CPT resistance that the roots were not severely inhibited as atm » this is not true, only atm rad52 perform better than atm on CPT. These repeated inaccurate statements in the text are really worrying.

7- There must be a mistake in the methods paragraph describing statistical analysis. Most experiments compare multiple genotypes, which requires an ANOVA for sure, but followed by a post-hoc test such as Tukey. This is not indicated.

8- The discussion has been re-written, but the working model still needs to be better supported by the data (notably the hypothesis that replication speed is modified in different genetic contexts).

First of all, we sincerely thank all the reviewers for their perceptive and constructive comments, which give us the chance to further improve the quality of our research. In this revision, we addressed all the issues mentioned, and answered all the questions. We have reorganized and modified the original figures and text accordingly, and this revised version contains 9 main figures and 13 supplementary figures.

Reviewer #1 (Remarks to the Author):

The authors have addressed most of my concerns and significantly improved the manuscript. I basically encourage the publication of this manuscript. However, due to the authors cannot apply the assays like the DNA fibre experiment to address my previous first major concern, the authors have to be very careful to make any conclusions or assumptions in the discussion part. Here is my remaining major concern for what the authors mentioned in the discussion and model. Line 435-437, “We propose that pol2a might promote replication fork progression, therefore, to decrease R-loops under topological stress (Fig.11).” I am confused that how a POL ϵ mutant (the mutations not in exonuclease domain) can promote fork speed, can authors give me a possibility. Secondly, back to my previous first major concern, the authors did some experiments to prove that pol2a mutants compromise POL ϵ function and DNA synthesis speed. So I don't understand why the authors can make this assumption. Obviously, this concern is relevant to Fig.11 (model figure). I recommend the authors should be more careful about this point to prevent overstating any conclusions without evidence. Another minor point is in line 443. So far as I know, the POLE3 and POLE4 subunits of POL ϵ are not essential to support cell proliferation, although POLE1 and POLE2 are essential. The final manuscript must cover the concerns above.

Thank you for these constructive comments and suggestions. Figure 5 and S10 of this revised version showed that *pol2a*^{L473F} and *asr20* delays the DNA replication and cell cycle under control condition (DMSO). However, DNA replication and cell cycle are inhibited by TOP1i in WT but not in *asr20* (Figure 5 in this version). Moreover, the R-loop levels in *asr20* remained unchanged at a low level after CPT treatment (Figure 4D in this version). As discussed in line 431-440, we proposed that *asr20* promotes DNA replication and therefore decreases R-loop accumulation under topological stress, even though the DNA replication and cell cycle of *asr20* and *pol2a*^{L473F} are delayed in normal condition. As we discussed in line 441-458, the POL ϵ works as a complex and POL2A interacts with many other factors in vivo, so the mutation site may not change the DNA replication speed directly, but its interaction partners might change the noncanonical and unclarified role beyond replication, which could be the reason why *pol2a*^{L473F} shows delayed DNA replication and cell cycle in control condition (DMSO), but presents faster S-phase progression with *atm* mutation and CPT treatment. In addition, this might be the reason why rNTPs incorporation errors could only be detected in *asr20* after CPT treatment (Figure 7 and S11 of this version).

As you mentioned, and reviewer 3# also suggested that the model figure should be more careful to prevent overstating conclusions especially about DNA replication speed, we have removed the

replication speed and modified the model figure (Figure 9 in this version) to avoid obscure conclusions and the corresponding description (legend in Figure 9).

As you commented that the POLE1(POL2A) and POLE2(Dpb2) are essential for cell survival, POLE3(Dpb3) and POLE4 (Dpb4) subunits of POL ϵ are involved in DNA binding and are required for replication fork progression (PMID: 22918589). Moreover, the C-terminal portion but not the N-terminal of POL2A is essential for cell viability, suggesting that the higher-order protein interactions between Pol ϵ and other cellular components are required for cellular viability (PMID: 22918589). This is in consistent with our hypothesis that discussed above. We think that the interactions between C-terminal domain of POL2A and other subunits of DNA polymerase ϵ are essential for cellular functions. To avoid any misleading, we modified the sentences as “For instance, it is reported that the C-terminal portion of POL2A could interact with POLE2, which is essential for cell proliferation (PMID: 10428796).”

Reviewer #2 (Remarks to the Author):

IN RELATION TO MAJOR COMMENT 1:

Unfortunately, the authors failed to do a proper job here. The figures have not changed in the paper, since the RNase H controls were not added to the main and supplementary figures (they have been shown only in the Response figures for the reviewers). The response Table S1 does not provide any statistical analysis (i.e. are the enrichments claimed in the paper statistically significant)? All the replicates need to be analyzed statistically and p values are represented on the graphs and in the paper figures next to slot blots, as originally requested (this is a requirement of Nature publishing journals). This type of information is a ‘must’ for any paper. The claims made in the paper should be only made if they are supported by a statistically significant data, which is currently not the case for any of the slot blot analysis in the paper.

Thanks for your constructive suggestions. We had added the slot blot results (Figure 1C, Figure 3E, Figure 8C, Figure S9D in this revised version) containing RNase H control (from the First Revision Response Figure 9 responding to the reviewers) in this revised version, and provided the quantification values and the p values on the statistical histogram.

IN RELATION TO MAJOR COMMENT 2:

The data showing discrepancy between RNA-seq and DRIP-seq is important to include in the paper (response Fig.12), which strengthens the point that the effects of CTP on R-loops are not just due to changes in transcription.

Thanks for the suggestions. We had added these results (response Figure 12) in the Figure S3 of this revised version.

IN RELATION TO MAJOR COMMENT 3:

Slot blot in Fig 3E does not show an increase R-loop following CPT treatment (in WT situation), as compared to slot blot in Fig 1E. The authors need to provide the quantification of the slot blot data with statistic and p values to show the significance of these data (as already requested in my Comment 1 above). This should be presented as a part of this paper.

Furthermore, the DRIP-seq tracks presented do not show increased level of R-loops following CPT treatment – I do not see it, and the authors need to highlight to show where it is and support it by DRIP-qPCR validation for the same gene locus highlighting the significance of these data.

Based on these two evidences, the conclusions that authors see the increase of R-loops following CTP treatment in WT situation is unjustified in this experimental model system.

Thanks for the suggestions. As previously mentioned in relation to major comments 1, we added the slot blot results containing RNase H control in Figure 1C and Figure 3E and provided the statistical data. Based on these results, the patterns of R-loop increase following CPT treatment in WT and *atm*. It is important to explain and emphasize here that, we find it is very difficult to combine signal calculations of different blotted membranes into one statistical graph. We had tried, but it is too rare to find the statistical results for slot or western blot replicates in the literature. Thus, we supplied the statistic results for each slot and/or blot instead.

In addition, we have analyzed the R-loops on total annotated elements (Figure S2B of this revised version), the metaplot result shows R-loops increased especially at TSS sites in Col-0 after CPT treatment. This increase trend is in line with the slot blot results. Besides, we have validated the R-loop increment after CPT by DRIP-qPCR (Figure S2E-F), and the DRIP-qPCR results are consistent with the DRIP-seq data.

IN RELATION TO MAJOR COMMENT 4:

The authors need to present which data points where compared for the statistics on the figure S8A. I would like to see the non-replication origin regions as controls, as originally requested, to demonstrate the specificity of such phenotypes over replication origins.

Thanks for the suggestions. We marked the location of DRIP-qPCR sites with light purple boxes (Figure S8B of this revised version). We supplemented the DRIP-qPCR results of non-replication origin nearby two selected origins (Figure S8B of this revised version), and the two non-replication origin sites are marked with light yellow boxes (Figure S8B of this revised version). The DRIP-qPCR results showed that higher R-loop signals were observed in non-replication origins of *asr20* compared to *atm* or Col-0 (Figure S8C of this revised version), and CPT slightly enhance the R-

loop accumulations in *asr20* at non-replication sites (Figure S8C of this revised version), suggesting the pattern of R-loop dynamics between DNA origins and non-origins is different in *asr20*.

IN RELATION TO MAJOR COMMENT 6:

Since the authors are convinced that transcription plays very little role in the R-loop phenotypes, do they have any data to support their conclusions that HU acts through DNA replication stalling – i.e. have they measured the rate of DNA replication in these cells to support such conclusion? It is clear from their data that HU has similar effects on the root growth in WT and mutant conditions (just like transcriptional inhibitors)?

Thanks for the suggestions. As we mentioned in major comment 6 in the first revision, HU, as the inhibitor of RNR, has been widely used to stall DNA replication by depleting dNTPs pools in Arabidopsis (PMID: 37354461, PMID: 37159556, PMID: 33450002). We applied EdU staining to detect the DNA replication progress with or without HU treatment, we observed that the efficiency of DNA replication inhibition in WT and mutants are similar (Response Figure 1).

Response Figure 1. EdU-labeling to detect DNA replication of root tips treated with replication inhibitor HU in WT and mutants. 5-day-old seedlings were firstly treated with 10 mM DMSO or HU for 1.5 h, then 10 μ M EdU was added for 30 min. The root tips were collected for EdU labeling. Scale bar: 100 μ m.

Minor comments:

In relation to my previous minor comment 1:

I still think that the number of main figures (11) and 13 supplementary is excessive and it could have been reorganized in a more sensible style.

Thanks for your kind suggestions. Combining your suggestions and those of reviewer #3, we appropriately removed some figures that had no impact on the main conclusion of the article (e.g. we removed the text and figures about strand-specific information of ssDRIP-seq, which is not essential for the conclusion; we had removed the Figure 6C of the previous version, because the Reviewer 3# is still confused by those graphs and we think the current results (Figure 4E-G) are enough and statistical to support the conclusion that POL2A could modulate R-loops by controlling DNA replication during the early replication steps). Also, we added some figures as suggested (Figure S2D and Figure S3 of this revised version). Finally, we reorganized the manuscript into 9 main figures and 13 supplementary figures.

In relation to my previous minor comment 3: The nature of the RNA/DNA hybrid substrate (length) is still not stated.

Thanks for the suggestions. We have provided the length of substrate in Figure S4 of this revised version, changed the RDH to RNA:DNA (37 bp) and ssRNA(37 nt), and supplemented the sequence of RNA/DNA hybrid substrate in Table S1.

Reviewer #3 (Remarks to the Author):

Authors have included a number of new experiments to try and address my comments. Notably, regarding my third point, they have considerably improved the description of the experiments in the methods section and in Figure legends. However, a number of points remain unresolved. The description of the data is still inaccurate or simply contradicting the provided data in a number of places, the interpretation of some experiments is in my opinion questionable, and the english still requires significant improvement.

Thanks for your constructive comments and suggestions. We had followed your suggestions to reorganize the main figures and text, and had thoroughly and meticulously checked and revised the description of the data and language.

1- Authors have not convincingly addressed my first point. They do show clear data about the specificity of their antibody. However, they missed the most important part of my remark, that was to point that their WB quantification does not show a significant increase in gamma H2AX accumulation contrarily to what they write. A change from 0.19 to 0.20 in the wild-type can hardly be considered as statistically relevant. There is no statistical test in Response table S1, but if authors

were to run one, I doubt they would conclude changes are statistically relevant. To me, it is a major ethical issue to write something in the text that is not supported by the data shown in the Figure. This behaviour shades doubts on all results. Besides, the WB and IF results on gamma H2AX are not consistent with each other. I have the exact same comment about the quantification of gamma H2AX shown on Figure 3.

Thanks for your perceptive comments and suggestions. We replaced the western results of Figure.1D with the results from the previous Revision Response Figure 5 responding to the reviewers, and added the p values on the statistical histogram. The statistical analysis results of the WB quantification showed that the increase in Col-0 was not significant, but the p-value was small and we indeed observed the slightly increased γ H2AX signals after CPT treatment in each replicate (see the response Figure 5 in the previous revision). The publication policy requested that the raw data will be provided to the journal, and we will do so accordingly.

In immunofluorescence assays, we focus on root meristem regions, because stem cells of this region are hypersensitive to DNA damage (PMID: 19933334), so the γ H2AX fluorescence signals were observed elevated after CPT treatment. While for the Western Blot, we collected the whole tissue of the root for the experiments, other cells that were not sensitive to DNA damage would average down the γ H2AX signal in the WB. As you could probably see the whole view of γ H2AX staining from Figure 6 and S5 of this revised version (Figure 8 and S4 of the previous revised version), a portion but not all cells have faithful fluorescence signals. Thus, the γ H2AX signal by CPT in WB is not obvious but still increased. Furthermore, the CPT treatment indeed induced the death of root stem cells from our results (PI staining in Figure 1B and Figure 3D of this version). Based on these observations, we finally concluded that “The γ H2AX signals in Col-0 and *atm* were increased after CPT treatment” in the text (line 132 in the main text of this version).

2- I understand the response made by authors about Figure S2C, ie that metaplots do not show a difference in R-loop levels between *atm* and Col0 because they averaged all protein-coding genes regardless of whether there was an increase in R-loops or not. However, I think this way of generating the figure is not very informative (besides it still does not explain the global decrease in R-loops that systematically appears in *atm* DMSO samples). It would be more informative to generate 2 sets of metaplots : one on genes that show more R-loops in *atm* CPT than in Col0, and one on genes that show less. This would allow appreciating the amplitude of the changes in the 2 cases. If authors are correct in writing that *atm* has fewer R-loops than Col0 under control conditions, it would be nice to see a scatter plot like the one shown in Figure S2B to confirm this, and also again, to see if the sites at which R-loop formation is modified are the same, or different, from the ones that vary in response to CPT.

Thanks for your suggestions. The metaplot results of Figure S2C only showed the R-loop levels on protein-coding genes, which are inconsistent with slot blot results (Figure 1C) and PCA results (Figure S2A) showing the genome-wide R-loops. We analyzed the R-loops on all annotated

elements (Figure S2B in this revised version), and the results showed R-loop levels in *atm* are higher than Col-0 near TSS sites, and CPT triggered R-loop accumulation at TSS in Col-0 and on gene body in *atm* (Figure S2B). In order not to cause any confusion, we had removed the metaplots of R-loops on protein-coding genes (Figure S2C of the previous version).

Response Figure 2. R-loop dynamics on protein-coding genes between different samples or treatments. (A) Scatterplots showing R-loop gene numbers that significantly changed between *atm* and Col-0. Normalized read counts are shown as $\log_{10}(n + 1)$. Red dots: $q\text{-value} < 0.05$, $\log_2\text{FC} > 1$; blue dots: $q\text{-value} < 0.05$, $\log_2\text{FC} < -1$; gray dots: other. (B) Venn diagram of up-regulated (left) or down-regulated R-loop genes (right) in different samples.

As you suggested, we analyzed the protein-coding genes with significantly different R-loop levels between Col-0 and *atm* in the control condition (Response Figure 2A). The scatterplot showed that there were indeed more genes in *atm* with significantly lower R-loop levels than Col-0 (Response Figure 2A), which may be the reason why the R-loops were globally decreased on protein-coding genes in *atm* compared with Col-0.

Moreover, we intersected the significantly up-regulated or down-regulated genes shown in Figure S2C of this version and Response Figure 2A. The number of the commonly and specifically regulated genes *atm* after CPT treatment compared to *atm* DMSO and Col-0 DMSO are comparable, but the genes with R-loop dynamics triggered by CPT are different with R-loop changed genes due to *atm* mutation (Response Figure 2B).

Also it would be interesting to know if the R-loops that increase in *atm* CPT vs Col-0 CPT are the same as the ones that increase in Col-0 CPT vs Col-0 DMSO.

In the same line as the remarks on Figure S2, on Figure 5, it would be nice to confirm the global decrease in R-loops in *asr20* mutants by generating a scatter plot in which they are compared to Col-0, to double check that results are consistent.

Thanks for the suggestions. We first analyzed the protein-coding genes with significantly different R-loop levels in Col-0 after CPT treatment. The scatterplot showed more genes with reduced R-

loops in Col-0 after CPT treatment (Response Figure 3A). The metaplot results showed increased R-loops of Col-0 were antisense R-loops which are mainly located near TSS, and the antisense R-loops are usually located on the upstream regions of TSS. And the genes locations used in scatterplot analysis exclude the upstream region of TSS, that could be the reason that more R-loops decreased genes are observed in Col-0 after CPT treatment. Then we intersected the significantly up-regulated or down-regulated genes in *atm* CPT vs Col-0 CPT and the ones in Col-0 CPT vs Col-0 DMSO (Response Figure 3B). The Venn plots showed few genes overlapped between them (Response Figure 3B), suggesting TOP1i triggered R-loop dynamics differ in *atm* mutant and WT background. Besides, we have generated the scatterplots of significantly differential R-loop genes in *asr20* and *pol2a^{L473F}* compared to Col-0 (Response Figure 3C). The numbers of decreasing R-loop genes are more than which are increased in both *asr20* and *pol2a^{L473F}*, and the number is greater in *asr20*.

Response Figure 3. R-loop dynamics on protein-coding genes between different samples or treatments. (A) Scatterplots showing R-loop gene numbers that significantly changed in Col-0 after CPT treatment. (B) Venn diagram of up-regulated (left) or down-regulated R-loop genes (right) between *atm* and Col-0 after CPT treatment. (C) Scatterplots showing R-loop gene numbers that significantly changed between *asr20* or *pol2a^{L473F}* and Col-0.

3- The additions to the legend of Figure 6 do help, however, I still cannot see what authors describe. For example, why are 2 peaks of R-loops drawn on all pannels except for *asr20* ? Globally, all plots seem very similar to me except I see much fewer lines in *asr20* and *pol2a^{L473F}* than in Col0 or *atm* (which does not match the clustering analysis shown on panel A). If anything, I see a reduction in signal intensity in *atm* + CPT vs *atm* DMSO, I certainly cannot make out the shifts of positions depicted on the cartoons (and authors forgot to specify if green is DMSO and orange CPT or the

other way around).

Thanks for the comments. Only 1 peak of R-loops is drawn on *asr20*, which intends to show that the number of R-loops on DNA origins in *asr20* is less than the other three samples, because we detected that R-loop levels are less in *asr20* than other backgrounds through slot blots and ssDRIP-seq (Figure 3E and Figure 4A of this revision). We are sorry about the missing descriptions that green is DMSO and orange is CPT. As all 3 reviewers suggested, we rearranged the manuscript more compactly, and drew the conclusions more carefully. As you were concerned, the heatmaps of Figure 6C are not suitable enough to verify the location shift of R-loops, we removed Figure 6C and the related results. The current results (Figure 4 E-G) are enough and statistical to support that POL2A could modulate R-loops by controlling DNA replication during the early replication steps.

4- The new Figure 7 is probably a better way of assessing the effect of the *pol2a* mutation on S-phase progression, but its interpretation is complex because it is the result of control of the G1/S transition and the S-phase progression. I wonder whether it wouldn't be easier to focus only on EdU + nuclei and to count the proportion of BrdU+ nuclei among those in each experimental set up. By doing this, authors are sure to focus on cells that are in S at the beginning of the experiment, and to test whether they have reached G2 or not by the time they perform the BrdU pulse. Dual labelling means they are still in S at the time of the BrdU pulse. The longer cells spend into S-phase, the higher this proportion should be after a long chase interval. If data are looked at this way, the conclusions of the experiment may change. Indeed, *atm* and *asr20* have much more dually labelled cells than *Col0* in the presence of CPT, which would indicate they lag into S longer. It is true however, that *asr20* mutants seem to progress through S-phase faster than *atm*, but all this is only a visual impression of course because I don't have the raw numbers. Analyzing the data in this manner would probably make the text accompanying the Figure easier to read. One point that I cannot understand is the behaviour of BrdU-only nuclei. Why are they completely absent in *Col-0* + CPT after a 2h thymidine chase ? This would suggest entry into S-phase is completely blocked by this treatment, which complexifies the picture.

Besides, this lacks replicates and statistical analysis.

Thanks for the comments and suggestions. As you may find that these double labeling experiments were performed to answer the previous question of Reviewer #1. We had spent a long time performing the DNA fiber spreading assay (see more details in the first revision), however, we failed to set up this experiment in our system. DNA spreading assay is rarely done in plants. Alternatively, we adopted the double-labeling strategy referred to "Distinct roles of Arabidopsis ORC1 proteins in DNA replication and heterochromatic H3K27me1 deposition" (PMID: 36882445) to interpret a more detailed replication progression of WT and mutant polymerase in vivo. Please find more details in our previous response file.

The three types of labelled nuclei reflect that the cells are at different stages of S-phase progression. BrdU+ only nuclei represent cells that have just entered the S-phase and are ready to replicate; EdU+

BrdU+ nuclei represent cells that are already in the S-phase; while EdU+ only nuclei represent cells that have completed replication and entered into the G2 phase. These three types of cells indicate the start, the middle, and the end of S-phase progression. For example, if a higher percentage of EdU+ BrdU+ nuclei in Col-0 after CPT treatment was detected, it could be caused by a delay of S-phase initiation, S-phase transition into G2, or both. Combining the results of EdU+ or BrdU+ only, we could determine the status of S-phase progress. We think this data is much more informative than EdU or BrdU staining only.

Take Figure 5D as an example, as you mentioned: “*atm* and *asr20* have much more dually labelled cells than Col0 in the presence of CPT, which would indicate they lag into S longer.” Combining the results of the other two single labelled nuclei showed that more BrdU+ only but less EdU+ only in *atm* compared to Col-0, it is clearly suggested that DNA replication (middle of S-phase) is not inhibited by CPT, and accomplishment of S-phase progression but not initiation is delayed in *atm*. While both the BrdU+ only and EdU+ only nuclei are less in *asr20* compared to Col-0, suggesting DNA replication is not inhibited but the start and end of S-phase progression is delayed in *asr20*. For “BrdU+ only nuclei are completely absent in Col-0 after 2h CPT treatment”, combining the proportion of double labelled nuclei is low, we conclude that the initiation of S-phase is severely inhibited by CPT in Col-0, so the BrdU+ only nuclei are completely absent.

For the replicates and statistical analysis, we added the related information in the legend. This dual labelling experiment followed the methods in the published paper (PMID: 36882445), we had analysed more than 300 nuclei from 2-3 roots in each background, and the mean values of 2-3 roots in each case were used for graph making. And here, we showed the statistical analysis results of these three types of nuclei in each sample (Response Figure 4). We could provide this data in the main figure if necessary, but it will take more space for the main Figures.

Response Figure 4. The statistical analysis results related to Figure 5. The percentages of three types nuclei with three chase time point in each sample were quantified separately. 2–3 roots were scored for each sample. Bars in the plot are the mean \pm SD.

5- The experiment on Figure S12 lacks quantification, I cannot see any difference on the gels, contrarily to what authors describe in the text.

Thanks for the suggestions. We added the quantification values above the bands, there is a slight elevation of DNA signals which synthesized with POL2A^{WT} compared with POL2A^{L473F} (Figure S12C), and the results are more obvious in lane 4 compared to lane 7, lane 9 compared to lane 11, and lane10 compared to lane 12 (Figure S12C).

6- Regarding the experiment with rad52 and ku70, although authors claim they have corrected the text, they still write « both atm/rad52 and atm/ku70 exhibited CPT resistance that the roots were not severely inhibited as atm » this is not true, only atm rad52 perform better than atm on CPT. These repeated inaccurate statements in the text are really worrying.

Thanks for your comments. The reason why we wrote “both *atm/rad52* and *atm/ku70* exhibited CPT resistance that the roots were not severely inhibited as *atm*” is that the *atm/ku70* exhibited comparable root length between DMSO and CPT (Figure 8E), while *atm* root length was severely inhibited by CPT. Thus, we wanted to conclude that *atm/ku70* exhibited CPT resistance and its roots were not severely inhibited by CPT, this performance is different from that in *atm*. To make it much clearer, we also calculated the ratio of root length before and after CPT (Response Figure 5), and from these results, we could conclude that the roots of *atm/rad52* and *atm/ku70* are much less inhibited by CPT compared to *atm*.

Response Figure 5. The root length ratio of different samples after CPT treatment. The mean values of root length in each sample as shown in Figure 8F of this revised version are used for ratio value. The raw data including the p value were shown in Figure 8F.

To avoid the misunderstanding, we corrected this sentence to “However, *rad52* and *ku70* presented insensitivity to CPT, and the roots of both *atm/rad52* and *atm/ku70* were slightly inhibited by CPT, which is different from the *atm* single mutant” (line 391-392 of this revision).

7- There must be a mistake in the methods paragraph describing statistical analysis. Most experiments compare multiple genotypes, which requires an ANOVA for sure, but followed by a post-hoc test such as Tukey. This is not indicated.

Thanks for your suggestions. We supplemented the information in the methods about the statistical analysis (line 646-647).

8- The discussion has been re-written, but the working model still needs to be better supported by the data (notably the hypothesis that replication speed is modified in different genetic contexts).

Thanks for your suggestions. As all of you suggested, we revised the manuscript and modified the working model (Figure 9 in this revised version). We replaced the hypothesis of DNA replication

speed with S-phase progression which is supported by Figure 5 in this revised version.

REVIEWER COMMENTS

Reviewer #2 (Remarks to the Author):

Overall I am satisfied with the revisions carried out by the authors- they have addressed my comments. The paper is suitable for publication in Nature Comm journal.

Reviewer #3 (Remarks to the Author):

Authors have addressed some of my comments, but I am afraid some issues remain.

Major points :

1-The description of Figure S2 is still a bit confusing. Authors write « Upon CPT treatment, R-loop levels increased in Col-0 ». There is indeed a very modest increase visible on the metaplot. Yet, when they compare R-loops on coding genes, there are more genes showing a significant decrease in R-loops than genes showing an increase in Col0 after CPT treatment. This would mean that there is a massive increase in R-loops at some few loci, and a modest decrease on a larger number of coding genes? Or most of the R-loops induced by CPT in Col0 are on non-coding sequences?

2-L132-134 « The γ H2AX signals in Col-0 and atm were increased after CPT treatment, while the γ H2AX levels of atm in the control condition were lower than that of Col-0 (Fig. 1D-E and Fig. S1D) » I understand the authors' response to my comment. For sure a WB performed on whole plantlets is less likely to reveal an increase in gamma-H2AX than IF on root tips. This does not mean it is acceptable to write this sentence. Authors can state that by WB they see a tendency towards the increase of gamma H2AX accumulation that is reproducible, but not statistically significant, and discuss that IF is probably more reliable (although quantification of IF is tricky), but they have to acknowledge that their WB DOES NOT show a significant increase in gamma-H2AX. Otherwise, anybody can describe figures in a way that suits their working hypothesis instead of sticking to what the figure actually shows.

3-About Figure 5 and S10:

First of all, Figure S10 requires quantification, because differences described in the text are not obvious. Besides, it would help with the interpretation of Figure 5 that remains extremely complex, because the quantified proportions depend of the regulation of G1/S transition, S progression and G2/M transition that could all be affected by the treatment and mutations in different ways. Actually, I think one difficulty with the experiment presented on Figure 5 is that graphs show only labelled cells, and not unlabelled ones. So between samples, the proportion 1 does not in fact correspond to the same number of nuclei nor does it correspond to the same proportion of the total meristem. One important control would be to check if the proportion of EdU+ cells in the meristem is the same for a given genotype regardless of the treatment (0, 1 or 2h of Thymidine and with or without CPT), this would prove that EdU+ cells were labelled almost exclusively during the pulse and not during the chase, and that we are actually looking at the same cells in all samples. This is critical especially to understand what is happening with the CPT treated samples. In the case of control samples, in the absence of CPT, I can see two points that may deserve to be amended.

First, the variations in the smallest proportions of the samples should be commented with caution. For example, on Figure 5B, the proportion of EdU+ BrdU- nuclei in Col0 appears to be around 0.02 or 0.03 from what I can see on the graph. If the total number of nuclei is 300, this corresponds to less than 10. This makes me wonder whether the difference with other genotypes is really to be trusted, so authors should be cautious when commenting it. This leads me to my second point : authors may want to perform simpler comparisons to make the data easier to interpret. This is what I was suggesting in my previous comment. Indeed, I totally agree using this dual labelling strategy makes a lot of sense to determine whether the pol2A mutation slows-down or accelerates S-phase progression.

Single labelling would indeed not allow them to conclude. However, in the paper the authors cite whether this technique is used, the dual labelling is used to identify G2 cells in a single genotype, and in the absence of genotoxic stress. Here, authors are trying to compare proportions of cells in different cell cycle stages. The problem to me, is that they count BrdU+, EdU+ and EdU+ BrdU+ cells, but not double negative cells. So 100% is potentially not the same number of cells in all genotypes, because this technique is blind to late G2 and G1 cells. This is why I was suggesting authors could simply count the proportion of BrdU+ cells amongst the EdU+ cells. In this way, they would be able to focus on cells that are in S, regardless of the fact that the proportion of cells in S at any time point is not the same in all genotypes, and question only how fast cells exit from S. This would be easier to read, and also easier to interpret without seeing confounding effects due to a block in the G1/S transition for example. Going back to their data, this type of comparison done on control samples would show that after 1h chase, about 46% of the cells that were in S when EdU was added are no longer in S when BrdU is added in Col0 (and this proportion reaches almost 60% after 2h, which makes sense). This percentage would be about 67% in atm after 1h, meaning that S-phase speed is increased (possibly because normal checkpoints are not properly activated), and about 39% in pol2AL473F, which would confirm that the effect of the point mutation in pol2A alone is to slow down S-phase progression. Surprisingly, this proportion is even lower in asr20 (about 33%), which indicates that atr is likely responsible for this slowing-down of S-phase progression due to the pol2A mutation. The same calculations on Figure 1F would lead to the same conclusion, and would, to me be much easier to follow.

However, I agree that this calculation cannot be applied to the CPT+ samples, which leads me to another issue. Now in Col0, the proportion of EdU+ only cells out of the total EdU+ population is at least 55% after 1h, which is more than in control conditions and would suggest S-phase progression is faster, a rather counter-intuitive result. Things become even worse after 2h on CPT since almost no dually labelled cells are observed in Col0. Authors appear to consider that in these conditions, EdU+ only cells can be interpreted either as cells that progressed from S to G2 or as cells that remained stuck in S and ceased replicating completely. This would explain the surprising behaviour of the wild-type, but if it is correct, it becomes quite difficult to conclude anything from the data, because a change in the proportion of EdU+ only cells can mean S is faster or slower. Indeed, after 2h of CPT treatment, the complete absence of BrdU+ only cells in Col0 suggests that S-phase entry (and possibly progression) is almost completely stopped. But after 1h of treatment, Col0 plants have more BrdU+ only cells than untreated samples, this does not make sense, but may stem from the fact that the number of BrdU only cells is very small and variations are in some instances simply random. Could author specify which statistical test they performed on response Figure 4 ? They have to make sure that the test is adapted to proportions and are not based on the assumption that the values follow a normal distribution. Looking at the values on Response Figure 4 after 1h, I find it hard to believe that using the correct test would lead to the conclusion that the proportion of BrdU+ only cells differs significantly between Col0 and pol2AL473F for example.

Minor points

L128-129, I would write « In addition, these topoR-loops do not correlate with mRNA levels after CPT treatment, suggesting that they do not result from differences in gene expression ». RNAseq does not measure transcription

L194-195 : « These results suggest the conserved junction domain of POL2A could have an essential role in releasing topological stress during replication ». If this junction domain played a role in releasing topological stress, mutations in this domain would be expected to aggravate topological stress rather than dampen it as observed here, this sentence should be rephrased.

L264-265 « and that the allelic mutant til1-4 (G469R) shows aberrant cell cycle features » This is not unique to til1-4: aberrant cell cycle features are also observed in abo4-1.

About Figure 8

I apologize for misunderstanding what authors meant in the text. If authors they wish to comment on relative root growth, maybe the graph shown in Response Figure 5 could appear in the manuscript,

with appropriate error bars and statistical analyses. However, I think they should be careful in the interpretation of the results. Since *atm rad52* grows better than *atm* alone on CPT, this could suggest that failure to delay S-phase progression in response to TRC leads to HR events that are potentially deleterious to the cells. The observation that *atm ku70* is very small already on control medium could mean that *ku70* is essential to repair TRCs in the absence of *atm*, and that loss of *KU70* somehow has a similar effect on *atm* as CPT treatment. In all cases, the two double mutants seem to behave very differently, and so I would not jump to the conclusion that HR and NHEJ function together to repair CPT-induced breaks. If they did, one would expect mutants lacking these repair pathways to be more sensitive than the wild type to CPT. They may function redundantly, but again, this would require further investigation, and this is not a key point of the manuscript to me.

We would like to express our sincere gratitude to Reviewer #3's insightful and valuable comments, which have helped us to further enhance the quality of our research. In this revised version, we have addressed all the issues and provided responses to all inquiries.

Reviewer #3 (Remarks to the Author):

Authors have addressed some of my comments, but I am afraid some issues remain.

Major points :

1-The description of Figure S2 is still a bit confusing. Authors write « Upon CPT treatment, R-loop levels increased in Col-0 ». There is indeed a very modest increase visible on the metaplot. Yet, when they compare R-loops on coding genes, there are more genes showing a significant decrease in R-loops than genes showing an increase in Col0 after CPT treatment. This would mean that there is a massive increase in R-loops at some few loci, and a modest decrease on a larger number of coding genes? Or most of the R-loops induced by CPT in Col0 are on non-coding sequences?

Thank you for your comments. As we previously mentioned in the second part of the response to Comment 2, “The metaplot results showed increased R-loops of Col-0 were antisense R-loops which are mainly located near TSS, and the antisense R-loops are usually located on the upstream regions of TSS. And the gene locations used in scatterplot analysis exclude the upstream region of TSS, which could be the reason that more R-loops decreased genes are observed in Col-0 after CPT treatment.” Moreover, 9033 coding genes in Col-0 show significantly changed R-loops upon CPT treatment ($q\text{-value} < 0.05$), of which 6157 are increased and 2876 are decreased (Response Figure 1). This aligns with the metaplot results that a slight increase of R-loops on gene body after CPT treatment in Col-0 from metaplot analysis, and supports our claim that CPT treatment globally increased R-loop levels in Col-0 by slot blot. We set a very strict threshold ($q\text{-value} < 0.05$, $|\log_2\text{FC}| > 1$) in the scatter plot (Fig.S2E) to identify the genes with significantly changed R-loops. Applying this criterion, the number of coding genes with significant R-loop increased (64) or decreased (543) in Col-0 is relatively small compared with *atm* (4158 increased, 1647 decreased) after CPT treatment. Besides, considering the distinct R-loop dynamic patterns observed in Col-0 and *atm* in response to CPT, it becomes evident that the scatter plot results for Col-0 may not fully capture the global dynamics, as seen in the case of *atm*.

Response Figure 1. Volcano plot showing R-loop gene numbers that significantly changed in Col-0 after CPT treatment. Red dots: q-value < 0.05.

2-L132-134 « The γ H2AX signals in Col-0 and *atm* were increased after CPT treatment, while the γ H2AX levels of *atm* in the control condition were lower than that of Col-0 (Fig. 1D-E and Fig. S1D) » I understand the authors' response to my comment. For sure a WB performed on whole plantlets is less likely to reveal an increase in gamma-H2AX than IF on root tips. This does not mean it is acceptable to write this sentence. Authors can state that by WB they see a tendency towards the increase of gamma H2AX accumulation that is reproducible, but not statistically significant, and discuss that IF is probably more reliable (although quantification of IF is tricky), but they have to acknowledge that their WB DOES NOT show a significant increase in gamma-H2AX. Otherwise, anybody can describe figures in a way that suits their working hypothesis instead of sticking to what the figure actually shows.

Thanks for your suggestions. We modified the sentence (Line 133-137) to “There is a tendency, but not statistically significant, towards the increase of γ H2AX accumulation in Col-0 and *atm* after CPT treatment by western blot, while the γ H2AX levels of *atm* in the control condition showed a decreasing trend compared to Col-0 (Fig. 1D). The immunostaining results confirmed that CPT increased γ H2AX signals in Col-0 and *atm* root tips (Fig. 1E and Fig. S1D),”

3-About Figure 5 and S10:

First of all, Figure S10 requires quantification, because differences described in the text are not obvious. Besides, it would help with the interpretation of Figure 5 that remains extremely complex, because the quantified proportions depend of the regulation of G1/S transition, S progression and G2/M transition that could all be affected by the treatment and mutations in different ways.

Thanks for your comments and suggestions. Firstly, we supplemented the quantification results of Figure S10A (in this revision) as shown in Figure S10B (in this revision). The results are consistent with our claims that *asr20* and *plo2a*^{L473F} had more EdU+ nuclei than Col-0 and *atm* in control condition, and CPT treatment significantly inhibited DNA replication in Col-0 and *plo2a*^{L473F} but slightly in *atm*. The cell cycle progression varies in mutants compared with Col-0 after CPT removal (FigS10B in this revision). Also, we modified the description of the results of Figure S10 in this revision (Line 275-283).

Actually, I think one difficulty with the experiment presented on Figure 5 is that graphs show only labelled cells, and not unlabelled ones. So between samples, the proportion 1 does not in fact correspond to the same number of nuclei nor does it correspond to the same proportion of the total meristem. One important control would be to check if the proportion of EdU+ cells in the meristem is the same for a given genotype regardless of the treatment (0, 1 or 2h of Thymidine and with or without CPT), this would prove that EdU+ cells were labelled almost exclusively during the pulse and not during the chase, and that we are actually looking at the same cells in all samples. This is critical especially to understand what is happening with the CPT treated samples.

Thanks for your comments, and we had modified the graph performance as you suggested. EdU+ BrdU+ (yellow) indicates the cells that are processing the S-phase, and EdU+ BrdU- (green) indicates the cells that exited from the S-phase before BrdU labeling. In the revision, we focus on the cells that are in S, and calculate the ratio of EdU+ BrdU+ cells in the EdU+ BrdU+ plus EdU+

BrdU- cells, which could reflect the S-phase progression that the higher ratio ('EdU+ BrdU+' / 'EdU+ BrdU+ plus EdU+ BrdU-') indicates slower S-phase progression.

In the case of control samples, in the absence of CPT, I can see two points that may deserve to be amended.

First, the variations in the smallest proportions of the samples should be commented with caution. For example, on Figure 5B, the proportion of EdU+ BrdU- nuclei in Col0 appears to be around 0.02 or 0.03 from what I can see on the graph. If the total number of nuclei is 300A, this corresponds to less than 10. This makes me wonder whether the difference with other genotypes is really to be trusted, so authors should be cautious when commenting it.

Thanks for your comments. We had performed three replicates for this assay, and indeed observed several EdU+ BrdU- nuclei only in Col-0 DMSO sample for each replicate (Figure 5B), even though the number of this kind of nuclei is low.

This leads me to my second point : authors may want to perform simpler comparisons to make the data easier to interpret. This is what I was suggesting in my previous comment. Indeed, I totally agree using this dual labelling strategy makes a lot of sense to determine whether the pol2A mutation slows-down or accelerates S-phase progression. Single labelling would indeed not allow them to conclude. However, in the paper the authors cite whether this technique is used, the dual labelling is used to identify G2 cells in a single genotype, and in the absence of genotoxic stress.

Thanks for your comments. The double-labeling strategy is used to evaluate S-phase progression, not to identify the G2 cells. As mentioned in the reference (PMID: 36882445): "When the two labeling pulses are consecutive, most cells would appear dual-labeled. The dual-labeling percentage will progressively decrease by increasing chasing times between the BrdU and EdU pulses, depending on the number of cells finishing the S-phase before the second pulse. Thus, measuring the fraction of nuclei with the two labels after different chasing times between the two pulses provides a useful indication of S-phase progression". They used dual-labeling to identify S-phase cells in a single genotype without genotoxic stress, and by comparison with wild-type and *orc1a-2* mutant plants, they concluded that S-phase progression in the *orc1b-2* mutant is delayed (PMID: 36882445).

It is also applicable in our case even after CPT treatment, our comparisons involved either contrasting the mutants with CPT-treated Col-0 or evaluating the same sample under different treatment conditions, with a consistent focus on a single variable, whether it be the genotype or treatment. Due to that the proportion of cells in S-phase is different in various genotypes, we used the gradually increased chase time to detect the changes of S-phase progress.

Here, authors are trying to compare proportions of cells in different cell cycle stages. The problem to me, is that they count BrdU+, EdU+ and EdU+ BrdU+ cells, but not double negative cells. So 100% is potentially not the same number of cells in all genotypes, because this technique is blind to late G2 and G1 cells. This is why I was suggesting authors could simply count the proportion of BrdU+ cells amongst the EdU+ cells. In this way, they would be able to focus on cells that are in S, regardless of the fact that the proportion of cells in S at any time point is not the same in all genotypes, and question only how fast cells exit from S. This would be easier to read, and also easier to interpret without seeing confounding effects due to a block in the G1/S transition for example.

Going back to their data, this type of comparison done on control samples would show that after 1h chase, about 46% of the cells that were in S when EdU was added are no longer in S when BrdU is added in Col0 (and this proportion reaches almost 60% after 2h, which makes sense). This percentage would be about 67% in atm after 1h, meaning that S-phase speed is increased (possibly because normal checkpoints are not properly activated), and about 39% in pol2AL473F, which would confirm that the effect of the point mutation in pol2A alone is to slow down S-phase progression. Surprisingly, this proportion is even lower in asr20 (about 33%), which indicates that atr is likely responsible for this slowing-down of S-phase progression due to the pol2A mutation. The same calculations on Figure 1F would lead to the same conclusion, and would, to me be much easier to follow.

Thanks for the suggestions and we modified the graph to focus on the cells that are in S (all EdU+ cells) as we mentioned above. Furthermore, the cells that have completed division and re-entered G1 will be blind to the cells in G2/M as the chase time increases. The duration of S-phase and G2+M phase in the Arabidopsis root tip is estimated to be 1-2 hours and 2-4 hours, respectively (PMID: 27697785), therefore the chase time we used is up to 2 hours can avoid this issue as much as possible.

However, I agree that this calculation cannot be applied to the CPT+ samples, which leads me to another issue. Now in Col0, the proportion of EdU+ only cells out of the total EdU+ population is at least 55% after 1h, which is more than in control conditions and would suggest S-phase progression is faster, a rather counter-intuitive result. Things become even worse after 2h on CPT since almost no dually labelled cells are observed in Col0. Authors appear to consider that in these conditions, EdU+ only cells can be interpreted either as cells that progressed from S to G2 or as cells that remained stuck in S and ceased replicating completely. This would explain the surprising behaviour of the wild-type, but if it is correct, it becomes quite difficult to conclude anything from the data, because a change in the proportion of EdU+ only cells can mean S is faster or slower. Indeed, after 2h of CPT treatment, the complete absence of BrdU+ only cells in Col0 suggests that S-phase entry (and possibly progression) is almost completely stopped. But after 1h of treatment, Col0 plants have more BrdU+ only cells than untreated samples, this does not make sense, but may stem from the fact that the number of BrdU only cells is very small and variations are in some instances simply random. Could author specify which statistical test they performed on response Figure 4 ? They have to make sure that the test is adapted to proportions and are not based on the assumption that the values follow a normal distribution. Looking at the values on Response Figure 4 after 1h, I find it hard to believe that using the correct test would lead to the conclusion that the proportion of BrdU+ only cells differs significantly between Col0 and pol2aL473F for example.

Thank you for your comments. CPT treatment inhibits DNA replication, as illustrated in Figure S10, and the proportion of EdU+ only cells reflects the initiation of S-phase or newly initiated DNA replication. It is convinced that S-phase progression, including progress and initiation, as indicated by the presence of EdU+BrdU+ co-labeled and BrdU+ only cells, was severely inhibited after 2h with CPT treatment in Col-0 (Fig. 5E-F). RNA:DNA hybrids form in Okazaki fragments when DNA is replicating and R-loops could promote replication firing (PMID: 32107311, PMID: 34494544 and PMID: 25972891). Combining our results that CPT strengthened R-loop increase and enrichment at the DNA origins (Fig. 4D-E), R-loops may create a bubble structure conducive to DNA replication initiation. Additionally, R-loops play a role in the repair and restart of DNA replication fork (PMID: 35459910, PMID: 31759821). After 1 hour of treatment, our data showed

that, Col-0 and mutants exhibited more BrdU+ only cells in the presence of CPT compared to DMSO. This increase may be attributed to the accumulation of R-loops near DNA origins, resulting in uncontrolled origin firing. However, after 2 hours, the percentage of BrdU+ only cells decreased, except in *atm* mutant. This decrease suggests the activation of the DNA replication checkpoint and the subsequent control of abnormal replication initiation, and this activation is dependent on functional ATM. In Figure 4, the statistics were conducted by using pairwise t-tests without conducting homogeneity of variance analysis. Therefore, calculating the BrdU+ only cells does not affect our primary conclusions. To avoid misleading, we removed the data about the BrdU+ only cells and focused on the EdU + cells as we mentioned above.

In all, as you suggested, it is easier to read and interpret that we only simply count the proportion of BrdU+ cells amongst the EdU+ cells, we re-analyzed the proportion results as you commended (Figure 5B, 5D, 5F in this revision) and modified the description of Figure 5 accordingly (Line 297-320).

Minor points

L128-129, I would write « In addition, these topoR-loops do not correlate with mRNA levels after CPT treatment, suggesting that they do not result from differences in gene expression ». RNAseq does not measure transcription

Thanks for your suggestions. We modified the sentence as you suggested (Line 128-130).

L194-195 : « These results suggest the conserved junction domain of POL2A could have an essential role in releasing topological stress during replication ». If this junction domain played a role in releasing topological stress, mutations in this domain would be expected to aggravate topological stress rather than dampen it as observed here, this sentence should be rephrased.

Thanks for your suggestions. We changed the word 'release' to 'modulate', and moved this sentence to the end of this part (after the root length results), as shown in the revised text (203-206), 'These results suggest the conserved junction domain of POL2A could have an essential role in modulating topological stress during replication, and residue mutations in the junction and DNA polymerase domain of POL2A have conserved effects of varying degrees on the release of topological stress and then the topoR-loops, which is in line with the location of the mutations in the functional domain of the POL2A protein (Fig. S6G)'.

L264-265 « and that the allelic mutant *till-4* (G469R) shows aberrant cell cycle features » This is not unique to *till-4*: aberrant cell cycle features are also observed in *abo4-1*.

Thanks for your comments. We modified the sentence (Line 269) to "the allelic mutant *till-4* (G469R) and *abo4-1* (G522N) show aberrant cell cycle features" (PMID: 16212602 and PMID: 28153919).

About Figure 8

I apologize for misunderstanding what authors meant in the text. If authors they wish to comment on relative root growth, maybe the graph shown in Response Figure 5 could appear in the manuscript, with appropriate error bars and statistical analyses. However, I think they should be careful in the interpretation of the results. Since *atm rad52* grows better than *atm* alone on CPT, this could suggest

that failure to delay S-phase progression in response to TRC leads to HR events that are potentially deleterious to the cells. The observation that *atm ku70* is very small already on control medium could mean that *ku70* is essential to repair TRCs in the absence of *atm*, and that loss of *KU70* somehow has a similar effect on *atm* as CPT treatment. In all cases, the two double mutants seem to behave very differently, and so I would not jump to the conclusion that HR and NHEJ function together to repair CPT-induced breaks. If they did, one would expect mutants lacking these repair pathways to be more sensitive than the wild type to CPT. They may function redundantly, but again, this would require further investigation, and this is not a key point of the manuscript to me.

Thanks for your comments and suggestions. We added the result in this revision (Figure S13D). As you mentioned above, failure to delay S-phase progression in response to TRC may lead to HR events that are potentially deleterious to the cells, which results in *atm/rad52* showing longer roots than *atm* but shorter than Col-0 under CPT. *atm/ku70* is shorter and smaller than *atm* and Col-0 under control condition, suggesting *KU70* is essential to repair DNA damages which may be caused by TRCs during normal growth and development in the absence of *ATM*. More DNA damages caused by CPT or failures of DNA damage repairs by *ku70* mutations show similar inhibition effects (*atm*_{CPT} vs *atm/ku70*_{DMSO}), suggesting *KU70* functions in CPT-induced DNA damage repair. Combining the introduction in the manuscript “HR is a high-fidelity but template-dependent repair pathway that is restricted in S phase, while NHEJ functions on a wide range of DNA-end configurations in a cell cycle-independent manner”, we modified the exposition of these results as you suggested: “TOP1i-increased topoR-loops are probably persistent and cause DNA damage throughout the cell cycle, with HR and NHEJ involved to repair the breaks” (Line 403-404).